# Estimates of Lightning NO$_x$ Production based on High Resolution OMI NO$_2$ Retrievals over the Continental US

Xin Zhang[1,2], Yan Yin[1,2], Ronald van der A[2,3], Jeff L. Lapierre[4], Qian Chen[1,2], Xiang Kuang[1,2], Shuqi Yan[2], Jinghua Chen[1,2], Chuan He[1,2], and Rulin Shi[1,2]

[1] Collaborative Innovation Center on Forecast and Evaluation of Meteorological Disasters/Key Laboratory for Aerosol-Cloud-Precipitation of China Meteorological Administration, Nanjing University of Information Science and Technology (NUIST), Nanjing 210044, China

[2] Department of Atmospheric Physics, Nanjing University of Information Science and Technology (NUIST), Nanjing 210044, China

[3] Royal Netherlands Meteorological Institute (KNMI), Department of Satellite Observations, De Bilt, the Netherlands

[4] Earth Networks, Germantown, Maryland, USA

**Correspondence:** Yan Yin (yinyan@nuist.edu.cn)

**Abstract.** Lightning serves as the dominant source of nitrogen oxides (NO$_x$ = NO + NO$_2$ ) in the upper troposphere (UT), with strong impact on ozone chemistry and the hydroxyl radical production. However, the production efficiency (PE) of lightning nitrogen oxides (LNO$_x$) is still quite uncertain (32 – 1100 mol NO per flash). Satellite measurements are a powerful tool to estimate LNO$_x$ directly as compared to conventional platforms. To apply satellite data in both clean and polluted regions, a

new algorithm for calculating LNO$_x$ has been developed that uses the Berkeley High Resolution (BEHR) v3.0B NO$_2$ retrieval algorithm and the Weather Research and Forecasting-Chemistry (WRF-Chem) model. LNO$_x$ PE over the continental US is estimated using the NO$_2$ product of the Ozone Monitoring Instrument (OMI) data and the Earth Networks Total Lightning Network (ENTLN) data. Focusing on the summer season during 2014, we find that the lightning NO$_2$ (LNO$_2$) PE is 32 $\pm$ 15 mol NO$_2$ flash$^{-1}$ and 6 $\pm$ 3 mol NO$_2$ stroke$^{-1}$ while LNO$_x$ PE is 90 $\pm$ 50 mol NO$_x$ flash$^{-1}$ and 17 $\pm$ 10 mol NO$_x$ stroke$^{-1}$.

Results reveal that our method reduces the sensitivity to the background NO$_2$ and includes much of the below-cloud LNO$_2$. As the LNO$_x$ parameterization varies in studies, the sensitivity of our calculations to the setting of the amount of lightning NO (LNO) is evaluated. Careful consideration of the ratio of LNO$_2$ to NO$_2$ is also needed, given its large influence on the estimation of LNO$_2$ PE.

## 1   Introduction

Nitrogen oxides (NO$_x$) near the Earth's surface are mainly produced by soil, biomass burning and fossil fuel combustion, while NO$_x$ in the middle and upper troposphere originates largely from lightning and aircraft emissions. NO$_x$ plays an important role in the production of ozone (O$_3$) and the hydroxyl radical (OH). While the anthropogenic sources of NO$_x$ are largely known, lightning nitrogen oxides (LNO$_x$) are still the source with the greatest uncertainty, though they are estimated to range between 2 and 8 Tg N yr$^{-1}$ (Schumann and Huntrieser, 2007). LNO$_x$ is produced in the upper troposphere (UT) by O$_2$ and N$_2$ dissociation

in the hot lightning channel as described by the Zel'dovich mechanism (Zel'dovich and Raizer, 1967). With the recent updates

of UT $NO_x$ chemistry, the day time lifetime of UT $NO_x$ is evaluated to be $\sim 3$ h near thunderstorms and $\sim 0.5 - 1.5$ days away from thunderstorms (Nault et al., 2016, 2017). This results in enhanced $O_3$ production in the cloud outflow of active convection (Pickering et al., 1996; Hauglustaine et al., 2001; DeCaria et al., 2005; Ott et al., 2007; Dobber et al., 2008; Allen et al., 2010; Finney et al., 2016). As $O_3$ is known as a greenhouse gas, strong oxidant and absorber of ultraviolet radiation (Myhre et al.,
2013), the contributions of $LNO_x$ to $O_3$ production also have an effect on climate forcing. Finney et al. (2018) found different impacts on atmospheric composition and radiative forcing when simulating future lightning using a new upward cloud ice flux (IFLUX) method versus the commonly used cloud-top height (CTH) approach. While global lightning is predicted to increase by 5 — 16% over the next century with the CTH approach (Clark et al., 2017; Banerjee et al., 2014; Krause et al., 2014), a 15% decrease in global lightning was estimated with IFLUX in 2100 under a strong global warming scenario (Finney
et al., 2018). As a result of the different effects on radiative forcing from ozone and methane, a net positive radiative forcing was found with the CTH approach while there is little net radiative forcing with the IFLUX approach (Finney et al., 2018). However, the convective available potential energy (CAPE) times the precipitation rate (P) proxy predicts a $12 \pm 5\%$ increase in the Continental US (CONUS) lightning strike rate per kelvin of global warming (Romps et al., 2014), while the IFLUX proxy predicts the lightning will only increase 3.4%/K over the CONUS. Recently, Romps (2019) compared the CAPE $\times$ P
proxy and IFLUX method in cloud-resolving models. They reported that higher CAPE and updraft velocities caused by global warming could lead to the large increases in tropical lightning simulated by CAPE $\times$ P proxy, while IFLUX proxy predicts little change in tropical lightning because of the small changes in the mass flux of ice.

In the view of the regionally dependent lifetime of $NO_x$ and the difficulty of measuring $LNO_x$ directly, a better understanding of the $LNO_x$ production is required, especially in the tropical and mid-latitude regions in summer. Using its distinct
spectral absorption lines in the near-ultraviolet (UV) and visible (VIS) range (Platt and Perner, 1983), $NO_2$ can be measured by satellite instruments like the Global Ozone Monitoring Experiment (GOME; Burrows et al., 1999; Richter et al., 2005), Scanning Imaging Absorption Spectrometer for Atmospheric Chartography (SCIAMACHY; Bovensmann et al., 1999), the Second Global Ozone Monitoring Experiment (GOME-2; Callies et al., 2000) and the Ozone Monitoring Instrument (OMI; Levelt et al., 2006). OMI has the highest spatial resolution, least instrument degradation and longest record among these satellites
(Krotkov et al., 2017). Satellite measurements of $NO_2$ are a powerful tool compared to conventional platforms, because of its global coverage, constant instrument features and temporal continuity.

Recent studies have determined and quantified $LNO_x$ using satellite observations. Beirle et al. (2004) constrained the $LNO_x$ production to 2.8 (0.8 – 14) Tg N yr$^{-1}$ by combining GOME $NO_2$ data and flash counts from the Lightning Imaging Sensor (LIS) aboard the Tropical Rainfall Measurement Mission (TRMM) over Australia. Boersma et al. (2005) estimated the global
$LNO_x$ production of 1.1 – 6.4 Tg N yr$^{-1}$ by comparing GOME $NO_2$ with distributions of $LNO_2$ modeled by Tracer Model 3 (TM3). Martin et al. (2007) analyzed SCIAMACHY $NO_2$ columns with Goddard Earth Observing System chemistry model (GEOS-Chem) simulations to identify $LNO_x$ production amounting to $6 \pm 2$ Tg N yr$^{-1}$.

As these methods focus on monthly or annual mean $NO_2$ column densities, more recent studies applied specific approaches to investigate $LNO_x$ directly over active convection. Beirle et al. (2006) estimated $LNO_x$ as 1.7 (0.6 – 4.7) Tg N yr$^{-1}$ based on
a convective system over the Gulf of Mexico, using National Lightning Detection Network (NLDN) observations and GOME

NO₂ column densities. However, it is assumed that all the enhanced NO$_2$ originated from lightning and did not consider the contribution of anthropogenic emissions. Beirle et al. (2010) analyzed LNO$_x$ production systematically using the global dataset of SCIAMACHY NO$_2$ observations combined with flash data from the World Wide Lightning Location Network (WWLLN). Their analysis was restricted to $30 \times 60$ km$^2$ satellite pixels where the flash rate exceeded 1 flash km$^{-2}$ hr$^{-1}$. But they found LNO$_x$ production to be highly variable and correlations between flash rate densities and LNO$_x$ production were low in some cases. Bucsela et al. (2010) estimated LNO$_x$ production as $\sim 100 - 250$ mol NO$_x$/flash for four cases, using the DC-8 and OMI data during NASA's Tropical Composition, Cloud and Climate Coupling Experiment (TC$^4$).

Based on the approach used by Bucsela et al. (2010), a special algorithm was developed by Pickering et al. (2016) to retrieve LNO$_x$ from OMI and the WWLLN. The algorithm takes the OMI tropospheric slant column density (SCD) of NO$_2$ (S$_{NO_2}$) as the tropospheric slant column density of LNO$_2$ (S$_{LNO_2}$) by using cloud radiance fraction (CRF) greater than 0.9 to minimize or screen the lower tropospheric background. To convert the S$_{LNO_2}$ to the tropospheric vertical column density (VCD) of LNO$_x$ (V$_{LNO_x}$), an air mass factor (AMF) is calculated by dividing the a priori S$_{LNO_2}$ by the a priori V$_{LNO_x}$. The a priori S$_{LNO_2}$ is calculated using a radiative transfer model and a profile of LNO$_2$ simulated by the NASA Global Modeling Initiative (GMI) chemical transport model. The a priori V$_{LNO_x}$ is also obtained from the GMI model. Results for the Gulf of Mexico during 2007 – 2011 summer yield LNO$_x$ production of $80 \pm 45$ mol NO$_x$ per flash. Since they considered NO$_2$ above the cloud as LNO$_2$ in the algorithm due to the difficulty and uncertainty in determining the background NO$_2$, their AMF and derived VCD of LNO$_x$ (LNO$_2$) is named as AMF$_{LNO_x Clean}$ (AMF$_{LNO2Clean}$) and LNO$_x$Clean (LNO$_2$Clean), respectively. Note that Pickering et al. (2016) considered the two estimates of background derived from aircraft flights in the Gulf of Mexico region (3% and 33%) and subtracted the mean value (18%) from the estimated mean LNO$_x$ production efficiency (PE) for the background bias. However, we use the original algorithm directly without correction to distinguish the effect of different AMFs on LNO$_x$ estimation in the remainder of this paper. Unless otherwise specified, abbreviations S and V are respectively defined as the tropospheric SCD and VCD in this paper.

More recently Bucsela et al. (2019) obtained an average PE of $180 \pm 100$ mol NO$_x$/flash over East Asia, Europe and North America based on a modification of the method used in Pickering et al. (2016). A power function between LNO$_x$ and lightning flash rate was established, while the minimum flash-rate threshold was not applied. The tropospheric NO$_x$ background was removed by subtracting the temporal average of NO$_x$ at each box where the value was weighted by the number of OMI pixels which meet the optical cloud pressure and CRF criteria required to be considered deep convection but have 1 flash or less instead. The lofted pollution was considered as 15% of total NO$_x$ according to the estimation from DeCaria et al. (2000, 2005) and the average chemical delay was adjusted by 15% following the 3-hour LNO$_x$ lifetime in the nearby field of convection (Nault et al., 2017). However, there were negative LNO$_x$ values caused by the overestimation of the tropospheric background and stratospheric NO$_2$ at some locations.

On the other hand, Lapierre et al. (2020) constrained LNO$_2$ to $1.1 \pm 0.2$ mol NO$_2$/stroke for intracloud (IC) strokes and $10.7 \pm 2.5$ mol NO$_2$/stroke for cloud-to-ground (CG) strokes over the CONUS. LNO$_2$ per stroke was scaled to 24.2 mol NO$_x$/flash using mean values of strokes per flash and the ratio of NO$_x$ to NO$_2$ in the UT. They used the regridded Berkeley High-Resolution (BEHR) V3.0A $0.05° \times 0.05°$ "visible only" NO$_2$ VCD (V$_{vis}$) product which includes two parts of NO$_2$ that

can be "seen" by the satellite. The first part is the $NO_2$ above clouds (pixels with CRF > 0.9) and the second part is the $NO_2$ detected from cloud free areas. A threshold of $3 \times 10^{15}$ molecules $cm^{-2}$, the typical urban $NO_2$ concentration, was applied to mask the contaminated grid cells (Beirle et al., 2010; Laughner and Cohen, 2017). The main difference between Lapierre et al. (2020) and Pickering et al. (2016) is the air mass factor for lightning ($AMF_{LNO_x}$) implemented in the basic algorithm. In Lapierre et al. (2020), the air mass factor was used to convert $S_{NO_2}$ to $V_{vis}$, while in Pickering et al. (2016) it was used to convert $S_{LNO_2}$ to $V_{LNO_x}$, assuming that all $S_{NO_2}$ is generated by lightning.

To apply the approach used by Bucsela et al. (2010), Pickering et al. (2016), Bucsela et al. (2019) and Lapierre et al. (2020) without geographic restrictions, the contamination by anthropogenic emissions must be taken into account in detail. The Weather Research and Forecasting (WRF) model coupled with chemistry (WRF-Chem) has been employed to evaluate the convective transport and chemistry in many studies (Barth et al., 2012; Wong et al., 2013; Fried et al., 2016; Li et al., 2017). Meanwhile, Laughner and Cohen (2017) showed that the OMI AMF is increased by $\sim$35% for summertime when $LNO_2$ simulated by WRF-Chem is included in the a priori profiles to match aircraft observations. The simulation agrees with observed $NO_2$ profiles and the bias of AMF related to these observations is reduced to $< \pm 4\%$ for OMI viewing geometries.

In this paper, we focus on the estimation of $LNO_2$ production per flash ($LNO_2$/flash), $LNO_x$ production per flash ($LNO_x$/flash), $LNO_2$ production per stroke ($LNO_2$/stroke) and $LNO_x$ production per stroke ($LNO_x$/stroke) in May–August (MJJA) 2014 by developing an algorithm similar to Pickering et al. (2016) based on the BEHR $NO_2$ retrieval algorithm (Laughner et al., 2018a, b), but it performs better over background $NO_2$ sources. Section 2 describes the satellite data, lightning data, model settings and the algorithm in detail. Section 3 explores the suitable data criteria, compares different methods and evaluates the effect of background $NO_2$, cloud and $LNO_x$ parameterization on $LNO_x$ production estimation. Section 4 examines the effect of different sources of the uncertainty on the results. Conclusions are summarized in Section 5.

## 2 Data and Methods

### 2.1 Ozone Monitoring Instrument (OMI)

OMI is carried on the Aura satellite (launched in 2004), a member of A-train satellite group (Levelt et al., 2006, 2018). OMI passes over the equator at $\sim$ 13:45 LT (ascending node) and has a swath width of 2600 km, with a nadir field-of-view resolution of $13 \times 24$ $km^2$. Since the beginning of 2007, some of the measurements have become useless as a result of anomalous radiances called the "row anomaly" (Dobber et al., 2008; KNMI, 2012). For the current study, we used the NASA standard product V3.0 (Krotkov et al., 2017) as input to the $LNO_x$ retrieval algorithm.

The main steps of calculating the $NO_2$ tropospheric VCD ($V_{NO_2}$) in the NASA product include:

1. SCDs are determined by the OMI-optimized differential optical absorption spectroscopy (DOAS) spectral fit;

2. A corrected ("de-striped") SCD is obtained by subtracting the cross-track bias caused by an instrument artifact from the measured slant column;

3. The AMF for stratospheric ($AMF_{strat}$) or tropospheric column ($AMF_{trop}$) is calculated from the $NO_2$ profiles integrated vertically using weighted scattering weights with the a priori profiles. These profiles are obtained from GMI monthly mean profiles using four years (2004 – 2007) simulation;

4. The stratospheric $NO_2$ VCD ($V_{strat}$) is calculated from the subtraction of a priori contribution from tropospheric $NO_2$ and a three-step (interpolation, filtering, and smoothing) algorithm (Bucsela et al., 2013);

5. $V_{strat}$ is converted to the slant column using $AMF_{strat}$ and subtracted from the measured SCDs to yield $S_{NO_2}$, leading to $V_{NO_2} = S_{NO_2}/AMF_{trop}$.

Based on this method, we developed a new $AMF_{LNO_x}$ to obtain the desired $V_{LNO_x}$ ($V_{LNO_x} = S_{NO_2}/AMF_{LNO_x}$) by replacing the original step 5. Details of this algorithm are discussed in section 2.4.

## 2.2 The Earth Networks Total Lightning Detection Network (ENTLN)

The Earth Networks Total Lightning Network (ENTLN) operates a system of over 1500 ground-based stations around the world with more than 900 sensors installed in the CONUS (Zhu et al., 2017). Both IC and CG lightning flashes are located by the sensors with detection frequency ranging from 1 Hz to 12 MHz based on the electric field pulse polarity and wave shapes. Groups of pulses are classified as a flash if they are within 700 ms and 10 km. In the preprocessed data obtained from the ENTLN, both strokes and lightning flashes composed of one or more strokes are included.

Rudlosky (2015) compared ENTLN combined events (IC and CG) with LIS flashes and found that the relative flash detection efficiency of ENTLN over CONUS increases from 62.4% during 2011 to 79.7% during 2013. Lapierre et al. (2020) also compared combined ENTLN and the NLDN dataset with data from the LIS during 2014 and found the detection efficiencies of IC flashes and strokes to be 88% and 45%, respectively. Since we only use the ENTLN data in 2014 as Lapierre et al. (2020) and NLDN detection efficiency of IC pulses should be lower than 33% which is calculated by the data in 2016 (Zhu et al., 2016), only the IC flashes and strokes are divided by 0.88 and 0.45, respectively, while CG flashes and strokes are unchanged because of the high detection efficiency.

## 2.3 Model Description

The present study uses WRF-Chem version 3.5.1 (Grell et al., 2005) with a horizontal grid size of $12 \times 12$ km$^2$ and 29 vertical levels (Fig. 1). The initial and boundary conditions of meteorological parameters are provided by the North American Regional Reanalysis (NARR) dataset with a 3 hourly time resolution. Based on Laughner et al. (2018b), 3D wind fields, temperature and water vapor are nudged towards the NARR data. Outputs from the version 4 of Model for Ozone and Related chemical Tracers (MOZART-4; Emmons et al., 2010) are used to generate the initial and boundary conditions of chemical species. Anthropogenic emissions are driven by the 2011 National Emissions Inventory (NEI), scaled to model years by the Environmental Protection Agency annual total emissions (EPA and OAR, 2015). The Model of Emissions of Gases and Aerosol from Nature (MEGAN; Guenther et al., 2006) is used for biogenic emissions. The chemical mechanism is the version 2 of Regional Atmospheric Chemistry Mechanism (RACM2; Goliff et al., 2013) with updates from Browne et al. (2014) and Schwantes et al. (2015). In addition, lightning flash rate based on the level of neutral buoyancy parameterization (Price and Rind, 1992; Wong et al., 2013)

and LNO$_x$ parameterizations are activated (200 mol NO flash$^{-1}$, the factor to adjust the predicted number of flashes is set to 1; hereinafter referred to as "1×200 mol NO flash$^{-1}$"). Simulated total flash densities are higher than ENTLN observations over the Southeast US and lower than observations in the North Central US (Fig. 2). The impact of these biases on LNO$_x$ production is discussed and mitigated in Sect. 3.1 and 3.4. The bimodal profile modified from the standard Ott et al. (2010) profile (Laughner and Cohen, 2017) is employed as the vertical distribution of lightning NO (LNO) in WRF-Chem, while

outputs of LNO and LNO$_2$ profiles are defined as the difference of vertical profiles between simulations with and without lightning.

## 2.4 Method for Deriving AMF

The V$_{\text{LNO}_x}$ near convection is calculated according:

$$V_{\text{LNO}_x} = \frac{S_{\text{NO}_2}}{AMF_{\text{LNO}_x}} \tag{1}$$

where $S_{\text{NO}_2}$ is the OMI-measured tropospheric slant column NO$_2$ and AMF$_{\text{LNO}_x}$ is a customized lightning air mass factor. The concept of AMF$_{\text{LNO}_x}$ was also used in Beirle et al. (2009) to investigate the sensitivity of satellite instruments to freshly produced lightning NO$_x$. In order to estimate LNO$_x$, we define the AMF$_{\text{LNO}_x}$ as the ratio of the "visible" modeled NO$_2$ slant column to the total modeled tropospheric LNO$_x$ vertical column (derived from the a priori NO and NO$_2$ profiles, scattering weights, and cloud radiance fraction):

$$AMF_{\text{LNO}_x} = \frac{(1 - f_r) \int_{p_{surf}}^{p_{tp}} w_{clear}(p) NO_2(p)\, dp + f_r \int_{p_{cloud}}^{p_{tp}} w_{cloudy}(p) NO_2(p)\, dp}{\int_{p_{surf}}^{p_{tp}} LNO_x(p)\, dp} \tag{2}$$

where $f_r$ is the cloud radiance fraction (CRF), $p_{surf}$ is the surface pressure, $p_{tp}$ is the tropopause pressure, $p_{cloud}$ is the cloud optical pressure (CP), $w_{clear}$ and $w_{cloudy}$ are respectively the pressure dependent scattering weights from the TOMRAD lookup table (Bucsela et al., 2013) for clear and cloudy parts, and $NO_2(p)$ is the modeled NO$_2$ vertical profile. Details of these standard parameters and calculation methods are given in Laughner et al. (2018a). $LNO_x(p)$ is the LNO$_x$ vertical profile calculated by

the difference of vertical profiles between WRF-Chem simulations with and without lightning.

Please note that the CP is a reflectance-weighted pressure retrieved by the collision-induced O$_2$-O$_2$ absorption band near 477 nm (Acarreta et al., 2004; Sneep et al., 2008; Stammes et al., 2008). For a deep convective cloud with lightning, the CP lies below the geometrical cloud top which is much closer to that detected by thermal infrared sensors, such as the CloudSat and the Aqua MODerate-resolution Imaging Spectrometer (MODIS) (Vasilkov et al., 2008; Joiner et al., 2012). Hence, much of

the tropospheric NO$_2$ measured by OMI lies inside the cloud rather than above the cloud top. In the following, "above cloud" or "below cloud" is relative to the cloud pressure detected by OMI. The sensitivity study of Beirle et al. (2009) compared the chemical composition from the cloud bottom to the cloud top and revealed that a significant fraction of the NO$_2$ within the cloud originating from lightning can be detected by the satellite. This valuable cloud pressure concept has been applied not only in the LNO$_x$ research but also in the cloud slicing method of deriving the UT O$_3$ and NO$_x$ (Ziemke et al., 2009; Choi et al.,

2014; Strode et al., 2017; Ziemke et al., 2017; Marais et al., 2018). As discussed in Pickering et al. (2016), the ratio of V$_{\text{LNO}_2}$ seen by OMI to V$_{\text{LNO}_x}$ is partly influenced by $p_{cloud}$. The effects of LNO$_2$ below the cloud will be discussed in Sect. 3.4.

To compare our results with those of Pickering et al. (2016) and Lapierre et al. (2020), we calculate their $AMF_{LNO_x Clean}$ and $AMF_{NO_2 Vis}$ respectively:

$$AMF_{LNO_x Clean} = \frac{(1 - f_r) \int_{p_{surf}}^{p_{tp}} w_{clear}(p) LNO_2(p) \, dp + f_r \int_{p_{cloud}}^{p_{tp}} w_{cloudy}(p) LNO_2(p) \, dp}{\int_{p_{surf}}^{p_{tp}} LNO_x(p) \, dp} \tag{3}$$


$$AMF_{NO_2 Vis} = \frac{(1 - f_r) \int_{p_{surf}}^{p_{tp}} w_{clear}(p) NO_2(p) \, dp + f_r \int_{p_{cloud}}^{p_{tp}} w_{cloudy}(p) NO_2(p) \, dp}{(1 - f_g) \int_{p_{surf}}^{p_{tp}} NO_2(p) \, dp + f_g \int_{p_{cloud}}^{p_{tp}} NO_2(p) \, dp} \tag{4}$$

where $f_g$ is the geometric cloud fraction and $LNO_2(p)$ is the modeled $LNO_2$ vertical profile. Besides these AMFs, another AMF called $AMF_{LNO_2 Vis}$ is developed for later comparison.

$$AMF_{LNO_2 Vis} = \frac{(1 - f_r) \int_{p_{surf}}^{p_{tp}} w_{clear}(p) NO_2(p) \, dp + f_r \int_{p_{cloud}}^{p_{tp}} w_{cloudy}(p) NO_2(p) \, dp}{(1 - f_g) \int_{p_{surf}}^{p_{tp}} LNO_2(p) \, dp + f_g \int_{p_{cloud}}^{p_{tp}} LNO_2(p) \, dp} \tag{5}$$

A full definition list of the used AMFs is shown in Appendix A.

## 2.5   Procedures for Deriving LNO$_x$

$V_{LNO_x}$ is re-gridded to $0.05° \times 0.05°$ grids using the constant value method (Kuhlmann et al., 2014). Then, it is analyzed in $1° \times 1°$ grid boxes with a minimum of fifty valid $0.05° \times 0.05°$ grids to minimize the noise. The main procedures of deriving LNO$_x$ are as follows:

CRFs (CRF $\geq$ 70%, CRF $\geq$ 90% and CRF = 100%) and CP $\leq$ 650 hPa are various criteria of deep convective clouds for OMI pixels (Ziemke et al., 2009; Choi et al., 2014; Pickering et al., 2016). The effect of different CRFs on the retrieved LNO$_x$ is explored in section 3.2. Furthermore, another criterion of cloud fraction (CF) is applied to the WRF-Chem results for the successful simulation of convection. The CF is defined as the maximum cloud fraction calculated by the Xu-Randall method between 350 and 400 hPa (Xu and Randall, 1996; Strode et al., 2017). This atmospheric layer (between 350 and 400 hPa)

avoids any biases in the simulation of high clouds. We choose CF $\geq$ 40% suggested by Strode et al. (2017) to determine cloudy or clear for each simulation grid.

Besides cloud properties, a time period and sufficient flashes (or strokes) are required for fresh LNO$_x$ to be detected by OMI. The time window ($t_{window}$) is the hours prior to the OMI overpass time. $t_{window}$ is limited to 2.4 h by the mean wind speed at pressure levels 500 – 100 hPa during OMI overpass time and the square root of the $1° \times 1°$ box over the CONUS (Lapierre

et al., 2020). Meanwhile, 2400 flashes box$^{-1}$ and 8160 strokes box$^{-1}$ per 2.4 hour time window are chosen as sufficient for detecting LNO$_x$ (Lapierre et al., 2020). These criteria will result in a low bias in the PE results, as Bucsela et al. (2019) found that the PE is larger at small flash rates which are discarded here. Since our study focuses on developing a new AMF and compare results with other works using the similar lightning thresholds (Lapierre et al., 2020; Pickering et al., 2016), we will only discuss results based on the strict criteria in the main text. For comparisons between 2400 flashes box$^{-1}$ criterion and 1

flash box$^{-1}$ criterion, scatter diagrams using different lightning criteria are presented in Appendix B.

To ensure that lightning flashes are simulated successfully by WRF-Chem, the threshold of simulated total lightning flashes (TL) per box is set to 1000, which is fewer than that used by the ENTLN lightning observation, considering the uncertainty of lightning parameterization. In view of other $NO_2$ sources in addition to $LNO_2$, the ratio of modeled lightning $NO_2$ above cloud ($LNO_2Vis$) to modeled $NO_2$ above cloud ($NO_2Vis$) is defined to check whether enough $LNO_2$ can be detected by OMI. The ratio $\geq 50\%$ indicates that more than half of the $NO_x$ above the cloud must has an $LNO_x$ source.

Finally, the $NO_2$ lifetime against oxidation should be taken into account. As estimated by Nault et al. (2016), the lifetime ($\tau$) of $NO_2$ in the near field of convections is $\sim 3$ h. The initial value of $NO_2$ is solved by Eq. 6 as

$$NO_2(0) = NO_2(OMI) \times e^{0.5t/\tau} \tag{6}$$

where $NO_2(0)$ is the moles of $NO_2$ emitted at time $t = 0$, $NO_2(OMI)$ is the moles of $NO_2$ measured at the OMI overpass time and $0.5t$ is the half cross grid time which is 1.2 h, assuming that lightning occurred at the center of each $1° \times 1°$ box. For each grid box, the mean $LNO_x$ vertical column is obtained by averaging $V_{LNO_x}$ values from all regridded $0.05° \times 0.05°$ pixels in the box. This mean value is converted to moles $LNO_x$ using the dimensions of the grid box. Two methods are applied to estimate the seasonal mean $LNO_2$/flash, $LNO_x$/flash, $LNO_2$/stroke and $LNO_x$/stroke:

(1) summation method: dividing the sum of $LNO_x$ by the sum of flashes (or strokes) in each $1° \times 1°$ box in MJJA 2014;

(2) linear regression method: applying the linear regression to daily mean values of $LNO_x$ and flashes (or strokes).

## 3   Results

### 3.1   Criteria Determination

To determine the suitable criteria from conditions defined in section 2.5, six different combinations are defined (Table 1) and applied to the original data with a linear regression method (Table 2).

A daily search of the $NO_2$ product for coincident ENTLN flash (stroke) data results in 99 (102) valid days under the CRF90_ENTLN condition. Taking the flash type ENTLN data as an example, the number of valid days decreases from 99 to 81 under the CRF90_ENTLN_TL1000_ratio50 condition, while $LNO_x$/flash increases from $52.1 \pm 51.1$ mol/flash to $54.5 \pm 48.1$ mol/flash. The result is almost the same as that under the CRF90_ENTLN_TL1000 condition which is without the condition of more than half of the above-cloud $NO_x$ having an $LNO_x$ source. Although this indicates the criterion of TL works well, it is better to include the ratio criterion in case of some exceptions in the different AMF methods. Since CF $\geq 40\%$ leads to a sharp loss of valid numbers and production, therefore, it is not a suitable criterion. Instead the CRF criteria are used. Finally, coincident ENTLN data, TL $\geq 1000$ and ratio $\geq 50\%$ are chosen as the thresholds to explore the effects of three different CRF conditions (CRF $\geq 70\%$, CRF $\geq 90\%$ and CRF $= 100\%$) on $LNO_x$ production (Table 3). Apart from the fewer valid days under higher CRF conditions (CRF $\geq 90\%$ and CRF $= 100\%$), $LNO_x$/flash increases from $35.7 \pm 36.8$ mol/flash to $54.5 \pm 48.1$ mol/flash and decreases again to $20.8 \pm 37.4$ mol/flash while $LNO_x$/stroke enhances from $4.1 \pm 3.9$ mol/stroke to $7.0 \pm 4.8$ mol/stroke and drops again to $2.6 \pm 4.0$ mol/stroke (Table 3), as the CRF criterion increases from 70% to 90% and to 100%. When the CRF increases from 90% to 100%, the $LNO_x$ PE decreases because of the higher lightning density with

fewer $LNO_x$ (not shown). The increment of $LNO_x$ PE caused by the CRF increase from 70% to 90% is opposite to the result of Pickering et al. (2016). This is an effect of the consideration of $NO_2$ contamination transported from the boundary layer in our method. Although enhanced $NO_x$ is often observed in regions with CRF > 70% (Pickering et al., 2016), the following analysis will be based on the criterion of CRF $\geq$ 90% considering the contamination by low and mid-level $NO_2$ and comparisons with the results of Pickering et al. (2016) and Lapierre et al. (2020).

### 3.2 Comparison of $LNO_x$ Production based on Different AMFs

Lapierre et al. (2020) derived $LNO_2$ production based on the BEHR $NO_2$ product. In order for our results to be comparable with those of Pickering et al. (2016) and Lapierre et al. (2020), we choose $NO_2$ instead of $NO_x$ to derive production per flash (production efficiency, PE). In Fig. 3, time series of $NO_2$Vis, $LNO_2$Vis, $LNO_2$ and $LNO_2$Clean production per day over CONUS are plotted for MJJA 2014 with the criterion of CRF $\geq$ 90% and a flash threshold of 2400 flashes per 2.4 h. $LNO_2$ PEs are mostly in the range from 20 to 80 mol/flash. $LNO_2$Vis PEs are smaller than $LNO_2$ PEs which contain $LNO_2$ below clouds. The simulation of GMI in Pickering et al. (2016) indicated that 25% – 30% of the $LNO_x$ column lies below the CP, while the ratio in our WRF-Chem simulation is 56 $\pm$ 20%. The effect of cloud properties on $LNO_x$ PE will be discussed in more detail in section 3.4. Generally, the order of estimated daily PEs is $LNO_2$Clean > $LNO_2$ > $NO_2$Vis > $LNO_2$Vis. The percent difference in the estimated PE ($\Delta$PE) between $NO_2$Vis and $LNO_2$Vis indicates a certain amount of background $NO_2$ exists above clouds. Overall, the tendency of that $\Delta$PE is consistent with another $\Delta$PE between $NO_2$Vis and $LNO_2$Clean. When the region is highly polluted ($\Delta$PE between $NO_2$Vis and $LNO_2$Vis is larger than 200%), PEs based on $NO_2$Vis and $LNO_2$Clean are significantly overestimated. In other words, $NO_2$Vis and $LNO_2$Clean are more sensitive to background $NO_2$. The extent of the overestimation of $NO_2$Vis is larger than that of $LNO_2$Clean in highly polluted regions, while it is usually opposite in most regions.

Figure 4 shows the linear regression for ENTLN data versus $NO_2$Vis, $LNO_2$Vis, $LNO_2$ and $LNO_2$Clean with the same criteria as shown in Fig. 3. $LNO_2$Clean PE (the largest slope) is 25.2 $\pm$ 22.3 mol $NO_2$/flash with a correlation of 0.25 and 2.3 $\pm$ 2.1 mol $NO_2$/stroke with a correlation of 0.22. As shown in Fig. 3, positive percent differences between $NO_2$Vis PE and $LNO_2$Clean PE occur much fewer than negative differences. As a result, $NO_2$Vis PE (17.1 $\pm$ 17.2 mol $NO_2$/flash and 0.4 $\pm$ 1.0 mol $NO_2$/stroke) is smaller than $LNO_2$Clean PE using the linear regression method.

In order to compare our result with that of Lapierre et al. (2020), we tried to remove the CP $\leq$ 650 hPa, TL $\geq$ 1000 and ratio $\geq$ 50% conditions from criteria. But, our result based on daily summed $NO_2$Vis values (3.8 $\pm$ 0.5 mol/stroke) is still larger than the value of 1.6 $\pm$ 0.1 mol/stroke mentioned in Lapierre et al. (2020). This may be caused by the different version of BEHR algorithm, as Lapierre et al. (2020) used BEHR V3.0A and our algorithm is based on BEHR V3.0B (Laughner et al., 2019). The input of $S_{NO_2}$ in both versions is from the NASA standard product V3 and the major improvements of BEHR V3.0B are listed below:

1. The profile (V3.0B) closest to the OMI overpass time was selected instead of the last profile (V3.0A) before the OMI overpass.

2. The AMF uses a variable tropopause height as opposed to the fixed 200 hPa tropopause.

3. The surface pressure is now calculated according to Zhou et al. (2009).

The detailed log of changes is available at https://github.com/CohenBerkeleyLab/BEHR-core (last access: March 8, 2020). Note that Lapierre et al. (2020) used the monthly $NO_2$ profile, while the daily profile is used in our study and the interval of our outputs from WRF-Chem is 30 min which is more frequent than 1 h in the BEHR daily product, the AMF could be affected by different $NO_2$ profiles. In view of these factors, we compare different methods based on our data to minimize these effects.

Meanwhile, $LNO_2$ PE (18.7 ± 18.1 mol/flash and 2.1 ± 1.8 mol/stroke) is between $LNO_2$Clean PE and $NO_2$Vis PE, which coincides with the daily results in Fig. 3. Furthermore, the $LNO_x$ PE based on the linear regression of daily summed values, the same method used in Pickering et al. (2016), is 114.8 ± 18.2 mol/flash (or 17.8 ± 2.9 mol/stroke) which is larger than 91 mol/flash in Pickering et al. (2016), possibly due to the differences in geographic location, lightning data and chemistry model.

The mean and standard deviation of $LNO_2$ PE under CRF ≥ 90% using the summation method is 46.2 ± 35.1 mol/flash and 9.9 ± 8.1 mol/stroke, while $LNO_x$ PE is 125.6 ± 95.9 mol/flash and 26.7 ± 21.6 mol/stroke (Fig. 5). The $LNO_2$ PE and $LNO_x$ PE are both higher in the Southeast U.S. (denoted by the red box in Fig. 5 panels, 75°W – 95°W, 25°N – 37°N), consistent with Lapierre et al. (2020) and Bucsela et al. (2019). Compared with Fig. 3, Figure 6a and b present some large differences between $NO_2$Vis PE and $LNO_2$Vis PE, which are consistent with what we expect for polluted regions. Meanwhile, the differences between $LNO_2$ PE and $NO_2$Vis PE depend on background $NO_2$, the strength of updraft and the profile. The negative differences are caused by background $NO_2$ carried by the updraft while parts of the below-cloud $LNO_2$ result in $LNO_2$ PE higher than $NO_2$Vis PE (Fig. 6c). Figure 6d shows that the ratio of $LNO_2$Vis to $LNO_2$ ranges from 10% – 80%. This may be caused by the height of the clouds and the profile of $LNO_2$. If the CP is near 300 hPa, the ratio should be smaller because of the coverage of clouds. While peaks of the $LNO_2$ profile are below the CP, the ratio would also be smaller. Therefore, a better understanding of $LNO_2$ profile and $LNO_x$ below clouds is required.

### 3.3 Effects of Tropospheric Background on $LNO_x$ Production

With respect to the $LNO_2$ production, the patterns in Fig. 6 indicate the improvement of our approach is different in polluted and clean regions. To simplify the quantification, we select six grids with similar $NO_2$ profile (∼ 100 pptv) above the cloud with CRF = 100%. These grid boxes contain the polluted and clean cities denoted by stars and triangles in Fig. 6a, respectively. Then, the differences between AMFs are dependent on fewer parameters:

$$AMF_{\text{LNO}_2} = \frac{\int_{p_{cloud}}^{p_{tp}} w_{cloudy}(p) NO_2(p)\,dp}{\int_{p_{surf}}^{p_{tp}} LNO_2(p)\,dp} \tag{7}$$

$$AMF_{\text{NO}_2\text{Vis}} = \frac{\int_{p_{cloud}}^{p_{tp}} w_{cloudy}(p) NO_2(p)\,dp}{\int_{p_{cld}}^{p_{tp}} NO_2(p)\,dp} \tag{8}$$

$$AMF_{\text{LNO}_2\text{Clean}} = \frac{\int_{p_{cloud}}^{p_{tp}} w_{cloudy}(p) LNO_2(p)\,dp}{\int_{p_{surf}}^{p_{tp}} LNO_2(p)\,dp} \tag{9}$$

Figure 7 compares the mean profiles of $NO_2$, background $NO_2$ and background $NO_2$ ratio in polluted and clean grids. Generally, the profiles of the ratio of background $NO_2$ over total $NO_2$ are C-shape because UT $LNO_2$ concentrations are higher than UT background $NO_2$ concentrations. However, the ratio profile in Fig. 7e has one peak between the cloud pressure and tropopause as background $NO_2$ increases and $LNO_2$ decreases. Besides, the percentage of UT background $NO_2$ in polluted regions is steady and higher than that in clean regions.

Table 4 presents the relative changes among three methods in six cities. The difference between $AMF_{LNO_2}$ (Eq. 7) and $AMF_{LNO_2Clean}$ (Eq. 9) is the numerator: $\int_{p_{cloud}}^{p_{tp}} w_{cloudy}(p) NO_2(p)\, dp$ and $\int_{p_{cloud}}^{p_{tp}} w_{cloudy}(p) LNO_2(p)\, dp$. When the ratio of $LNO_2$ is higher or the region is cleaner, the relative difference is smaller (e.g. 5.0% – 12.0%, Fig. 7d – f). The largest relative difference (46.3%) occurs when the ratio of background $NO_2$ is continuously high in the UT (Fig. 7c). As a result, our approach is less sensitive to background $NO_2$ and more suitable for convective cases over polluted locations. In contrast, production esti-

mated by our method is larger than that based on $NO_2Vis$ due to the $LNO_2$ below the cloud. When the cloud is higher, especially the peak of LNO profile is lower than the cloud (Fig. 7b), the relative difference is larger (121.2%) because more $LNO_2$ can not be included into the $NO_2Vis$, which has been discussed in Sect. 3.2. The relative change between $AMF_{LNO_2Clean}$ (Eq. 9) and $AMF_{NO_2Vis}$ (Eq. 8) depends on $\int_{p_{cloud}}^{p_{tp}} w_{cloudy}(p) LNO_2(p)\, dp / \int_{p_{surf}}^{p_{tp}} w_{cloudy}(p) LNO_2(p)\, dp$, which is also affected by cloud not the background $NO_2$. The largest relative change (153.8%) occurs at New Orleans, which has the lowest cloud pressure

and consequently the smallest visible column.

### 3.4   Effects of Cloud and $LNO_x$ Parameterization on $LNO_x$ Production

Figure 8a presents the daily distribution of CP and the ratio of $LNO_2Vis$ to $LNO_2$ during MJJA 2014 with the criteria defined in section 3.1 under $CRF \geq 90\%$. Since the ratio of $LNO_2Vis$ to $LNO_2$ decreases from 0.8 to 0.2 as the cloud pressure decreases from 600 to 300 hPa, $NO_2Vis$ PE is smaller than $LNO_2$ PE in relatively clean areas as shown in Fig. 4. Apart from $LNO_2Vis$,

the $LNO_2$ PE is also affected by CP. For $LNO_2$ PEs larger than 30 mol/stroke, the CPs are all smaller than 550 hPa (Fig. 8b). However, smaller $LNO_2$ PEs (< 30 mol/stroke) occur on all levels between 650 hPa and 200 hPa. Because of the limited amount of large $LNO_2$ PEs and lightning data, we cannot derive the relationship between $LNO_2$ PE and cloud pressure or other lightning properties at this stage. Because the CP only represents the development of clouds, the vertical structure of flashes can not be derived from the CP values only. As discussed in several previous studies, the flash channel length varies and

depends on the environmental conditions (Carey et al., 2016; Mecikalski and Carey, 2017; Fuchs and Rutledge, 2018). Davis et al. (2019) compared two kinds of flash: normal flashes and anomalous flashes. Because updrafts are stronger and flash rates are higher in anomalous storms, UT $LNO_x$ concentrations are larger in anomalous than normal polarity storms. In general, normal flashes are coupled with an upper-level positive charge region and a mid-level negative charge region, while anomalous flashes are opposite (Williams, 1989). It is not straightforward to estimate the error resulting from the vertical distribution of

$LNO_x$. There are mainly two methods of distributing $LNO_x$ in models: $LNO_x$ profiles (postconvection) in which $LNO_x$ has already been redistributed by convective transport, while the other one (preconvection) uses $LNO_x$ production profiles made before the redistribution of convective transport (Allen et al., 2012; Luo et al., 2017). However, given the similarity of results

compared to other LNO$_x$ studies, we believe that our $1° \times 1°$ results based on postconvective LNO$_x$ profile are sufficient for estimating average LNO$_x$ production.

The LNO production settings in WRF-Chem varied in different studies. Zhao et al. (2009) set a NO$_x$ production rate of 250 mol NO per flash in a regional-scale model, while Bela et al. (2016) chose 330 mol NO per flash used by Barth et al. (2012). Wang et al. (2015) assumed approximately 500 mol NO per flash which was derived by a cloud-scale chemical transport model and in-cloud aircraft observations (Ott et al., 2010). To illustrate the impact of LNO$_x$ parameterization on LNO$_x$ estimation, we apply another WRF-Chem NO$_2$ profile setting (2×base flashrate, 500 mol NO flash$^{-1}$; hereinafter referred to as "2×500

mol NO flash$^{-1}$") to a priori profiles and evaluate the changes in AMF$_{LNO_2}$, AMF$_{LNO_x}$, LNO$_2$ PE and LNO$_x$ PE. For the linear regression method (Fig. 9), LNO$_2$ PE is $29.8 \pm 20.5$ mol/flash which is 59.4% larger than the basic one ($18.7 \pm 18.1$ mol/flash). Meanwhile, LNO$_x$ PE (increasing from $54.5 \pm 48.1$ mol/flash to $88.5 \pm 61.1$ mol/flash) also depends on the configuration of LNO production in WRF-Chem. The comparison between Fig. 4 and Fig. 9 shows that LNO$_2$Clean PE and LNO$_2$ PE are more similar while LNO$_2$ PE and NO$_2$Vis PE present the same tendency. It remains unclear as to whether the NO-NO$_2$-O$_3$ cycle or

other LNO$_x$ reservoirs accounts for the increment of LNO$_x$ PE. This would need detailed source analysis in WRF-Chem and is beyond the scope of this study.

Figure 10 shows the average percentage changes in AMF$_{LNO_2}$, AMF$_{LNO_x}$, LNO$_2$ and LNO$_x$ between retrievals using profiles based on 1×200 mol NO flash$^{-1}$ and 2×500 mol NO flash$^{-1}$. These results were obtained by averaging data over MJJA 2014 based on the method described in Sect. 2.5 with the criterion of CRF $\geq$ 90%. The effects on LNO$_2$ and LNO$_x$ retrieval from

increasing LNO profile values show mostly the same tendency: smaller AMF$_{LNO_2}$ and AMF$_{LNO_x}$ leads to larger LNO$_2$ and LNO$_x$, but the changes are regionally dependent. This is caused by the nonlinear calculation of AMF$_{LNO_2}$ and AMF$_{LNO_x}$. As the contribution of LNO$_2$ increases, both the numerator and denominator of Eq. (2) increase. Note that the LNO$_2$ accounts for a fraction of NO$_2$ above the clouds, the magnitude of increasing denominator could be different than that of increasing numerator, resulting in a different effect on the AMF$_{LNO_2}$ and AMF$_{LNO_x}$. As mentioned in Zhu et al. (2019), the lightning densities in the

Southeast U.S. might be overestimated using the 2×500 mol NO flash$^{-1}$ setting and the same lightning parameterization as ours. Fortunately, the AMFs and estimated LNO$_2$ change little in that region. Because the Southeast U.S. has the highest flash density (Fig. 2), the NO$_2$ in the numerator of AMF is dominated by LNO$_2$. Both the SCD and VCD will increase when the model uses higher LNO$_2$. In other words, the sensitivity to the LNO setting decreases and the relative distribution of LNO$_2$ matters.

Figure 11 shows the comparison of the mean LNO and LNO$_2$ profiles in two specific regions where the 2×500 mol NO flash$^{-1}$ setting leads to lower and higher LNO$_2$ PEs, respectively. The first one (Fig. 11a) is the region (36°N – 37°N, 89°W – 90°W) containing the minimal negative percent change in LNO$_2$ (Fig. 10c). The second one (31°N – 32°N, 97°W – 98°W), Figure 11b, has the largest positive percent change in LNO$_2$ (Fig. 10c). Although the relative distributions of mean LNO and LNO$_2$ profiles are similar in both regions, the magnitude differs with a factor of 10. This phenomenon implies that the

performance of lightning parameterization in WRF-Chem is regionally dependent and an unrealistic profile could appear in the UT. Although this sensitivity analysis is false in some regions, it allows the calculation of an upper limit on the NO$_2$ due to LNO and LNO$_2$ profiles. As discussed in Laughner and Cohen (2017), the scattering weights are uniform under cloudy

conditions and the sensitivity of $NO_2$ is nearly constant with different pressure levels because of the high albedo. However, the relative distribution of $LNO_2$ within the UT should be taken carefully into consideration. If the $LNO_2/NO_2$ above the cloud is large enough (Fig. 11a), the $AMF_{LNO_2}$ is largely determined by the ratio of $LNO_2Vis$ to $LNO_2$ which is related to the relative distribution. When the condition of high $LNO_2/NO_2$ is not met, both relative distribution and ratio are important (Fig. 11b).

To clarify this, we applied the same sensitivity test of different simulating LNO amounts for all four methods mentioned in Sect. 2.4: $LNO_2$, $LNO_2Vis$, $LNO_2Clean$ and $NO_2Vis$ (Fig. 12). Note that the threshold for CRF is set to 100% to simplify Eq. (2) to Eq. (7). The overall differences of $LNO_2Clean$ and $NO_2Vis$ are smaller than those of $LNO_2$ and $LNO_2Vis$. Comparing the numerator and denominator in the equations, it is clear why the impact of different simulating LNO amounts is smaller in Fig. 12c and d. For $LNO_2Clean$ and $NO_2Vis$, both the SCD and VCD will increase (decrease) when more (less) $LNO_2$ or $NO_2$ presents. The difference between Fig. 12a and Fig. 12b is the denominator: the total tropospheric $LNO_2$ vertical column and visible $LNO_2$ vertical column respectively. As a result, the negative values in Fig. 12a are caused by the part of $LNO_2$ below the cloud. The uncertainty of retrieved $LNO_2$ and $LNO_x$ PEs is driven by this error, and we conservatively estimate this to be $\pm 13\%$ and $\pm 25\%$ respectively.

## 4  Uncertainties Analysis

The uncertainties of the $LNO_2$ and $LNO_x$ PEs are estimated following Pickering et al. (2016), Allen et al. (2019), Bucsela et al. (2019), Laughner et al. (2019) and Lapierre et al. (2020). We determine the uncertainty due to BEHR tropopause pressure, cloud radiance fraction, cloud pressure, surface pressure, surface reflectivity, profile shape, profile location, $V_{strat}$, the detection efficiency of lightning, $t_{window}$ and $LNO_2$ lifetime numerically by perturbing each parameter in turn and re-retrieving the $LNO_2$ and $LNO_x$ with the perturbed values (Table 5).

The GEOS-5 monthly tropopause pressure, which is consistent with the NASA Standard Product, is applied instead of the variable WRF tropopause height to evaluate the uncertainty (6% for $LNO_2$ PE and 4% for $LNO_x$ PE) caused by the BEHR tropopause pressure. The cloud pressure bias is given as a function of cloud pressure and fraction by Acarreta et al. (2004) implying an uncertainty of 32%, the most likely uncertainty in the production analysis, for $LNO_2$ PE and 34% for $LNO_x$ PE. The resolution of GLOBE terrain height data is much higher than the OMI pixel and a fixed scale height is assumed in the BEHR algorithm. As a result, Laughner et al. (2019) compared the average WRF surface pressures to the GLOBE surface pressures and arrived at the largest bias of 1.5%. Based on the largest bias, we vary the surface pressure (limited to less than 1020 hPa) and the uncertainty can be neglected.

The error in cloud radiance fraction is transformed from cloud fraction using:

$$\sigma = 0.05 \cdot \left. \frac{\partial f_r}{\partial f_g} \right|_{f_{g,pix}} \tag{10}$$

where $f_r$ is the cloud radiance fraction, $f_g$ is the cloud fraction and $f_{g,pix}$ is the cloud fraction of a specific pixel. We calculate $\partial f_r / \partial f_g$ under $f_{g,pix}$ by the relationship between all binned $f_r$ and $f_g$ with the increment of 0.05 for the each specific OMI

orbit. Considering the relationship, the error in cloud fraction is converted to an error in cloud radiance fraction of 2% for the
$LNO_2$ and $LNO_x$ PEs.

The accuracy of the 500 m MODIS albedo product is usually within 5% of albedo observations at the validation sites and those exceptions with low quality flags have been found to be primarily within 10% of the field data (Schaaf et al., 2011). Since we use the bidirectional reflectance distribution function (BRDF) data directly, rather than including a radiative transfer model, 14% Lambertian equivalent reflectivity (LER) error and 10% uncertainty are combined to get a perturbation of 17% (Laughner
et al., 2019). The uncertainty due to surface reflectivity can be neglected with the 17% perturbation.

As discussed at the end of Sect. 3.4, another setting of $LNO_2$ ($2 \times 500$ mol NO flash$^{-1}$) is applied to determine the uncertainty of the lightning parameterization and the vertical distribution of LNO in WRF-Chem. Differences between the two profiles lead to an uncertainty of 13% and 25% in the resulting PEs of $LNO_2$ and $LNO_x$. Another sensitivity test allows each pixel to shift by - 0.2, 0, or + 0.2 degrees in the directions of longitude and latitude, taking advantage of the high-resolution profile location
in WRF-Chem. The resulting uncertainty of $LNO_x$ PE is 1% including the error of transport and chemistry by shifting pixels.

Compared to the NASA standard product v2, Krotkov et al. (2017) demonstrated that the noise in $V_{strat}$ is $1 \times 10^{14}$ cm$^{-2}$. Errors in polluted regions can be slightly larger than this value, while errors in the cleanest areas are typically significantly smaller (Bucsela et al., 2013). We estimated the uncertainty of $V_{strat}$ component and the slant column errors to be 10% and 5%, respectively, following Allen et al. (2019).

Based on the standard deviation of the detection efficiency estimation over the CONUS relative to LIS, ENTLN detection efficiency uncertainties are $\pm$ 16% for total and IC flashes/strokes. Due to the high detection efficiency of CG over the CONUS, the uncertainty is estimated to be $\pm$ 5% (Lapierre et al., 2020). It is found that the resulting uncertainty of detection efficiency is 15% in the production analysis. We have used the $t_{window}$ of 2.4 h for counting ENTLN flashes and strokes to analyze $LNO_2$ and $LNO_x$ production. Because $t_{window}$ derived from the ERA5 reanalysis can not represent the variable wind speeds, a sensitivity
test is performed which yields an uncertainty of 10% for production per flash and 8% for production per stroke using $t_{window}$ of 2 h and 4 h. Meanwhile, the lifetime of UT $NO_x$ ranges from 2 hours to 12 hours depending on the convective location, the methyl peroxy nitrate and alkyl and multifunctional nitrates (Nault et al., 2017). The lifetime ($\tau$) of $NO_2$ in Eq. (6) is replaced by 2 and 12 hours to determine the uncertainty as 24% due to lifetime. This is comparable with the uncertainty (25%) caused by lightning parameterization for the $LNO_x$ type.

Recent works revealed that the modeled $NO/NO_2$ ratio departs from the data in the SEAC[4]RS aircraft campaign (Travis et al., 2016; Silvern et al., 2018). Silvern et al. (2018) attributed this to the positive interference on the $NO_2$ measurements or errors in the cold-temperature $NO$-$NO_2$-$O_3$ photochemical reaction rate. We assign a 20% bias with $\pm$ 15% uncertainty to this error considering the possible positive $NO_2$ measurements interferences (Allen et al., 2019; Bucsela et al., 2019) and estimate the uncertainty to be 15% for $LNO_x$ PE.

In addition, the estimation of $LNO_x$ PE also depends on the tropospheric background $NO_2$. In our method, main factors affecting this factor are the emissions inventory and the amount of transported $NO_2$. For the emissions inventory, the sources of uncertainty are assumptions, methods, input data and calculation errors. As a result, the uncertainties for different species or pollutants related to $NO_2$ are different and EPA also doesn't publish the quantified uncertainty measures because the parties

that submit emissions estimates to EPA are not asked to include quantitative uncertainty measurements or estimates (EPA, 2015). For the simulated convective transport, Li et al. (2018) compared the cloud-resolving simulations with these based on convective parameterization and pointed out that the convective transport was weaker in the parameterization. But, we believe that the ratio condition ($LNO_2 Vis/NO_2 Vis \geq 50\%$) should reduce these two kinds of uncertainty and assume an uncertainty of 10%, which is less than 20% assigned in Allen et al. (2019) and Bucsela et al. (2019).

The overall uncertainty is estimated as the square root of the sum of the squares of all individual uncertainties in Table 5. The net uncertainty is 48% and 56% for $LNO_2$ type and $LNO_x$ type respectively. The mean $LNO_2$/flash, $LNO_x$/flash, $LNO_2$/stroke, $LNO_x$/stroke based on the linear regression and summation method are 32 mol/flash, 90 mol/flash, 6 mol/stroke and 17 mol/stroke. Applying the corresponding uncertainty to these mean values, we arrive at $32 \pm 15$ mol $LNO_2$/flash, $90 \pm 50$ mol $LNO_x$/flash, $6 \pm 3$ mol $LNO_2$/stroke and $17 \pm 10$ mol $LNO_x$/stroke. This is in the range of current literature estimate ranging from 33 to 500 mol $LNO_x$/flash (Schumann and Huntrieser, 2007; Beirle et al., 2010; Bucsela et al., 2010). Bucsela et al. (2010) estimated $LNO_x$ PE of $100 - 250$ mol/flash which is higher than but overlaps with our estimate. Pickering et al. (2016) estimated $LNO_x$ PE to be $80 \pm 45$ mol per flash for the Gulf of Mexico. This is 50% smaller than our flash-based results over the CONUS, if we use the same linear regression method which is based on the daily summed values instead of daily mean values. Note that the criteria defined in Sect. 3.1 lead to many missing data over the Gulf of Mexico, thus it is actually a comparison between different regions. For the stroke-based results, Lapierre et al. (2020) yields lower $LNO_2$ PE of $1.6 \pm 0.1$ mol per stroke, the difference is caused by the different version of BEHR algorithm and several settings as mentioned in Sect. 3.2. Bucsela et al. (2019) inferred an average value of $200 \pm 110$ moles (122% larger than our results) $LNO_x$ produced per flash over the North America, this is related to the different algorithm, lightning data and lightning thresholds.

## 5 Conclusions

In this study, a new algorithm for retrieving $LNO_2$ ($LNO_x$) from OMI, including $LNO_2$ ($LNO_x$) below cloud, has been developed for application over active convection. It works in both clean and polluted regions because of the consideration of tropospheric background pollution in the definition of AMFs. It uses specific criteria combining with several other conditions (sufficient CRF, coincident ENTLN data, TL $\geq 1000$ and ratio $\geq 50\%$) to ensure that the electrically active regions are detected by OMI and simulated by WRF-Chem successfully. We conducted an analysis on $1° \times 1°$ daily boxes in MJJA 2014 and obtained the seasonal mean $LNO_2$ and $LNO_x$ production efficiencies over the CONUS. Considering all the uncertainties (Table 5) and applying the summation and regression method, the final mean production efficiencies are estimated to be $32 \pm 15$ mol $LNO_2$/flash, $90 \pm 50$ mol $LNO_x$/flash, $6 \pm 3$ mol $LNO_2$/stroke and $17 \pm 10$ mol $LNO_x$/stroke.

Compared with Lapierre et al. (2020), we find that the $LNO_2$ production could be larger when the below-cloud $LNO_2$ is taken into account, especially for the high clouds. Meanwhile, if the method of Pickering et al. (2016) is applied without the background $NO_2$ correction, the derived $LNO_x$ production efficiency is similar to ours in clean regions or regions with high $LNO_2$ concentration above the cloud, but it could be overestimated more than 18% in polluted regions. Finally, implementing profiles generated with different model settings of lightning ($1 \times 200$ mol NO flash$^{-1}$ and $2 \times 500$ mol NO flash$^{-1}$), we find that

the larger LNO production setting leads to 62% larger retrieval of $LNO_x$ on average despite some regionally dependent effects caused by the nonlinear calculation of AMF. Both the ratio of the tropospheric $LNO_2$ above the cloud to the total tropospheric $LNO_2$ and the ratio of $LNO_2$ to $NO_2$ cause different comprehensive effects due to the nonlinear calculation of $AMF_{LNO_2}$ and $AMF_{LNO_x}$.

Since other regions, like China and India, have much more $NO_2$ pollution than the CONUS, it is necessary to consider the background $NO_2$ in detail. These analyses will be complemented by the recently launched satellite instrument (TROPOspheric Monitoring Instrument [TROPOMI]) (Veefkind et al., 2012; Boersma et al., 2018; Griffin et al., 2019) and Lightning Mapping Imager (LMI) on the new generation Chinese geostationary meteorological satellites Fengyun-4 (Min et al., 2017; Yang et al., 2017; Zhang et al., 2019). Future work investigating the flash channel length and more detailed lightning parameterization in WRF-Chem would greatly benefit $LNO_x$ estimation. Applying current method in future studies may enhance the accuracy of $LNO_x$ production at both local and global scales.

*Code and data availability.* The retrieval algorithm used in Sect. 2.4 is available at https://github.com/zxdawn/BEHR-LNOx (last access: March 8, 2020; Zhang and Laughner, 2019). The WRF-Chem model output and $LNO_x$ product are available upon request to Xin Zhang (xinzhang1215@gmail.com).

## Appendix A: AMF Definitions used in this Study

$$AMF_{\mathrm{LNO_2}} = \frac{(1-f_r)\int_{p_{surf}}^{p_{tp}} w_{clear}(p)NO_2(p)\,dp + f_r \int_{p_{cloud}}^{p_{tp}} w_{cloudy}(p)NO_2(p)\,dp}{\int_{p_{surf}}^{p_{tp}} LNO_2(p)\,dp} \tag{A1}$$

$$AMF_{\mathrm{LNO_x}} = \frac{(1-f_r)\int_{p_{surf}}^{p_{tp}} w_{clear}(p)NO_2(p)\,dp + f_r \int_{p_{cloud}}^{p_{tp}} w_{cloudy}(p)NO_2(p)\,dp}{\int_{p_{surf}}^{p_{tp}} LNO_x(p)\,dp} \tag{A2}$$

where $f_r$ is the cloud radiance fraction, $p_{surf}$ is the surface pressure, $p_{tp}$ is the tropopause pressure, $p_{cloud}$ is the cloud optical pressure (CP), $w_{clear}$ and $w_{cloudy}$ are respectively the pressure dependent scattering weights from the TOMRAD lookup table (Bucsela et al., 2013) for clear and cloudy parts, and $NO_2(p)$ is the modeled $NO_2$ vertical profile. $LNO_2(p)$ and $LNO_x(p)$ are respectively the $LNO_2$ and $LNO_x$ vertical profile calculated by the difference of vertical profiles between WRF-Chem simulations with and without lightning.

$$AMF_{\mathrm{LNO_2Clean}} = \frac{(1-f_r)\int_{p_{surf}}^{p_{tp}} w_{clear}(p)LNO_2(p)\,dp + f_r \int_{p_{cloud}}^{p_{tp}} w_{cloudy}(p)LNO_2(p)\,dp}{\int_{p_{surf}}^{p_{tp}} LNO_2(p)\,dp} \tag{A3}$$

$$AMF_{\mathrm{NO_2Vis}} = \frac{(1-f_r)\int_{p_{surf}}^{p_{tp}} w_{clear}(p)NO_2(p)\,dp + f_r \int_{p_{cloud}}^{p_{tp}} w_{cloudy}(p)NO_2(p)\,dp}{(1-f_g)\int_{p_{surf}}^{p_{tp}} NO_2(p)\,dp + f_g \int_{p_{cloud}}^{p_{tp}} NO_2(p)\,dp} \tag{A4}$$

$$AMF_{\text{NO}_x\text{Vis}} = \frac{(1-f_r)\int_{p_{\text{surf}}}^{p_{\text{tp}}} w_{\text{clear}}(p)NO_2(p)\,dp + f_r \int_{p_{\text{cloud}}}^{p_{\text{tp}}} w_{\text{cloudy}}(p)NO_2(p)\,dp}{(1-f_g)\int_{p_{\text{surf}}}^{p_{\text{tp}}} NO_x(p)\,dp + f_g \int_{p_{\text{cloud}}}^{p_{\text{tp}}} NO_x(p)\,dp} \tag{A5}$$

$$AMF_{\text{LNO}_2\text{Vis}} = \frac{(1-f_r)\int_{p_{\text{surf}}}^{p_{\text{tp}}} w_{\text{clear}}(p)NO_2(p)\,dp + f_r \int_{p_{\text{cloud}}}^{p_{\text{tp}}} w_{\text{cloudy}}(p)NO_2(p)\,dp}{(1-f_g)\int_{p_{\text{surf}}}^{p_{\text{tp}}} LNO_2(p)\,dp + f_g \int_{p_{\text{cloud}}}^{p_{\text{tp}}} LNO_2(p)\,dp} \tag{A6}$$

where $f_g$ is the geometric cloud fraction and $NO_x(p)$ is the modeled $NO_x$ vertical profile.

## Appendix B:  LNO$_x$ Production based on Lower Lightning Thresholds

While we used 2400 flashes box$^{-1}$ and 8160 strokes box$^{-1}$ per 2.4 hour time window for detecting LNO$_x$, here we show results obtained when using 1 flash box$^{-1}$ and 3.4 strokes box$^{-1}$ in the same time window. We note that the WRF total lightning threshold is also reduced to 1 flash box$^{-1}$, but we keep the ratio condition unchanged. Briefly, the condition is CRF90_ENTLN1(3.4)_TL1_ratio50 as shown in Table 1.

Similarly, the order of estimated daily PEs is LNO$_2$Clean > LNO$_2$ > NO$_2$Vis > LNO$_2$Vis (Fig. B1). Compared with Fig. 4,
the LNO$_2$ per flash and LNO$_x$ per flash are larger while PEs based on stroke data are smaller. Considering the additional boxes of fewer lightning counts, differences in the daily mean flashes and NO$_x$ results in different PEs and the relationship presents more like the power function as mentioned in Bucsela et al. (2019).

Instead of using the nonlinear regression of power function:

$$y = \alpha x^{\beta} \tag{B1}$$

where $x$ is flashes or strokes and $y$ is NO$_2$ or NO$_x$, we take the logarithm of both sides and apply the linear regression to data:

$$\log_{10} y = \log_{10} \alpha + \beta \log_{10} x \tag{B2}$$

As expected, the linear regression based on logarithmized data performs better in this situation and yields $\alpha$ = 38 kmol, and $\beta$ = 0.3 for LNO$_x$ per flash (Fig. B2). Since we use the unbinned data (flashes not divided into many groups), we compare our results with Bucsela et al. (2019) based on the same kind of data ($\alpha$ = 10.3 kmol, and $\beta$ = 0.42). The large difference of $\alpha$
is related to the method of estimating LNO$_x$, different lightning data (WWLLN and ENTLN) and different regions (northern midlatitudes and CONUS). Note that the resolution (13 $\times$ 24 km$^2$) of OMI could weaken the signal of LNO$_x$. We believe the phenomenon of higher production efficiency as flash rate decreases (Fig. B3) could be explored in much detail with higher resolution data like the TROPOMI data.

*Author contributions.* YY directed the research and RJvdA, XZ and YY designed the research with feedback from the other co-authors;
RJvdA and XZ developed the algorithm; JLL provided guidance and supporting data on the ENTLN data; XZ performed simulations and

analysis with the help of YY, RJvdA, QC, XK, SY, JC, CH and RS; YY, RJvdA, JLL and XZ interpreted the data and discussed the results. XZ drafted the manuscript with comments from the co-authors; JLL, RJvdA and YY edited the manuscript.

*Competing interests.* The authors declare that they have no conflict of interest.

*Acknowledgements.* This work was funded by the National Natural Science Foundation of China (91644224 and 41705118). We acknowl-
edge use of the computational resource provided by the National Supercomputer Centre in Guangzhou (NSCC-GZ). We thank the Uni-
versity of California Berkeley Satellite Group for the basic BEHR algorithm. We also thank Earth Networks Company for providing the
Earth Networks Total Lightning Network (ENTLN) datasets. We appreciate the discussions with Joshua L. Laughner for BEHR codes and
Mary Barth for the WRF-Chem lightning $NO_x$ module. MOZART-4 global model output is available at https://www.acom.ucar.edu/wrf-
chem/mozart.shtml (last access: March 8, 2020). The authors would also like to thank all anonymous reviewers as well as Kenneth E.
Pickering, Eric J. Bucsela and Dale J. Allen for detailed comments which greatly improved this manuscript. Finally, we thank all contribu-
tors of Python packages used in this paper (Met Office, 2010 - 2015; Hoyer and Hamman, 2017; Hunter, 2007; Jiawei Zhuang et al., 2019;
McKinney, 2011; Inc., 2015; Seabold and Perktold, 2010; van der Walt et al., 2011; Waskom et al., 2017).

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

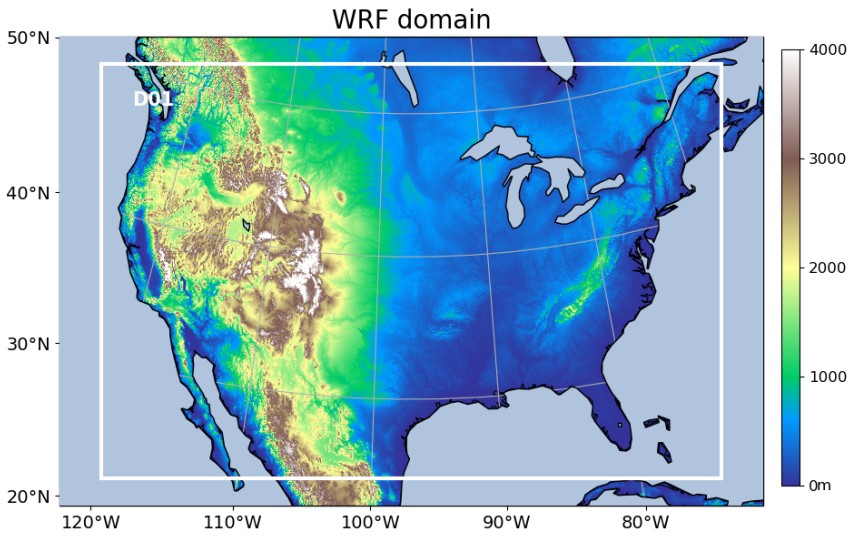

**Figure 1.** Domain and terrain height (m) of the WRF-Chem simulation with 350 x 290 grid cells and a horizontal resolution of 12 km.

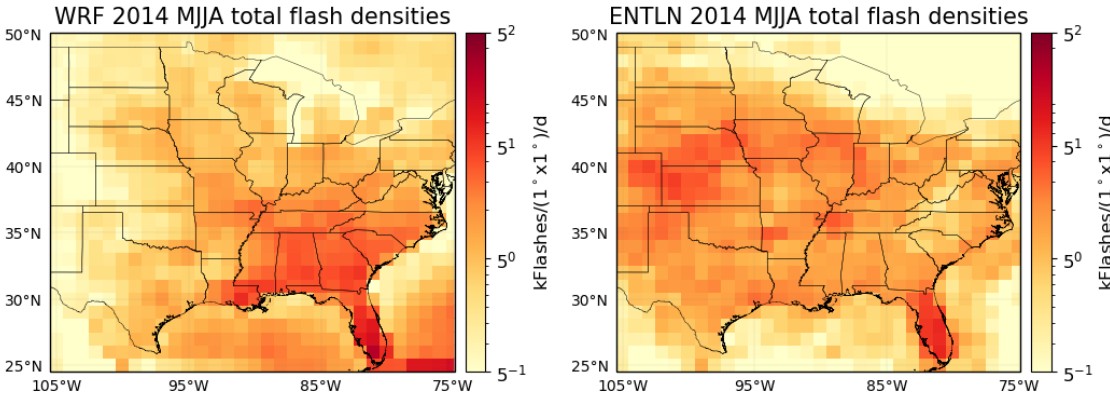

**Figure 2.** Comparison between total flash densities from ENTLN and WRF-Chem during MJJA 2014.

**Table 1.** Definitions of the abbreviations for the criteria used in this study.

| Abbreviations | Full form [source] |
| --- | --- |
| CRF | Cloud radiance fraction [OMI] |
| CP | Cloud optical pressure [OMI] |
| CF | Cloud fraction [WRF-Chem] |
| TL | Total lightning flashes [WRF-Chem] |
| ratio | modeled $LNO_2$ Vis / modeled $NO_2$ Vis [WRF-Chem] |
| CRF$\alpha$_ENTLN | CRF $\geq \alpha$ + ENTLN flashes(strokes) $\geq$ 2400(8160) [ENTLN] |
| CRF$\alpha$_CF40_ENTLN | CRF $\geq \alpha$ + ENTLN flashes(strokes) $\geq$ 2400(8160) + CF $\geq$ 40% |
| CRF$\alpha$_ENTLN_TL1000 | CRF $\geq \alpha$ + ENTLN flashes(strokes) $\geq$ 2400(8160) + TL $\geq$ 1000 |
| CRF$\alpha$_CF40_ENTLN_TL1000 | CRF $\geq \alpha$ + ENTLN flashes(strokes) $\geq$ 2400(8160) + CF $\geq$ 40% + TL $\geq$ 1000 |
| CRF$\alpha$_ENTLN_TL1000_ratio50 | CRF $\geq \alpha$ + ENTLN flashes(strokes) $\geq$ 2400(8160) + TL $\geq$ 1000 + ratio $\geq$ 50% |
| CRF$\alpha$_CF40_ENTLN_TL1000_ratio50 | CRF $\geq \alpha$ + ENTLN flashes(strokes) $\geq$ 2400(8160) + CF $\geq$ 40% + TL $\geq$ 1000 + ratio $\geq$ 50% |
| CRF$\alpha$_ENTLN1(3.4)_TL1_ratio50 | CRF $\geq \alpha$ + ENTLN flashes(strokes) $\geq$ 1(3.4) + TL $\geq$ 1 + ratio $\geq$ 50% |

$\alpha$ has three options: 70%, 90% and 100%

**Table 2.** $LNO_x$ production efficiencies for different combinations of criteria defined in Table 1.

| Condition[1] | ENTLN data type[2] | $LNO_x$/flash or $LNO_x$/stroke | R value | Intercept ($10^6$ mol) | Days[3] |
| --- | --- | --- | --- | --- | --- |
| CRF90_ENTLN | Flash | $52.1 \pm 51.1$ | 0.20 | 0.21 | 99 |
| CRF90_CF40_ENTLN | Flash | $84.2 \pm 31.5$ | 0.54 | -0.04 | 70 |
| CRF90_ENTLN_TL1000 | Flash | $61.9 \pm 49.1$ | 0.27 | 0.33 | 83 |
| CRF90_CF40_ENTLN_TL1000 | Flash | $63.4 \pm 52.9$ | 0.38 | 0.26 | 38 |
| CRF90_ENTLN_TL1000_ratio50 | Flash | $54.5 \pm 48.1$ | 0.25 | 0.39 | 81 |
| CRF90_CF40_ENTLN_TL1000_ratio50 | Flash | $90.0 \pm 65.0$ | 0.46 | 0.15 | 32 |
| CRF90_ENTLN | Stroke | $6.7 \pm 4.1$ | 0.31 | 0.23 | 102 |
| CRF90_CF40_ENTLN | Stroke | $10.3 \pm 3.6$ | 0.55 | 0.08 | 79 |
| CRF90_ENTLN_TL1000 | Stroke | $7.5 \pm 5.1$ | 0.29 | 0.38 | 94 |
| CRF90_CF40_ENTLN_TL1000 | Stroke | $8.6 \pm 6.2$ | 0.39 | 0.27 | 46 |
| CRF90_ENTLN_TL1000_ratio50 | Stroke | $7.0 \pm 4.8$ | 0.29 | 0.42 | 93 |
| CRF90_CF40_ENTLN_TL1000_ratio50 | Stroke | $8.9 \pm 7.0$ | 0.39 | 0.31 | 40 |

[1] These conditions are defined in Table 1. [2] The thresholds of ENTLN data are 2400 flashes box$^{-1}$ and 8160 strokes box$^{-1}$ during the period of 2.4 h before OMI overpass time. [3] The number of valid days with specific criteria in MJJA 2014.

**Table 3.** $LNO_x$ production efficiencies for different thresholds of CRF with coincident ENTLN data, $TL \geq 1000$ and ratio $\geq 50\%$.

| CRF (%) | ENTLN data type[1] | $LNO_x$/flash or $LNO_x$/stroke | R value | Intercept ($10^5$ mol) | Days[2] |
|---------|---------------------|---------------------------------|---------|------------------------|---------|
| 70 | Flash | $35.7 \pm 36.8$ | 0.21 | 4.91 | 85 |
| 90 | Flash | $54.5 \pm 48.1$ | 0.25 | 3.90 | 81 |
| 100 | Flash | $20.8 \pm 37.4$ | 0.13 | 5.67 | 71 |
| 70 | Stroke | $4.1 \pm 3.9$ | 0.21 | 5.16 | 96 |
| 90 | Stroke | $7.0 \pm 4.8$ | 0.29 | 4.16 | 93 |
| 100 | Stroke | $2.6 \pm 4.0$ | 0.14 | 5.41 | 82 |

[1] The thresholds of ENTLN data are 2400 flashes $box^{-1}$ and 8160 strokes $box^{-1}$ during the period of 2.4 h before OMI overpass time. [2] The number of valid days with specific criteria in MJJA 2014.

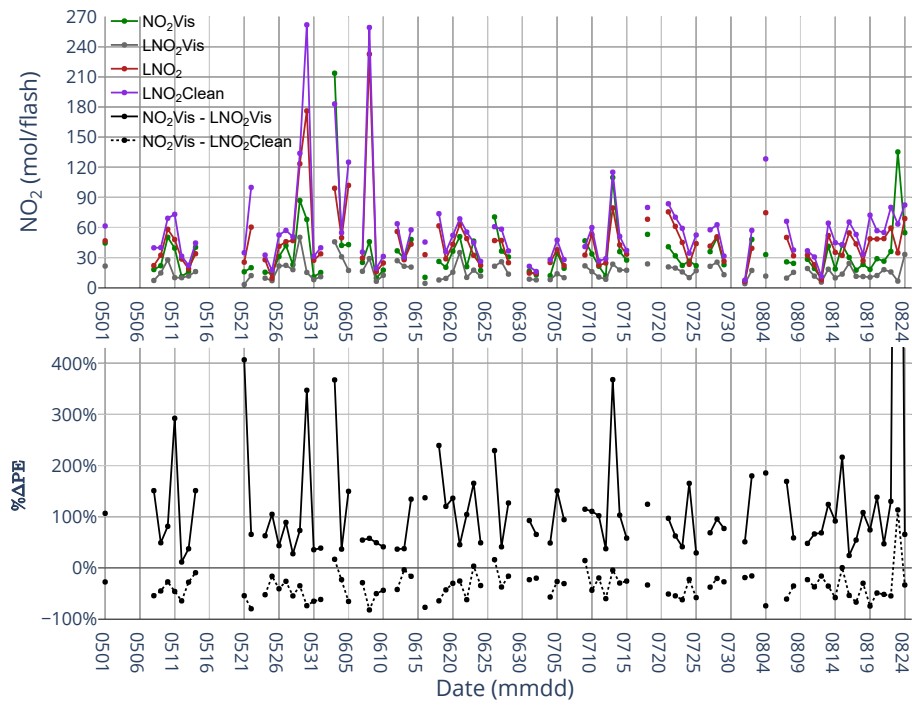

**Figure 3.** (top) Time series of $NO_2$ Vis, $LNO_2$ Vis, $LNO_2$ and $LNO_2$ Clean production per day over the CONUS for MJJA 2014 with $CRF \geq 90\%$ and a flash threshold of 2400 flashes per 2.4 h. (bottom) Time series of the percent differences between $NO_2$ Vis and $LNO_2$ Vis and the percent differences between $NO_2$ Vis and $LNO_2$ Clean with $CRF \geq 90\%$. The value of black dot on August 23 (not shown) is 1958%.

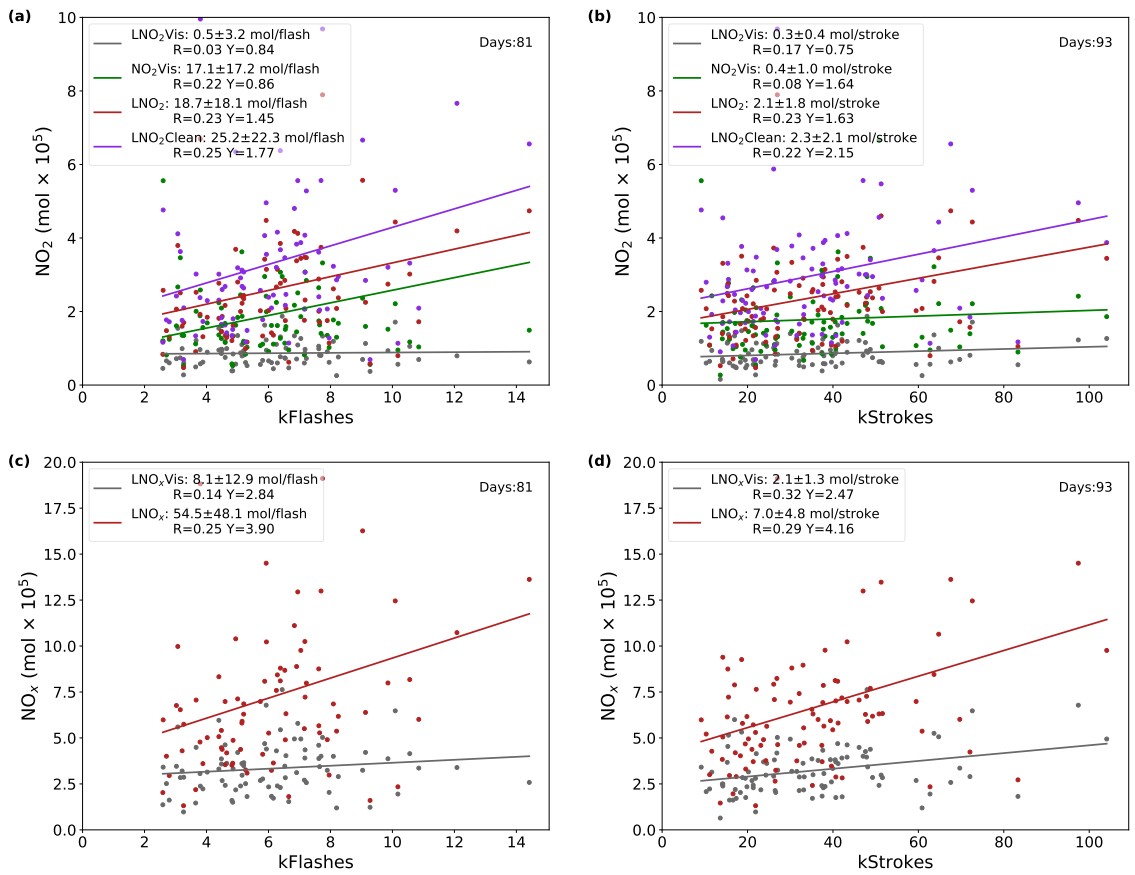

**Figure 4.** (a) Daily $NO_2$Vis, $LNO_2$Vis, $LNO_2$ and $LNO_2$Clean versus ENTLN total flashes data. (b) Same as (a) but for strokes. (c) Daily $LNO_x$Vis and $LNO_x$ versus total flashes. (d) Same as (c) but for strokes.

**Table 4.** The percent changes in the estimated production when using different methods based on the same a priori profiles.

|  | City[1] | ($LNO_2$Clean - $LNO_2$)/$LNO_2$ | ($LNO_2$ - TropVis)/TropVis | ($LNO_2$Clean-TropVis)/TropVis |
|---|---|---|---|---|
| **Polluted** | Lansing | 24.2% | 49.5% | 85.6% |
|  | New Orleans | 13.3% | 121.2% | 153.8% |
|  | Orlando | 46.3% | 37.5% | 101.3% |
| **Clean** | Huron | 12.0% | 56.4% | 75.2% |
|  | Charles Town | 12.0% | 82.2% | 104.1% |
|  | Tarboro | 5.0% | 86.0% | 95.3% |

[1] Locations are denoted in Fig. 6a.

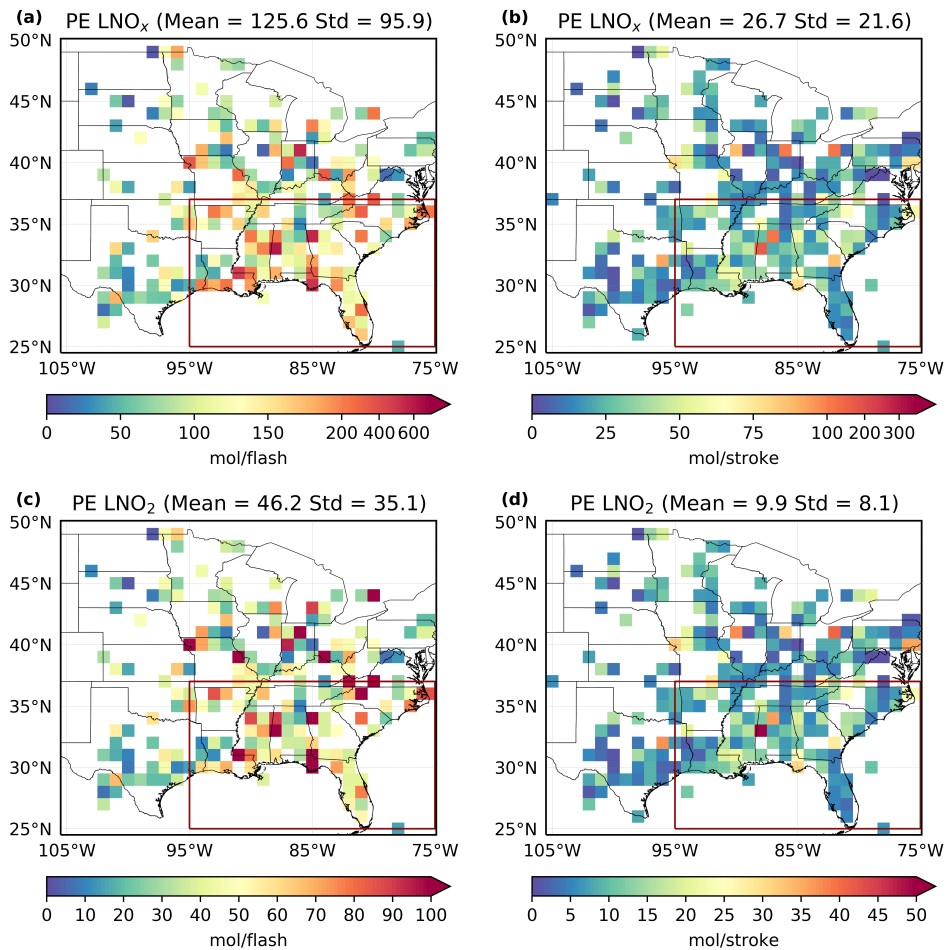

**Figure 5.** (a) and (c) Maps of $1° \times 1°$ gridded values of mean $LNO_x$ and $LNO_2$ production per flash with CRF $\geq$ 90% for MJJA 2014. (b) and (d) Same as (a) and (c) except for strokes. The southeastern US is denoted by the red box in panels a – d.

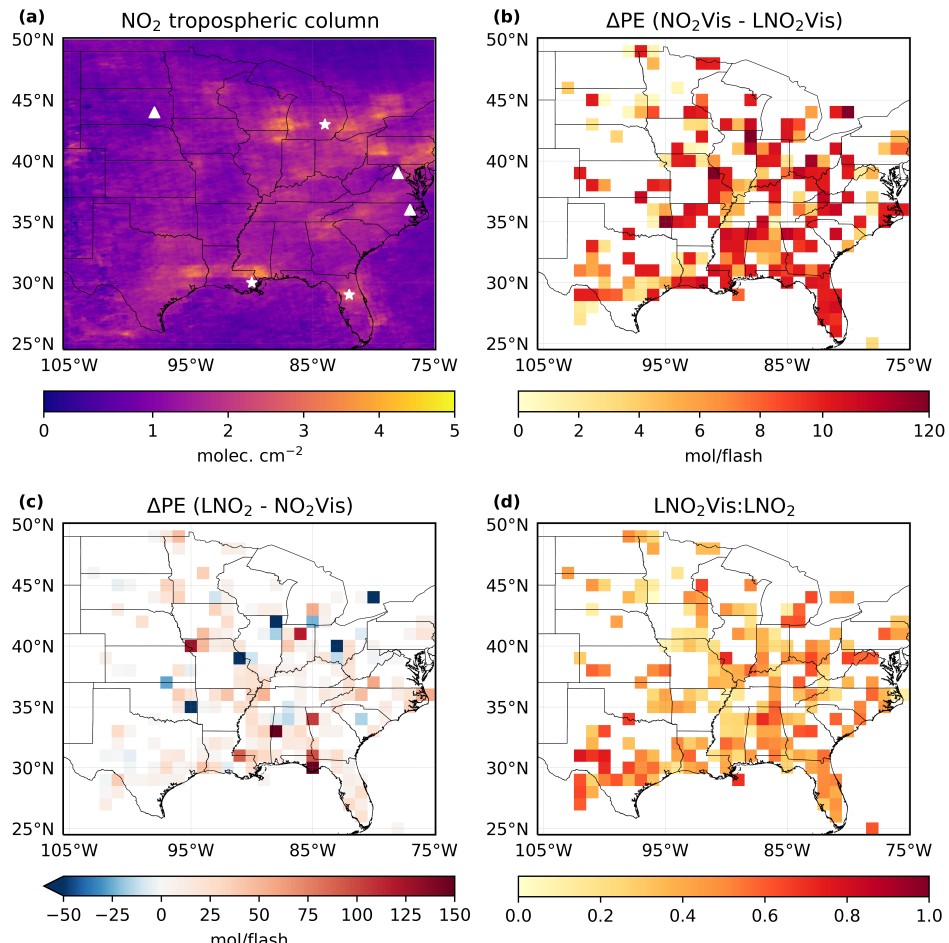

**Figure 6.** (a) Mean (MJJA 2014) $NO_2$ tropospheric column. Polluted cities are denoted by stars: Lansing, New Orleans and Orlando while clean cities are denoted by triangles: Huron, Charles Town and Tarboro. (b) The differences of the estimated mean production efficiency between $NO_2$Vis and $LNO_2$Vis with CRF $\geq$ 90%. (c) The same differences as (b) but between $LNO_2$ and $NO_2$Vis. (d) The ratio of $LNO_2$Vis to $LNO_2$.

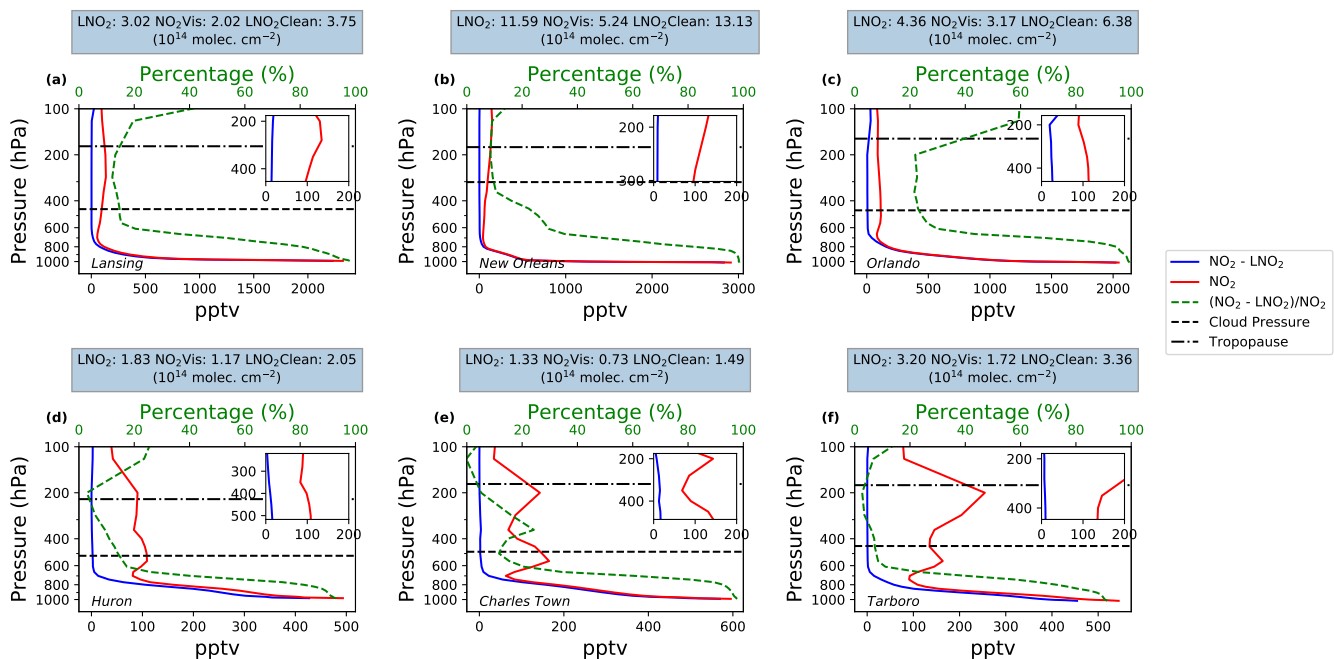

**Figure 7.** Comparisons of mean WRF-Chem $NO_2$ and background $NO_2$ profiles in six grids with CRF $\geq$ 100% on specific days during MJJA 2014. The top row data are selected from polluted regions (stars in Fig. 6a) while the bottom row data are from clean regions (triangles in Fig. 6a). The green dashed lines are the mean ratio profiles of background $NO_2$ to total $NO_2$. The zoomed figures show the profiles from the cloud pressure to the tropopause. The titles present the mean productions based on three different methods mentioned in Sect. 2.4.

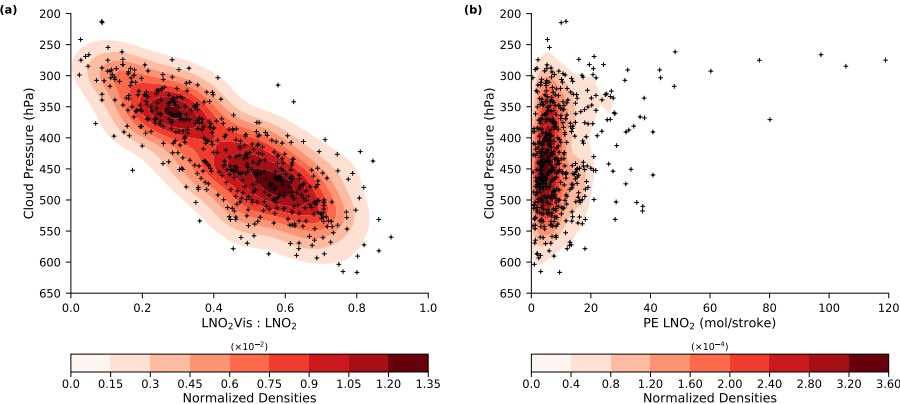

**Figure 8.** Kernel density estimation of the (a) daily ratio of $LNO_2$Vis to $LNO_2$ and (b) daily $LNO_2$ production efficiency versus the daily cloud pressure measured by OMI with CRF $\geq$ 90% for MJJA 2014. The kernel density estimation was generated by kdeplot in the Python package named seaborn.

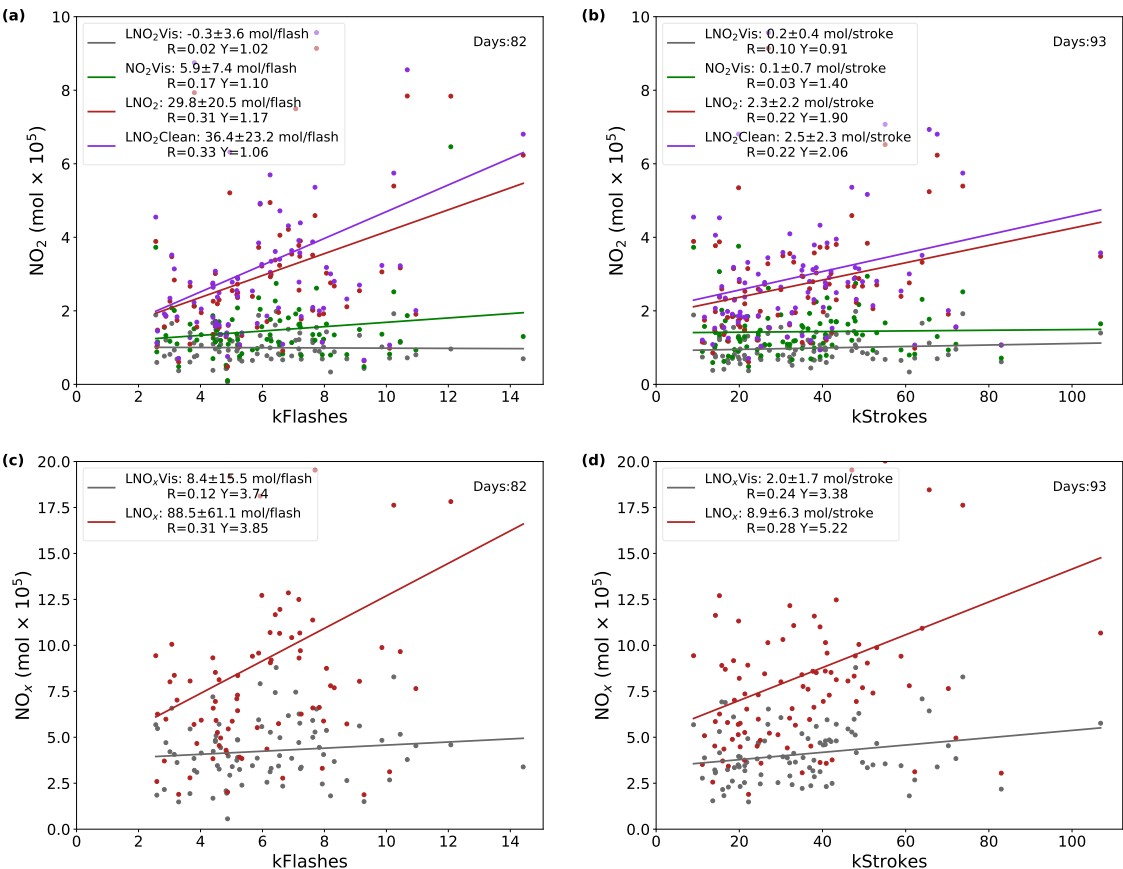

**Figure 9.** Same as Fig. 4 except for the $2 \times 500$ mol NO flash$^{-1}$ configuration.

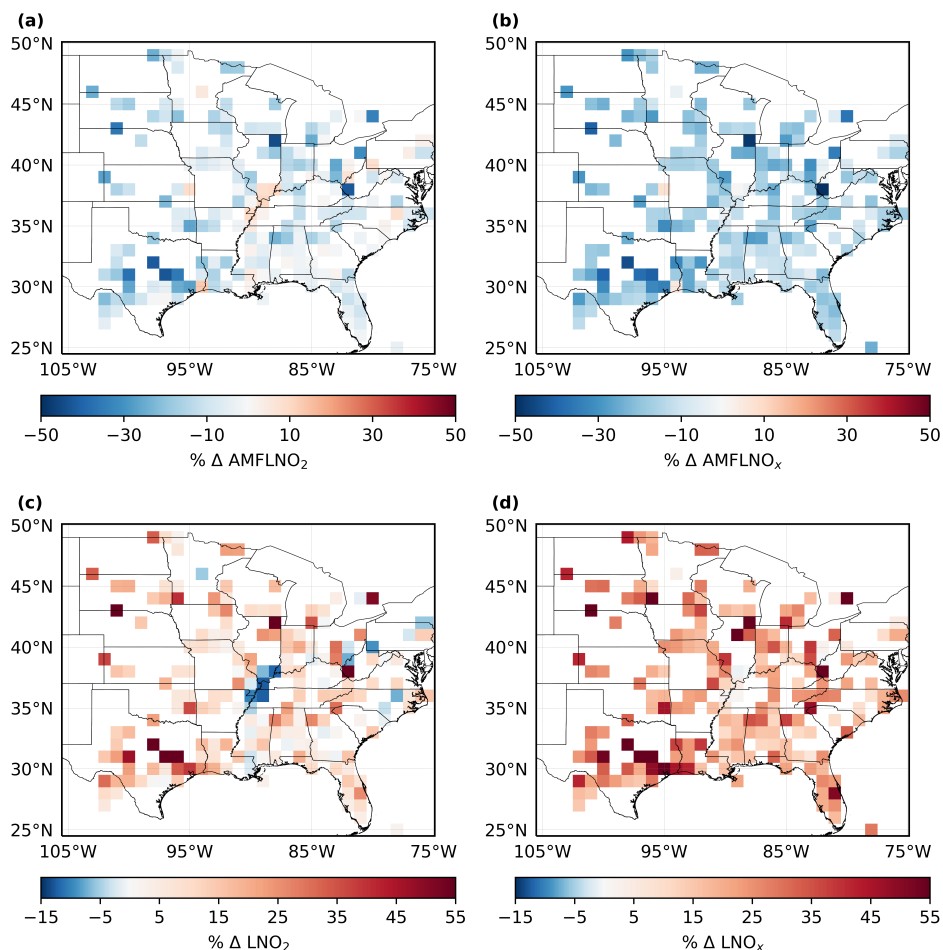

**Figure 10.** Average percent differences in (a) $AMF_{LNO_2}$, (b) $AMF_{LNO_x}$, (c) $LNO_2$ and (d) $LNO_x$ with CRF $\geq 90\%$ over MJJA 2014. Differences between profiles are generated by $2 \times 500$ mol NO flash$^{-1}$ and $1 \times 200$ mol NO flash$^{-1}$.

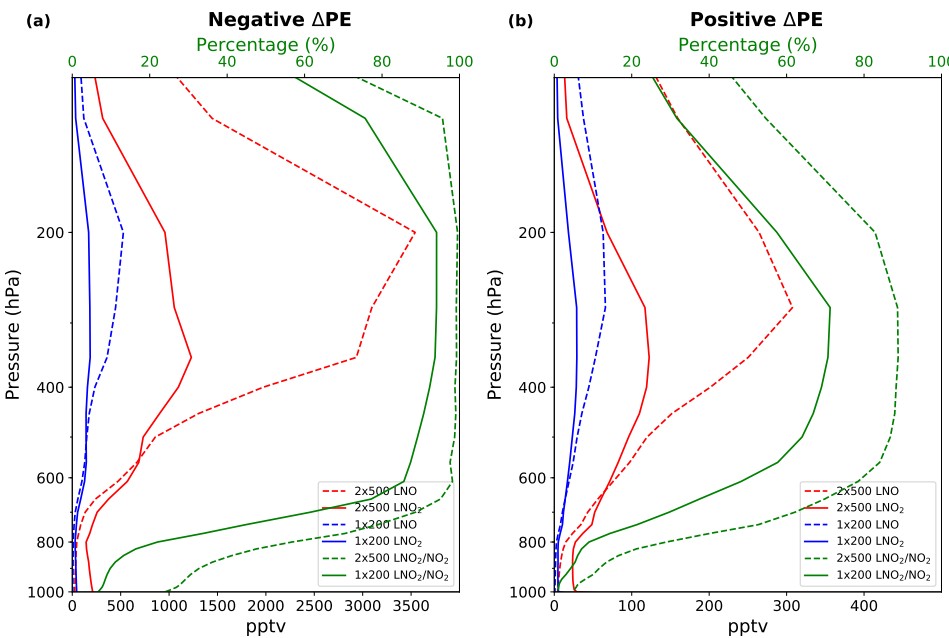

**Figure 11.** LNO and LNO$_2$ profiles with different LNO settings at (a) the region containing the minimal negative percent change in LNO$_2$ and (b) the region containing the largest positive percent change in LNO$_2$ when the LNO setting is changed from $1 \times 200$ mol NO flash$^{-1}$ to $2 \times 500$ mol NO flash$^{-1}$, averaged over MJJA 2014. The profiles using $1 \times 200$ ($2 \times 500$) mol NO flash$^{-1}$ are shown in blue (red) lines. Solid (dashed) green lines are the mean ratio of LNO$_2$ to NO$_2$ with $1 \times 200$ ($2 \times 500$) mol NO flash$^{-1}$.

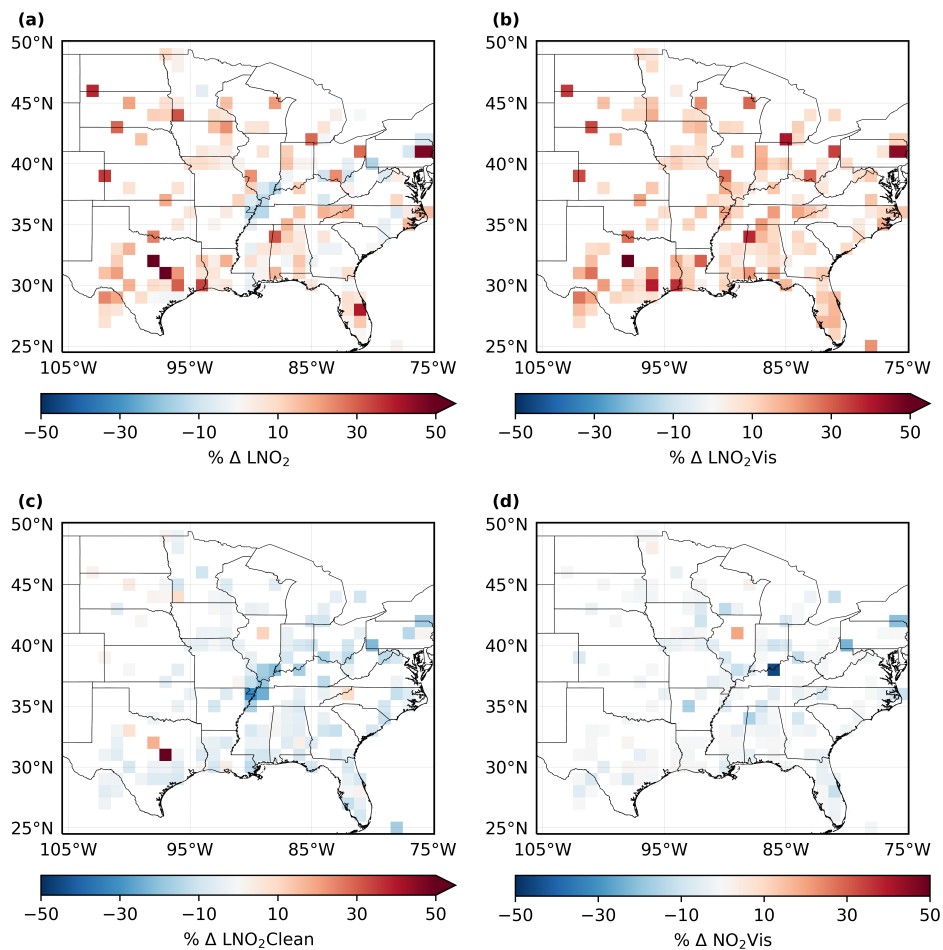

**Figure 12.** Average percent differences in (a) $LNO_2$, (b) $LNO_2Vis$, (c) $LNO_2Clean$ and (d) $NO_2Vis$ with CRF = 100% over MJJA 2014.

**Table 5.** Uncertainties for the estimation of LNO$_2$/flash, LNO$_x$/flash, LNO$_2$/stroke and LNO$_x$/stroke.

| Type | Perturbation | LNO$_2$/flash[5] | LNO$_x$/flash[5] | LNO$_2$/stroke[5] | LNO$_x$/stroke[5] |
|---|---|---|---|---|---|
| BEHR tropopause pressure[1] | NASA product tropopause | 6 | 4 | 6 | 4 |
| Cloud radiance fraction[1] | $\pm$ 5% | 2 | 2 | 2 | 2 |
| Cloud pressure[2] | Variable | 32 | 34 | 32 | 34 |
| Surface pressure[1] | $\pm$ 1.5% | 0 | 0 | 0 | 0 |
| Surface reflectivity[1] | $\pm$ 17% | 0 | 0 | 0 | 0 |
| LNO$_2$ profile[1] | 2$\times$500 mol NO flash$^{-1}$ | 13 | 25 | 13 | 25 |
| Profile location[1] | Quasi-Monte Carlo | 0 | 1 | 0 | 1 |
| Lightning detection efficiency[3] | IC: $\pm$ 16%, CG: $\pm$ 5% | 15 | 15 | 15 | 15 |
| $t_{window}$[3] | 2 – 4 hours | 10 | 10 | 8 | 8 |
| LNO$_x$ lifetime[3] | 2 – 12 hours | 24 | 24 | 24 | 24 |
| $V_{strat}$[4] | - | 10 | 10 | 10 | 10 |
| Systematic errors in slant column[4] | - | 5 | 5 | 5 | 5 |
| Tropospheric background[4] | - | 10 | 10 | 10 | 10 |
| NO/NO$_2$[4] | 20% $\pm$ 15% | 0 | 15 | 0 | 15 |
| Net | - | 49 | 56 | 48 | 56 |

PE$_{uncertainty}$ = (Error$_{rising\ perturbed\ value}$ - Error$_{lowering\ perturbed\ value}$)/2 where Error$_{perturbed\ value}$ = (PE$_{perturbed\ value}$ - PE$_{original\ value}$)/PE$_{original\ value}$.
[1] Laughner et al. (2019) [2] Acarreta et al. (2004) [3] Lapierre et al. (2020) [4] Allen et al. (2019) and Bucsela et al. (2019) [5] Uncertainty (%)

**Table A1.** Simple forms of abbreviations for AMFs.

| Abbreviations | Numerator[1] | Denominator[2] |
|---|---|---|
| AMF$_{LNO_2}$ | S$_{NO_2}$ | V$_{LNO_2}$ |
| AMF$_{LNO_2\,Vis}$ | S$_{NO_2}$ | V$_{LNO_2\,Vis}$ |
| AMF$_{LNO_2\,Clean}$ | S$_{LNO_2}$ | V$_{LNO_2}$ |
| AMF$_{NO_2\,Vis}$ | S$_{NO_2}$ | V$_{NO_2\,Vis}$ |
| AMF$_{LNO_x}$ | S$_{NO_2}$ | V$_{LNO_x}$ |
| AMF$_{NO_x\,Vis}$ | S$_{NO_2}$ | V$_{NO_x\,Vis}$ |

[1] The part of simulated VCD seen by OMI [2] The simulated VCD

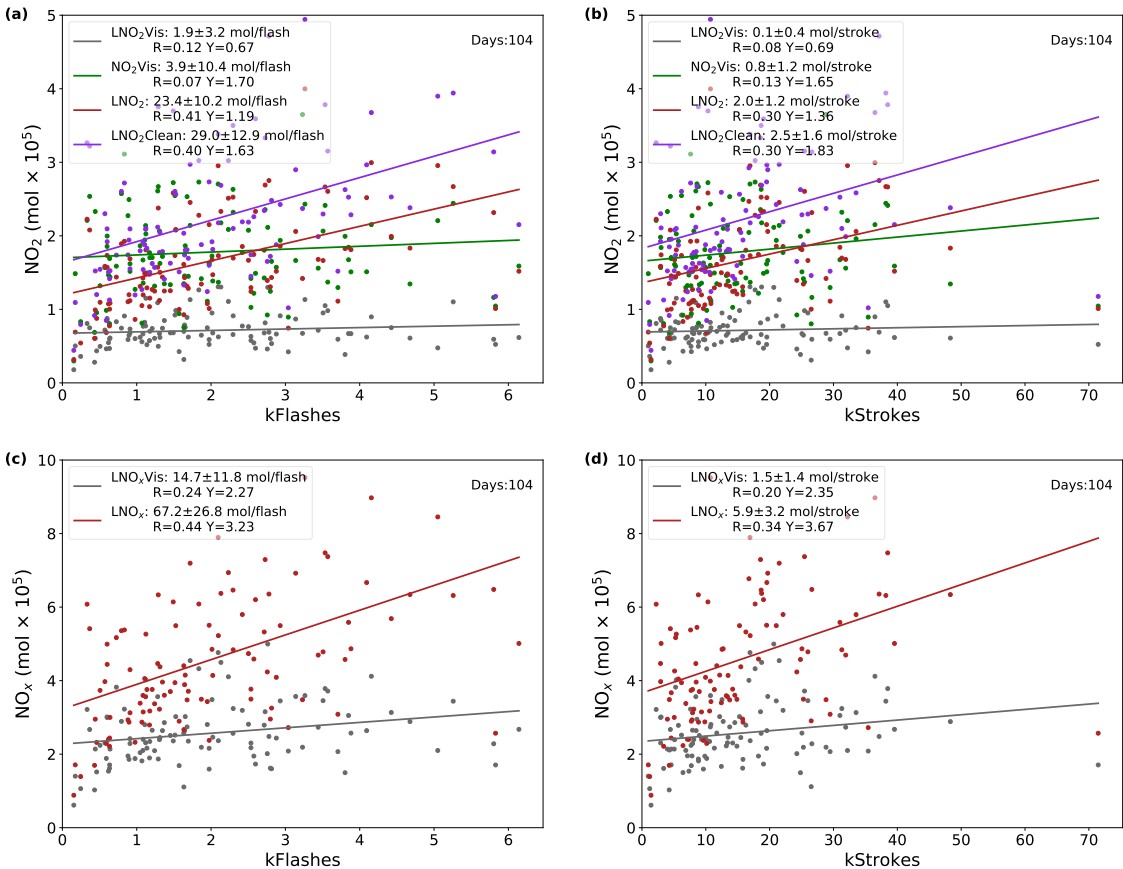

**Figure B1.** Linear regressions with CRF $\geq$ 90% and a flash threshold of 1 flash box$^{-1}$ or 3.4 strokes box$^{-1}$ per 2.4 h. (a) Daily NO$_2$Vis, LNO$_2$Vis, LNO$_2$ and LNO$_2$Clean versus ENTLN total flashes data. (b) Same as (a) but for strokes. (c) Daily LNO$_x$Vis and LNO$_x$ versus total flashes. (d) Same as (c) but for strokes.

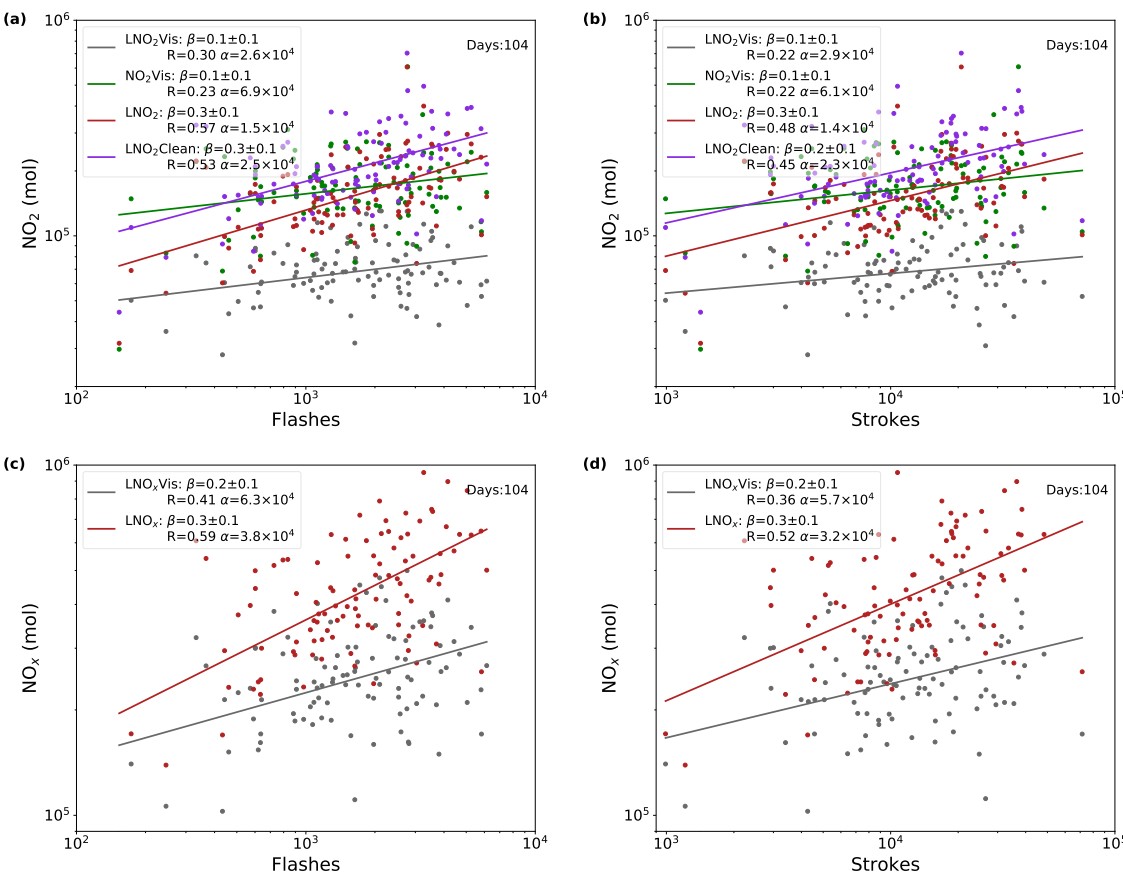

**Figure B2.** Same as Fig. B1 but using log-log axes.

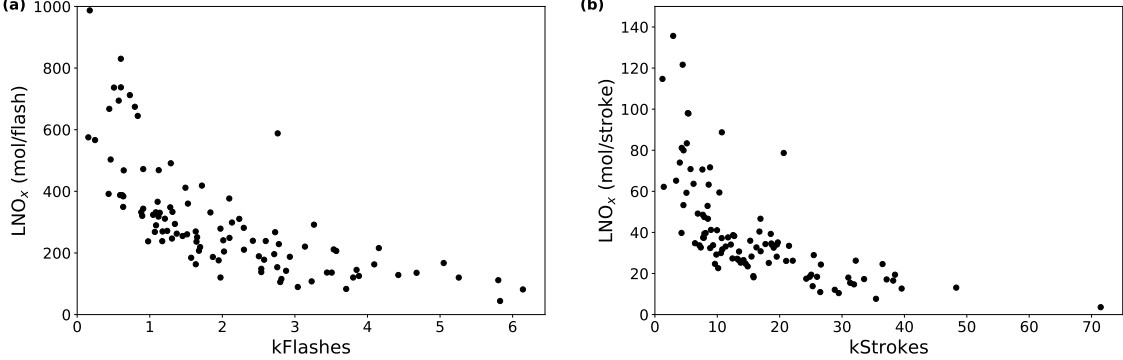

**Figure B3.** (a) Daily LNO$_x$ production efficiencies versus ENTLN total flashes data, with CRF $\geq$ 90% and a flash threshold of 1 flash box$^{-1}$. (b) Same as (a) but for strokes.