# Peer review of "Estimates of Lightning NOx Production based on High Resolution OMI NO2 Retrievals over the Continental US"

_Atmospheric Measurement Techniques, 2019_

## Referee Comment (RC1) · Anonymous Referee #1 · 4 Dec 2019

Xin Zhang et al. Estimates of LNOx Production based on high resolution OMI NO2 retrievals over the Continental US.

Overall Summary

Will be solid addition to the literature on LNOx Production

General Comments

A new satellite-based approach is used to estimate the production of NOx by lightning.

I believe the section describing the impact of differences in model profiles on the production efficiency to be of great use because it is novel.

When revising the manuscript, the authors should emphasize why they believe their approach is less sensitive to the tropospheric background than other approaches and why it may be more suitable over polluted locations.

They also need a more clear rationale for how they determined what detection efficiency to use for the ENTLN flashes.

Finally, there are a number of minor technical corrections that are needed for readability.

Specific Comments

L10: Put a positive spin or your method as opposed to a negative spin on others work, i.e. mention that your method reduces sensitivity to the background and includes much of the below-cloud NOx

L30: Why does an increase in lightning lead to a net positive forcing but a decrease in lightning as no net effect?

L63: "Since they assumed NO2 above the clouds are all NOx". I think this is misleading. Pickering et al. did not assume all NO2 above the clouds was LNOx. They assumed it included LNOx and BL NOx from pollution. They subtracted off a tropospheric background to account for the latter.

L68: The 3-30% value is an estimate of LNOx/(LNOx+NOx from other sources). It is not an estimate of the uncertainty. Move the 3-30% parentheses to after "background NOx".

L72: What do you mean by "weighted"?

L73: You need to specify whether the 15% adjustments were positive or negative also the "lofted pollution" was not corrected the "LNOx production" was corrected.

L86: You may want to add that "Pickering et al. then subtracted off an estimate of the tropospheric background to obtain the LNOx Production.

L93: The bias of AMF relative to what?

L106: Verify whether you used v3.0 or v3.1 of the NASA standard product

L110: The bias with respect to what is subtracted off?

L127: What region did Rudlosky analyze?

L121-131: Are you using ENTLN flashes or flashes from a combined ENTLN/NLDN data set?

L121-131: Are Rudlosky's value for CG, IC, or combined flashes?

L121-131: The DE values found by Rudlosky of 35% or so need to be reconciled with the 88%/45% values found by Lapierre.

L121-131: Why is the NLDN DE of IC pulses relevant?

L121-131: You may want to include a plot showing the mean ENTLN flashes during MJJA of 2014

L174: LNOx is regridded. At this point in the manuscript it is unclear how you generated pixel level values of LNOx that were regridded. Start paragraph by saying how you determined LNOx and then mention regridding method. Is it as simple as multiplying VLNOx by area?

L181: Why does choosing the 350-400 hPa layer avoid biases in the simulation of high clouds?

L193: I don't think "not polluted much" is appropriate for a 50% threshold. Simply state that more than half of the NOx must have an LNOx source

L216-220: Why do you think the production is relatively insensitive to the CRF threshold? Wouldn't you expect it to decrease as the CRF is increased due to less BL contamination?

L228: 10-80% range is so wide as to be useless. In addition to this range include the

mean and standard deviation

L266: Are the differences between NO2Vis production and LNO2Vis production larger over polluted grid boxes?

L271: Not sure what you mean by "trend". Please re-write this sentence more clearly.

L295-L230: State clearly why the difference is larger for LNOx (25%) than for LNO2 (11.5%)?

L300: I'd suggest deleting Figure 10. It is discussed in passing and the most important numbers from that Figure are already included in the text.

L334: Would the profile uncertainty be less than 13/26% for the LNO2Clean approach, although the latter (apparently) has more uncertainty due to the tropospheric background?

L335-338: Why don't uncertainties in the tropospheric background contribute to uncertainties in the production?

L335-338: How would biases in the modeled NO/NO2 ratio contribute to uncertainties in the production of LNOx?

L392: Expand on why this approach can be used in polluted regions? Is this different than other approaches?

L391-409: Minimize the use to lesser known acronyms in the conclusion section. Many readers will only read the abstract and conclusions. For e.g., TL and ratio and LNO2Clean

L401-402: The last sentence of the 2nd paragraph of the conclusion section is poorly written. Please re-write to make its meaning clear.

Figure 1 caption. Please specify what is plotted here. Also, did you choose this domain or is this a standard domain used by EPA or others (e.g., US1 or US2 domain)? How

many east-west and north-south grid points in domain

Table 1: Capitalize entln

Table 2: under different conditions > for different thresholds

Figure 3: May not be needed. Consider removing.

Technical Corrections

L5: based on the program of new > that uses the

L15: surface is > surface are

L16: originate largely > originates largely

L27: method and the commonly used the widely used cloud-top > method versus the commonly used cloud-top

L27: As lightning > While lightning

L28: have reported 5-16% increases > is predicted to increase by 5-16%

L41: qualified > quantified

L50: it is > this study

L50: without the contribution > and did not consider the contribution

L53: Replace sentence beginning with "The threshold of"> Their analysis was restricted to 30 x 60 km2 satellite pixels where the flash rate exceeded 1 flash km-2 hr-1.

L54: But the results of LNOx production are highly > But they found LNOx production to be highly

L72: removed by the weighted > removed by subtracting the weighted

L72: CRF criteria but has > CRF criteria required to be considered deep convection

but has

L76: at those locations > at some locations

L77: On another hand > On the other hand

L79: by strokes > using mean values of strokes

L80: NO2 can be "seen" > NO2 that can be "seen"

L94: we focused on > we focus on

L99: examines different sources of the uncertainty of > examines the effect of different sources of uncertainty on the

L128: and the detection > and found the detection

L128: are 88% > to be 88%

L130: corrected by 88% and 45% > divided by 0.88 and 0.45

L137: was used to generate > were used to generate

L170: part of the NO2 originated from lightning > part of the NO2 within the cloud originating from lightning

L180: convections > convection

L184: properties of cloud > cloud properties

L187: box-1 are chosen > box-1 per 2.4 hour time window are chosen

L191: sources except LNO2 > sources in addition to LNO2

L208: search of NO2 > search of the NO2

L206-210: When referring to combinations use the abbreviations given in Table 1.

L225: productions are > production values are

L275: 0.8 to 0.2 while the cloud is higher (smaller pressure value) > 0.8 to 0.2 as the cloud pressure decreases from 600 to 300 hPa. Note: Please correct my pressure ranges if incorrect.

L279: we can not derive that higher LNO2 production relates to higher clouds > we cannot derive the relationship between LNO2 production and cloud pressure

L284: larger than normal > larger in anomalous than normal

L292: chose the same method > chose the same value

L313: relative distribution > relative distributions

L314: is similar > are similar

L315: the unrealistic > an unrealistic

L319: carefully in our research > carefully into consideration

L321: are involved > are important

L401: find that the regionally > find a regionally

L403: pollutions than > pollution than

L409: level > scales

---

## Referee Comment (RC2) · Anonymous Referee #2 · 2 Jan 2020

This manuscript presents new methods for estimating the lightning NO2 and NOx production using OMI NO2 retrievals. The Berkeley high resolution OMI retrievals are used, which should provide better a priori NO2 profiles than use of the much coarser GMI model previous LNOx analyses. The new methods that are discussed should provide better accounting for background NOx; however, it is not made clear in the manuscript how this is achieved. This point needs to be addresses in a revision. There are numerous minor points which need attention, which are listed below.

Line 10: previous work does not neglect below-cloud LNO2. Full LNO2 and LNOx profiles extending to the surface are used from the GMI model is such work as Pickering

et al. (2016), Bucsela et al. (2019) and Allen et al. (2019).

line 15: ....surface are...

line 16: originates

lines 26-27: ...radiative forcing between when simulating future lightning using a new upward cloud ice flux (IFLUX) method and when the commonly used cloud-top height (CTH) approach is used.

line 28-29: 5 - 16% increases in what? 15% decrease in what?

line 30: ....radiative forcing due to ozone was found....

line 31: also need to compare with results of Romps et al. (2014)

line 39: Satellite...

line 48: convection.

line 71: ....based on a modification of the method.... Need to describe what is different.

line 75: Are you sure that it is the background subtraction causing the negatives? Many of the negatives resulted from the removal of the stratospheric slant column NO2 from the total NO2 slant column.

line 80: two parts of NO2 that can be....

line 81: ....above clouds (pixels with CRF >0.9) and....

line 88: ....contamination by anthropogenic....

lines 129-130: not sure why this phrase about NLDN is here when you are using ENTLN and not NLDN. Remove?

line 142: How is flash rate parameterized?

line 175: how is this comparison of pixels computed?

line 181: Does the Xu and Randall method consider subgrid convective clouds or only grid-scale cloud based on the microphysics parameterization that generates the grid-scale clouds? If it is the latter, this method is not appropriate as a criterion to evaluate model convection.

line 184: a time period......LNOx to be detected by

lines 184-188: Using these criteria will result in a low bias in the PE results. Bucsela et al. (2019) found that PE is larger at small flash rates. These small flash rates are being discarded here.

line 203: daily summations Is this what is done in Pickering et al. (2016)?

line 218:....(Table 2), as the CRF criterion increases from 70% to 90% and to 100%.

line 223:...to derive production per flash (production efficiency, PE).

line 228: cloud properties

line 229: PEs

line 259: ratio of CG to IC I don't think there was any particular assumption of this ratio in Pickering et al. (2016).

lines 263-264: this is not obvious from the Figure 5 plots

line 271: peaks of the LNO2 profile are...

line 275: from 0.8 to 0.2 as the cloud height increases....

line 278: LNO2 production (<30 mol/stroke) occurs

line 284: Is this necessarily true? LNOx production per flash may be smaller in high flash rate storms.

lines 287-288: ...  (postconvection) in which LNOx has already been redis-tributed.........uses LNOx production profiles.......(Allen et al., 2012; Luo et al., 2017)

line 296: 2X500 mol NO flash-1 This designation can be confusing. Some readers may think you mean 1000 mol/flash

line 308: different than that....

line 401-402: we find that the effect is regionally dependent. Both........NO2 cause different comprehensive effects due to nonlinear.....

Table 2: How many grid boxes per day typically qualify under your criteria?

Table 3: Need to indicate how the percentage difference was calculated in the caption. Which one was in the denominator?

Figures 7 and 8: What do the various colors of shading indicate? What are the scales for the horizontal plot at the top and the vertical plot on the side?

---

## Referee Comment (RC3) · Anonymous Referee #3 · 22 Jan 2020

Zhang et al. present estimates of lightning NOx production from satellite observations of NO2. They provide a formalism which considers both lightning and background NOx and discuss the impact of the latter. The study derives LNOx estimates plus uncertainties for the continental US. The paper is generally well structured and comprehensible. It matches the scope of AMT and should be published after dealing with the issues raised below.

The authors derive equations for LNOx by treating clouds as reflecting surfaces. This simplification is required for many radiative transfer models which can not handle multiple scattering in 3d clouds. Thus, several previous studies follow this approach. How-

[Figure]

ever, the authors should still be aware of this simplification and state this clearly in the manuscript. Formulations like "NO2 above/below the cloud" are misleading, as for thick thunderstorm clouds, most of the LNOx is WITHIN the cloud, with a high sensitivity from OMI at the cloud top, gradually decreasing towards the cloud bottom. In this context, the authors should discuss what the "cloud top" derived from OMI O2-O2 measurements acutally means for a thunderstorm cloud.

There is one crucial omission which might require major revisions: The derived AM-FLNOx will strongly depend on pcloud (e.g. eq. 2). But this key input parameter is missing in the error budget in table 3. This has to be revised and the uncertainties caused by pcloud have to be discussed accordingly.

Minor corrections:

5: Skip "program of"

130: must be square km

146: The concept of defining an "AMF" for converting SNO2 into LNOx was also used in Beirle et al., AMT 9, 1077-1094, 2009, (see eq. 9 therein).

268: I recommend to extend "Production" to "Production estimates"

395: "we find that the regionally dependent effect" - unclear, please revise.

Fig. 1: Please make the legends clearer and remove cryptic labels ("crf90_entin_tl")

Figs. 2 and 9: Please use less, but larger labels on x-axis.

---

## Author Comment (AC1) · 7 Feb 2020

**Estimates of Lightning NO$_x$ Production based on High Resolution OMI NO$_2$ Retrievals over the Continental US**

**Response to Anonymous Referee #1**

Xin Zhang, Yan Yin, Ronald van der A, Jeff L. Lapierre, Qian Chen,
Xiang Kuang, Shuqi Yan, Jinghua Chen, Chuan He and Rulin Shi

February 7, 2020

We thank the reviewer for his/her positive comments and very careful reading of the main article. The individual corrections suggested are addressed below. The reviewer's comments will be shown in **red**, our response in **blue**, and changes made to the paper are shown in **black** block quotes. Unless otherwise indicated, page and line numbers correspond to the original paper. Figures, tables, or equations referenced as "R$n$" are numbered within this response; if these are used in the changes to the paper, they will be replaced with the proper number in the final paper.

**General Comments**

1) When revising the manuscript, the authors should emphasize why they believe their approach is less sensitive to the tropospheric background than other approaches and why it may be more suitable over polluted locations.

The Section 3.2 compares the different $LNO_x$ productions based on different AMFs. The differences are partly caused by the different consideration of tropospheric background $NO_2$. To make it clearer, we add another section to discuss the improved background $NO_2$ consideration in our approach.

[revised manuscript text omitted]

[1] Locations are denoted in Fig. 6a.

**Specific Comments**

1) L10: Put a positive spin on your method as opposed to a negative spin on others work, i.e. mention that your method reduces sensitivity to the background and includes much of the below-cloud $NO_x$.

Thank you. We have reformatted the sentence:

> Focusing on the summer season during 2014, we find that the lightning $NO_2$ ($LNO_2$) PE is $44 \pm 16$ mol $NO_2$ flash$^{-1}$ and $8 \pm 3$ mol $NO_2$ stroke$^{-1}$ while $LNO_x$ PE is $120 \pm 52$ mol $NO_x$ flash$^{-1}$ and $22 \pm 9$ mol $NO_x$ stroke$^{-1}$. **Results reveal that our method reduces sensitivity to the background $NO_2$ and includes much of the below-cloud $LNO_2$.**

2) L30: Why does an increase in lightning lead to a net positive forcing but a decrease in lightning as no net effect?

Because the effect of CTH and IFLUX on radiative forcing from ozone and methane is different, the net effect should not be determined only by lightning. The detail of this is shown by Fig. 3 in Finney et al. (2018). To make this more clear, we have added the "radiative forcing" specifically.

> Finney et al. (2018) found different impacts on atmospheric composition and radiative forcing between when simulating future lightning using a new upward cloud ice flux (IFLUX) method and when the commonly used cloud-top height (CTH) approach is used. While lightning is predicted to increase by $5 - 16\%$ over the next century with the CTH approach (Clark et al., 2017; Banerjee et al., 2014; Krause et al., 2014), a 15% decrease in lightning was estimated with IFLUX in 2100 under a strong global warming scenario (Finney et al., 2018). **As a result of the different effects on radiative forcing from ozone and methane, a net positive radiative forcing was found with the CTH approach while there is little net radiative forcing with the IFLUX approach (Finney et al., 2018).**

3) L63: "Since they assumed $NO_2$ above the clouds are all $NO_x$". I think this is misleading. Pickering et al. (2016) did not assume all $NO_2$ above the clouds was $LNO_x$. They assumed it included $LNO_x$ and BL $NO_x$ from pollution. They subtracted off a tropospheric background to account for the latter.

Although the tropospheric background $NO_2$ is subtracted as shown by Eq. 1 in Pickering et al. (2016), they didn't apply it to the rest of analysis. As explained by the fifth paragraph in Sect. 3.4 and the fifth paragraph in Sect. 4.4 (Pickering et al., 2016): "Given the difficulty in estimating the background, we focus on areas of active or very recently active convection where the lower tropospheric background is minimized." and "In our calculations of $LNO_x$ in sections 4.1 through 4.3 we have assumed a zero tropospheric background due to uncertainty in how it should be determined.". To make it more clear, we have rewritten the sentence:

> To convert the $S_{LNO_2}$ to the tropospheric vertical density (VCD) of $LNO_x$ ($V_{LNO_x}$), an air mass factor (AMF) is calculated by dividing the a priori $S_{LNO_2}$ by the a priori $V_{LNO_x}$. **Since they considered $NO_2$ above the cloud as $LNO_2$ in the algorithm due to the difficulty and uncertainty in determining of the background $NO_2$, their AMF and derived VCD of $LNO_x$ ($LNO_2$) is named as $AMF_{LNO_x Clean}$ ($AMF_{LNO_2 Clean}$) and $LNO_x$Clean ($LNO_2$Clean), respectively.**

4) L68: The $3 - 30\%$ value is an estimate of $LNO_x/(LNO_x+NO_x$ from other sources). It is not an estimate of the uncertainty. Move the $3 - 30\%$ parentheses to after "background $NO_x$".

Moved.

Results for the Gulf of Mexico during 2007 – 2011 summer yield $LNO_x$ production of $80 \pm 45$ mol $NO_x$ per flash. **Among several substantial sources of uncertainty, significant uncertainty exists in characterizing background $NO_x$ (3% $\sim$ >30%) in this region (Pickering et al., 2016).**

5) L72: What do you mean by "weighted"?

"Weighted" means that box data are weighted according to the number of OMI pixels contributing to each. We have added the explanation of "weighted":

More recently Bucsela et al. (2019) obtained an average production efficiency (PE) of $180 \pm 100$ mol per flash over East Asia, Europe and North America based on the method used in Pickering et al. (2016). **The tropospheric $NO_x$ background was removed by subtracting temporal average of $NO_x$ at each box where the value was weighted by the number of OMI pixels which meet the optical cloud pressure and CRF criteria required to be considered deep convection but has 1 flash or less instead.**

6) L73: You need to specify whether the 15% adjustments were positive or negative also the "lofted pollution" was not corrected the "$LNO_x$ production" was corrected.

They considered 15% of total $NO_x$ is the transported pollution. Fixed.

**The lofted pollution was considered as 15% of total $NO_x$ according to the estimation from DeCaria et al. (2000, 2005) and the average chemical delay was adjusted by 15% following the 3-hour $LNO_x$ lifetime in the nearby field of convection (Nault et al., 2017).** However, there were negative $LNO_x$ values caused by the overestimation of the tropospheric background at those locations.

7) L86: You may want to add that "Pickering et al. then subtracted off an estimate of the tropospheric background to obtain the $LNO_x$ Production.

As mentioned before (*Response 3, Sect. Specific Comments*), they did not subtract off the background $NO_2$.

8) L93: The bias of AMF relative to what?

The bias of AMF is related to aircraft observations. Fixed.

Meanwhile, Laughner and Cohen (2017) showed that the OMI AMF is increased by $\sim$35% for summertime when $LNO_2$ simulated by WRF-Chem is included in the a priori profiles to match aircraft observations. **The simulation agrees with observed $NO_2$ profiles and the bias of AMF related to these observations is reduced to $< \pm 4\%$ for OMI viewing geometries.**

9) L106: Verify whether you used v3.0 or v3.1 of the NASA standard product

Verified.

For the current study, we used the NASA standard product **V3.0** (Krotkov et al., 2017) as input to the $LNO_x$ retrieval algorithm.

10) L110: The bias with respect to what is subtracted off?

The bias is caused by an instrument artifact. Corrected.

A corrected ("de-striped") SCD is obtained by subtracting **the cross-track bias caused by an instrument artifact** from the measured slant column;

11) L127: What region did Rudlosky analyze? Are Rudlosky's value for CG, IC, or combined flashes? Are you using ENTLN flashes or flashes from a combined ENTLN/NLDN data set? The DE values found by Rudlosky of 35% or so need to be reconciled with the 88%/45% values found by Lapierre.

Thanks for your comments on lightning data. We have changed the DE values to focus on CONUS and specified the kind of ENTLN data.

Rudlosky (2015) compared ENTLN **combined events (IC and CG) with LIS flashes** and found that the relative flash detection efficiency of ENTLN **over CONUS increases from 62.4% during 2011 to 79.7% during 2013**.

12) L121-131: Why is the NLDN DE of IC pulses relevant?

Because we only used ENTLN data instead of the combined ENTLN and the NLDN dataset, it is necessary to clarify that the NLDN DE of IC pulses is lower than ENTLN's and ENTLN DE of CG flashes or strokes is high enough to keep unchanged.

Lapierre et al. (2019) also compared combined ENTLN and the NLDN dataset with data from the LIS during 2014 and the detection efficiencies of IC flashes and strokes are 88% and 45%, respectively. **Since we only use the ENTLN data in 2014 as Lapierre et al. (2019) and NLDN detection efficiency of IC pulses should be lower than 33% which is calculated by the data in 2016 (Zhu et al., 2016), only the IC flashes and strokes are corrected by 88% and 45%, respectively**, while CG flashes and strokes are unchanged because of the high detection efficiency.

13) L121-131: You may want to include a plot showing the mean ENTLN flashes during MJJA of 2014.

Added (Fig. R2).

In addition, lightning flash rate based on the Price and Rind level of neutral buoyancy parameterization (Price and Rind, 1992) and $LNO_x$ parameterizations are activated (200 mol NO flash$^{-1}$, the factor to adjust the predicted number of flashes is set to 1; hereinafter referred to as "1×200 mol NO flash$^{-1}$"). **Although the simulated total flash densities are higher in the Southeast US and lower in the North Central US (Fig. 2), the criteria in Sect. 3.1 could limit this effect on the estimation of $LNO_x$ production and Sect. 3.4 will use another simulation to test this problem.**

[Figure]

**Figure R2 .** Comparison between total flash densities from ENTLN and WRF-Chem during MJJA 2014.

14) L174: LNO$_x$ is regridded. At this point in the manuscript it is unclear how you generated pixel level values of LNO$_x$ that were regridded. Start paragraph by saying how you determined LNO$_x$ and then mention regridding method. Is it as simple as multiplying VLNOx by area?

Sorry for the misleading. This paragraph describes how to re-grid the $V_{\mathrm{LNO}_x}$ pixel values obtained in Section 2.4 to $0.05° \times 0.05°$ grids.

> **$V_{\mathbf{LNO}_x}$ is re-gridded to $0.05° \times 0.05°$ grids using the constant value method (Kuhlmann et al., 2014)**. Then, it is analyzed in $1° \times 1°$ grid boxes with a minimum of fifty valid $0.05° \times 0.05°$ grids which is equivalent to five satellite pixels in Pickering et al. (2016).

15) L181: Why does choosing the 350-400 hPa layer avoid biases in the simulation of high clouds?

If the model has biases in the 3D cloud fraction in the upper troposphere (higher than 350 hPa), our method is not sensitive to that bias since we don't consider the cloud fraction at levels above 350 hPa. This pressure layer was also used in Strode et al. (2017). We have added the reference for the method.

> The CFs are defined as the maximum cloud fraction calculated by the Xu-Randall method between 350 and 400 hPa **(Xu and Randall, 1996; Strode et al., 2017)**. This atmospheric layer (between 350 and 400 hPa) avoids any biases in the simulation of high clouds. We choose CFs $\geq 40\%$ suggested by Strode et al. (2017) to determine cloudy or clear for each simulation grid.

16) L193: I don't think "not polluted much" is appropriate for a 50% threshold. Simply state that more than half of the NO$_x$ must have an LNO$_x$ source.

Fixed.

> The ratio $\geq 50\%$ indicates that more than half of the NO$_x$ above the cloud **must have a LNO$_x$ source**.

17) L216-220: Why do you think the production is relatively insensitive to the CRF threshold? Wouldn't you expect it to decrease as the CRF is increased due to less BL contamination?

Yes, it is true for the method of Pickering et al. (2016) and Lapierre et al. (2019). But, in our new method, the BL contamination is considered in the numerator of AMF$_{\mathrm{LNO}_x}$ (Eq. 2). As a result, when the CRF is increased from 70% to 90%, the LNO$_x$ PE rises from 35.7 LNO$_x$/flash to 54.5 LNO$_x$/flash, which is opposite compared to the result of Pickering et al. (2016). On the other hand, the updraft would also affect the BL contamination detected by OMI. We have added these information to make the choice of CRF more reasonable:

> **The increment of LNO$_x$ PE caused by the CRF increase from 70% to 90% is opposite to the result of Pickering et al. (2016). This is an effect of the consideration of NO$_2$ contamination transported from the boundary layer in our method.** Although enhanced NO$_x$ is often observed in regions with CRF > 70% (Pickering et al., 2016), the following analysis will be based on the criterion of CRF $\geq 90\%$ considering the contamination by low and mid-level NO$_2$ and comparisons with the results of Pickering et al. (2016) and Lapierre et al. (2019).

18) L228: 10-80% range is so wide as to be useless. In addition to this range include the mean and standard deviation.

Thank you for your remind. We have added the mean and standard deviation of the ratio of LNO$_x$ below the cloud.

> The simulation of GMI in Pickering et al. (2016) indicated that $25\% - 30\%$ of the LNO$_x$ column lies below the CP, while the ratio in our WRF-Chem simulation is **$56 \pm 20\%$**.

19) L266: Are the differences between $NO_2$Vis production and $LNO_2$Vis production larger over polluted grid boxes?

Yes. We mentioned that in L265.

20) L271: Not sure what you mean by "trend". Please re-write this sentence more clearly.

Clarified.

> ... the ratio of $LNO_2$Vis to $LNO_2$ ranges from 10% – 80%. This may be caused by the height of the clouds and the profile of $LNO_2$. If the CP is near 300 hPa, the ratio should be smaller because of the coverage of clouds. **The ratio would also be smaller** while peaks of the $LNO_2$ profile are below the CP.

21) L295-L300: State clearly why the difference is larger for $LNO_x$ (25%) than for $LNO_2$ (11.5%)?

As suggested by Eric J. Bucsela in private, we have redone the linear regressions and stated the results instead of comparing the difference in $LNO_2$ PE and that in $LNO_x$ PE.

> For the linear regression method (Fig. 9), $LNO_2$ production is $29.8 \pm 20.5$ mol/flash which is 59.4% larger than the basic one ($18.7 \pm 18.1$ mol/flash). Meanwhile, $LNO_x$ production (increasing from $54.5 \pm 48.1$ mol/flash to $88.5 \pm 61.1$ mol/flash) **also depends on** the configuration of LNO production in WRF-Chem.

22) L300: I'd suggest deleting Figure 10. It is discussed in passing and the most important numbers from that Figure are already included in the text.

We agreed with you and have deleted that figure.

23) L334: Would the profile uncertainty be less than 13/26% for the $LNO_2$Clean approach, although the latter (apparently) has more uncertainty due to the tropospheric background?

We have checked the profile uncertainty of retrieved $LNO_2$ and $LNO_x$ productions for the $LNO_2$Clean approach to be $\sim 0\%$ and 39% respectively. As a result, the $LNO_2$Clean approach is suitable for clean regions and the effect of profile need to be considered carefully.

24) L335-338: Why don't uncertainties in the tropospheric background contribute to uncertainties in the production?

Yes, the tropospheric background $NO_2$ should affect the uncertainties in the estimation of production. We have added this to the section of uncertainty.

> In addition, the estimation of $LNO_x$ PE is also dependent on the tropospheric background $NO_2$. In our method, main factors affecting this factor are the emissions inventory and the amount of transported $NO_2$. For the emissions inventory, the sources of uncertainty are assumptions, methods, input data and calculation errors. As a result, the uncertainties for different species or pollutants related to $NO_2$ are different and EPA also doesn't publish the quantified uncertainty measures because the parties that submit emissions estimates to EPA are not asked to include quantitative uncertainty measurements or estimates (EPA, 2015). For the simulated convective transport, Li et al. (2018) compared the cloud-resolving simulations with

these based on convective parameterization and pointed out that the convective transport was weaker in the parameterization. But, we believe that the ratio condition ($LNO_2 Vis/NO_2 Vis \geq 50\%$) should reduce these two kinds of uncertainty and assume an uncertainty of 10%, which is less than 20% assigned in Allen et al. (2019) and Bucsela et al. (2019).

25) L335-338: How would biases in the modeled $NO/NO_2$ ratio contribute to uncertainties in the production of $LNO_x$?

We have added that kind of uncertainty according to Travis et al. (2016), Silvern et al. (2018) and Bucsela et al. (2019). The uncertainty in the production of $LNO_x$ is 20%.

Recent works revealed that the modeled $NO/NO_2$ ratio departs from the data in the SEAC[4]RS aircraft campaign (Travis et al., 2016; Silvern et al., 2018). Silvern et al. (2018) attributed this to the positive interference on the $NO_2$ measurements or errors in the cold-temperature $NO-NO_2-O_3$ photochemical reaction rate. We assign an uncertainty of 20% to this error considering the possible positive $NO_2$ measurements interferences (Allen et al., 2019; Bucsela et al., 2019).

26) L392: Expand on why this approach can be used in polluted regions? Is this different than other approaches?

Added.

In this study, a new algorithm for retrieving $LNO_2$ ($LNO_x$) from OMI, including $LNO_2$ ($LNO_x$) below cloud, has been developed for application over active convection. **It works in both clean and polluted regions because of the consideration of tropospheric background pollution in the definition of AMFs.**

27) L391-409: Minimize the use to lesser known acronyms in the conclusion section. Many readers will only read the abstract and conclusions. For e.g., TL and ratio and LNO2Clean.

We reformatted the section, thanks.

Compared with former methods, our method has reduced the sensitive to background $NO_2$, while the method in Lapierre et al. (2019) underestimates $LNO_x$ production efficiency because of the neglected below-cloud $LNO_2$ and $LNO_2$ production is overestimated using the method in Pickering et al. (2016) due to the over-cloud background $NO_2$ in polluted regions. Finally, implementing profiles generated with different model settings of lightning ($1\times200$ mol NO flash$^{-1}$ and $2\times500$ mol NO flash$^{-1}$), we find that the larger LNO production model setting leads to larger retrieval of $LNO_x$ despite some regionally dependent effects caused by nonlinear calculation of AMF. Both the ratio of the tropospheric $LNO_2$ above the cloud to the total tropospheric $LNO_2$ and the ratio of $LNO_2$ to $NO_2$ cause different comprehensive effects due to the nonlinear calculation of $AMF_{LNO_2}$ and $AMF_{LNO_x}$.

28) L401-402: The last sentence of the 2nd paragraph of the conclusion section is poorly written. Please re-write to make its meaning clear.

Rewritten as the response above.

29) Figure 1 caption. Please specify what is plotted here. Also, did you choose this domain or is this a standard domain used by EPA or others (e.g., US1 or US2 domain)? How many east-west and north-south grid points in domain?

We chose the simulation domain and the information has been added to the caption of Figure 1 (Fig. R3).

[Figure]

**Figure R3 .** Domain and terrain height (m) of the WRF-Chem simulation with 350 x 290 grid cells and a horizontal resolution of 12 km.

30) Table 1: Capitalize entln

Fixed (Table R2).

31) Table 2: under different conditions > for different thresholds

Fixed (Table R3).

32) Figure 3: May not be needed. Consider removing.

Figure 3 shows the time series of $NO_2$Vis, $LNO_2$Vis, $LNO_2$ and $LNO_2$Clean production per day and compares those with the percent differences between $NO_2$Vis and $LNO_2$Vis and the percent differences between $NO_2$Vis and $LNO_2$Clean. Readers could distinguish the difference between different methods in one figure. For example, these peaks of $LNO_2$Clean coincide with the $NO_2$ pollution above the cloud (the peaks of the percent differences between $NO_2$Vis and $LNO_2$Vis). As a result, we decide to keep this figure.

**Table R2 .** Definitions of the abbreviations for the criteria used in this study.

| Abbreviations | Full form [source] |
|---|---|
| CRF | Cloud radiance fraction [OMI] |
| CP | Cloud optical pressure [OMI] |
| CF | Cloud fraction [WRF-Chem] |
| TL | Total lightning flashes [WRF-Chem] |
| ratio | modeled $LNO_2$ Vis / modeled $NO_2$ Vis [WRF-Chem] |
| CRF$\alpha$_ENTLN | CRF $\geq \alpha$ + ENTLN flashes(strokes) $\geq$ 2400(8160) [ENTLN] |
| CRF$\alpha$_CF40_ENTLN | CRF $\geq \alpha$ + ENTLN flashes(strokes) $\geq$ 2400(8160) + CF $\geq$ 40% |
| CRF$\alpha$_ENTLN_TL1000 | CRF $\geq \alpha$ + ENTLN flashes(strokes) $\geq$ 2400(8160) + TL $\geq$ 1000 |
| CRF$\alpha$_CF40_ENTLN_TL1000 | CRF $\geq \alpha$ + ENTLN flashes(strokes) $\geq$ 2400(8160) + CF $\geq$ 40% + TL $\geq$ 1000 |
| CRF$\alpha$_ENTLN_TL1000_ratio50 | CRF $\geq \alpha$ + ENTLN flashes(strokes) $\geq$ 2400(8160) + TL $\geq$ 1000 + ratio $\geq$ 50% |
| CRF$\alpha$_CF40_ENTLN_TL1000_ratio50 | CRF $\geq \alpha$ + ENTLN flashes(strokes) $\geq$ 2400(8160) + CF $\geq$ 40% + TL $\geq$ 1000 + ratio $\geq$ 50% |

$\alpha$ has three options: 70%, 90% and 100%

**Table R3 .** $LNO_x$ production for different thresholds of CRF with coincident ENTLN data, TL $\geq$ 1000 and ratio $\geq$ 50%.

| CRF (%) | ENTLN data type[1] | $LNO_x$/flash or $LNO_x$/stroke | R value | Intercept ($10^5$ mol) | Days[2] |
|---|---|---|---|---|---|
| 70 | Flash | $35.7 \pm 36.8$ | 0.21 | 4.91 | 85 |
| 90 | Flash | $54.5 \pm 48.1$ | 0.25 | 3.90 | 81 |
| 100 | Flash | $20.8 \pm 37.4$ | 0.13 | 5.67 | 71 |
| 70 | Stroke | $4.1 \pm 3.9$ | 0.21 | 5.16 | 96 |
| 90 | Stroke | $7.0 \pm 4.8$ | 0.29 | 4.16 | 93 |
| 100 | Stroke | $2.6 \pm 4.0$ | 0.14 | 5.41 | 82 |

[1] The threshold of ENTLN data is 2400 flashes box$^{-1}$ and 8160 strokes box$^{-1}$ during the period of 2.4 h before OMI overpass time. [2] The number of valid days with specific criteria in MJJA 2014.

**Technical Corrections**

1) L5: based on the program of new > that uses the

Fixed, thanks.

> To apply satellite data in both clean and polluted regions, a new algorithm for calculating $LNO_x$ has been developed **that uses the** Berkeley High Resolution (BEHR) v3.0B $NO_2$ product and the Weather Research and Forecasting-Chemistry (WRF-Chem) model.

2) L15: surface is > surface are
L16: originate largely > originates largely

Fixed.

> Nitrogen oxides ($NO_x$) near the Earth's surface **are** mainly produced by soil, biomass burning and fossil fuel combustion, while $NO_x$ in the middle and upper troposphere **originates** largely from lightning and aircraft emissions.

3) L27: method and the commonly used the widely used cloud-top > method versus the commonly used cloud-top
L27: As lightning > While lightning
L28: have reported 5 – 16% increases > is predicted to increase by 5 – 16%

Fixed.

> Finney et al. (2018) found different impacts on atmospheric composition and radiative forcing when simulating future lightning using a new upward cloud ice flux (IFLUX) **method versus the commonly used** cloud-top height (CTH) approach. **While** lightning **is predicted to increase by 5 — 16%** over the next century with the CTH approach (Clark et al., 2017; Banerjee et al., 2014; Krause et al., 2014), a 15% decrease in lightning was estimated with IFLUX in 2100 under a strong global warming scenario (Finney et al., 2018).

4) L41: qualified > quantified

Fixed.

> Recent studies have determined and **quantified** $LNO_x$ using satellite observations.

5) L50: it is > this study
L50: without the contribution > and did not consider the contribution

Fixed.

> However, **this study** assumed that all the enhanced $NO_2$ originated from lightning and **did not consider the contribution** of anthropogenic emissions.

6) L53: Replace sentence beginning with "The threshold of" > Their analysis was restricted to $30\times60$ km$^2$ satellite pixels where the flash rate exceeded 1 flash km$^{-2}$ hr$^{-1}$.
L54: But the results of $LNO_x$ production are highly > But they found $LNO_x$ production to be highly

Fixed.

**Their analysis was restricted to $30 \times 60$ km$^2$ satellite pixels where the flash rate exceeded 1 flash km$^{-2}$ hr$^{-1}$. But they found LNO$_x$ production to be highly** variable and correlations between flash rate densities and LNO$_x$ production are low in some cases.

7) L72: removed by the weighted > removed by subtracting the weighted
L72: CRF criteria but has > CRF criteria required to be considered deep convection but has

Fixed.

The tropospheric NO$_x$ background was removed by **subtracting** temporal average of NO$_x$ at each box where the value was weighted by the number of OMI pixels which meet the optical cloud pressure and CRF criteria **required to be considered deep convection** but has 1 flash or less instead.

8) L76: at those locations > at some locations

Fixed.

However, there were negative LNO$_x$ values caused by the overestimation of the tropospheric background at **some** locations.

9) L77: On another hand > On the other hand
L79: by strokes > using mean values of strokes

Fixed.

**On the other hand**, Lapierre et al. (2019) constrained LNO$_2$ to $1.1 \pm 0.6$ mol NO$_2$/stroke for intracloud (IC) strokes and $10.0 \pm 4.9$ mol NO$_2$/stroke for cloud-to-ground (CG) strokes over the continental US (CONUS). LNO$_2$ per stroke was scaled to 54.4 mol NO$_x$/flash **using mean values of strokes** per flash and the ratio of NO to NO$_2$ in the UT.

10) L80: NO$_2$ can be "seen" > NO$_2$ that can be "seen"

Fixed.

They used the regridded Berkeley High-Resolution (BEHR) V3.0A $0.05° \times 0.05°$ "visible only" NO$_2$ VCD ($V_{vis}$) product which includes two parts of NO$_2$ **that** can be "seen" by the satellite.

11) L94: we focused on > we focus on

Fixed.

In this paper, we **focus** on the estimation of LNO$_2$ production per flash (LNO$_2$/flash) ...

12) L99: examines different sources of the uncertainty of > examines the effect of different sources of uncertainty on the

Fixed.

Section 4 examines **the effect of** different sources of the uncertainty **on** the results.

13) L128: and the detection > and found the detection
L128: are 88% > to be 88%
L130: corrected by 88% and 45% > divided by 0.88 and 0.45

Fixed.

Lapierre et al. (2019) also compared combined ENTLN and the NLDN dataset with data from the LIS during 2014 and **found** the detection efficiencies of IC flashes and strokes **to be** 88% and 45%, respectively. Since we only use the ENTLN data in 2014 as Lapierre et al. (2019) and NLDN detection efficiency of IC pulses should be lower than 33% which is calculated by the data in 2016 (Zhu et al., 2016), only the IC flashes and strokes are **divided by 0.88 and 0.45**, respectively, while CG flashes and strokes are unchanged because of the high detection efficiency.

14) L137: was used to generate > were used to generate

Fixed.

Outputs from the version 4 of Model for Ozone and Related chemical Tracers (MOZART-4; Emmons et al., 2010) **were** used to generate the initial and boundary conditions of chemical species.

15) L170: part of the $NO_2$ originated from lightning > part of the $NO_2$ within the cloud originating from lightning

Fixed.

The sensitivity study of Beirle et al. (2009) compared the chemical compositions from the cloud bottom to the cloud top and revealed that a significant fraction the $NO_2$ within the cloud **originating** from lightning can be detected by the satellite.

16) L180: convections > convection

Fixed.

Furthermore, another criterion of cloud fractions (CFs) is applied to the WRF-Chem results for the successful simulation of **convection**.

17) L184: properties of cloud > cloud properties

Fixed.

Besides **cloud properties**, the time period and sufficient flashes (or strokes) are required for fresh $LNO_x$ detected by OMI.

18) L187: $box^{-1}$ are chosen > $box^{-1}$ per 2.4 hour time window are chosen

Fixed.

Meanwhile, 2400 flashes $box^{-1}$ and 8160 strokes $box^{-1}$ **per 2.4 hour time window** are chosen as sufficient for detecting $LNO_x$ (Lapierre et al., 2019).

19) L191: sources except $LNO_2$ > sources in addition to $LNO_2$

Fixed.

In view of other $NO_2$ sources **in addition to** $LNO_2$, the ratio of modeled lightning $NO_2$ above cloud ($LNO_2$Vis) to modeled $NO_2$ above cloud ($NO_2$Vis) is defined to check whether enough $LNO_2$ can be detected by OMI.

20) L208: search of NO$_2$ > search of the NO$_2$
L206-210: When referring to combinations use the abbreviations given in Table 1.

Rewritten.

A daily search of **the** NO$_2$ product for coincident ENTLN flash (stroke) data results in 99 (102) valid days under the CRF90_ENTLN condition. Taking the flashes type ENTLN data as an example, the number of valid days decreases from 99 to 81 **under the CRF90_ENTLN_TL1000_ratio50 condition**, while LNO$_x$/flash increases from 86.0 ± 14.0 mol/flash to 114.8 ± 18.2 mol/flash. The result is almost the same as that under **the CRF90_ENTLN_TL1000 condition which is without the condition of ratio ≥ 50%**.

21) L225: productions are > production values are

Fixed.

LNO$_2$ **production values** are mostly in the range from 20 to 80 mol/flash.

22) L275: 0.8 to 0.2 while the cloud is higher (smaller pressure value) > 0.8 to 0.2 as the cloud pressure decreases from 600 to 300 hPa. Note: Please correct my pressure ranges if incorrect.

Fixed, thanks. The range of cloud pressure is right.

Since the ratio of LNO$_2$Vis to LNO$_2$ decreases from 0.8 to 0.2 **as the cloud pressure decreases from 600 to 300 hPa** ...

23) L279: we can not derive that higher LNO$_2$ production relates to higher clouds > we cannot derive the relationship between LNO$_2$ production and cloud pressure

Fixed.

Because of the limited amount of large LNO$_2$ production and lightning data, we cannot derive **the relationship between LNO$_2$ production and cloud pressure** or different lightning properties at this stage.

24) L284: larger than normal > larger in anomalous than normal

Fixed.

Because updrafts are stronger and flash rates are higher in anomalous storms, UT LNO$_x$ concentrations is **larger in anomalous than normal** polarity storms.

25) L292: chose the same method > chose the same value

Fixed.

Zhao et al. (2009) set a NO$_x$ production rate of 250 mol NO per flash in a regional-scale model, while Bela et al. (2016) chose the **same value** (330 mol NO per flash) that was used by Barth et al. (2012).

26) L313: relative distribution > relative distributions
L314: is similar > are similar
L315: the unrealistic > an unrealistic

Fixed.

Although the **relative distributions** of mean LNO and $LNO_2$ profiles **are** similar in both regions, the magnitude differs with a factor of 10. This phenomenon implies that the performance of lightning parameterization in WRF-Chem is region dependent and **an** unrealistic profile could appear in the UT.

27) L319: carefully in our research > carefully into consideration

Fixed.

However, the relative distribution of $LNO_2$ within the UT should be **taken carefully into consideration**.

28) L321: are involved > are important

Fixed.

When the condition of high $LNO_2/NO_2$ is not met, both relative distribution and ratio are **important**.

29) L401: find that the regionally > find a regionally

Rewritten.

Finally, implementing profiles generated with different model settings of lightning ($1\times200$ mol NO flash$^{-1}$ and $2\times500$ mol NO flash$^{-1}$), we find that the larger LNO production in model leads to larger retrieval of $LNO_x$ despite some regionally dependent effects.

30) L403: pollutions than > pollution than

Fixed.

Since other regions, like China and India, have much more $NO_2$ **pollution** than the CONUS, it is necessary to consider the background $NO_2$ in detail.

31) L409: level > scales

Fixed.

[revised manuscript text omitted]

February 7, 2020

We thank the reviewer for his/her positive comments and very careful reading of the main article. The individual corrections suggested are addressed below. The reviewer's comments will be shown in **red**, our response in **blue**, and changes made to the paper are shown in **black** block quotes. Unless otherwise indicated, page and line numbers correspond to the original paper. Figures, tables, or equations referenced as "R$n$" are numbered within this response; if these are used in the changes to the paper, they will be replaced with the proper number in the final paper.

**Major Comments**

The new methods that are discussed should provide better accounting for background NOx; however, it is not made clear in the manuscript how this is achieved. This point needs to be addresses in a revision.

Thank you for your advice. This is the same comment mentioned by Reviewer#1 (See *No. 1, Sect. General Comments*). We have made it clearer in a new section (Sect. 3.3).

**Minor Comments**

1) Line 10: previous work does not neglect below-cloud LNO$_2$. Full LNO$_2$ and LNO$_x$ profiles extending to the surface are used from the GMI model is such work as Pickering et al. (2016); Bucsela et al. (2019); Allen et al. (2019).

Thank you for your correction. Instead of the negative spin on others' work, we have reformatted the sentence to put a positive spin on our method, as mentioned in the Response to Reviewer#1 (See *No. 1, Sect. Specific Comments*). Because Lapierre et al. (2019) neglected the below-cloud LNO$_2$, it is still necessary to mention that in the summarization of our results.

> Focusing on the summer season during 2014, we find that the lightning NO$_2$ (LNO$_2$) PE is 44 $\pm$ 16 mol NO$_2$ flash$^{-1}$ and 8 $\pm$ 3 mol NO$_2$ stroke$^{-1}$ while LNO$_x$ PE is 120 $\pm$ 52 mol NO$_x$ flash$^{-1}$ and 22 $\pm$ 9 mol NO$_x$ stroke$^{-1}$. **Results reveal that our method reduces sensitivity to the background NO$_2$ and includes much of the below-cloud LNO$_2$.**

2) line 15: ....surface are...
line 16: originates

Corrected.

> Nitrogen oxides (NO$_x$) near the Earth's surface **are** mainly produced by soil, biomass burning and fossil fuel combustion, while NO$_x$ in the middle and upper troposphere **originates** largely from lightning and aircraft emissions.

3) lines 26-27: ...radiative forcing between when simulating future lightning using a new upward cloud ice flux (IFLUX) method and when the commonly used cloud-top height (CTH) approach is used.

Corrected.

> Finney et al. (2018) found different impacts on atmospheric composition and radiative forcing when simulating future lightning using a new upward cloud ice flux (IFLUX) **method versus the commonly used** cloud-top height (CTH) approach.

4) line 28-29: 5 – 16% increases in what? 15% decrease in what?

These percentage is relative to the current lightning. Corrected.

> While **lightning** is predicted to increase by 5 — 16% over the next century with the CTH approach (Clark et al., 2017; Banerjee et al., 2014; Krause et al., 2014), a 15% **decrease in lightning** was estimated with IFLUX in 2100 under a strong global warming scenario (Finney et al., 2018).

5) line 30: ....radiative forcing due to ozone was found.... line 31: also need to compare with results of Romps et al. (2014)

We have reformatted the sentence and added the discussion of Romps et al. (2014) and Romps (2019).

> As a result of the different effects on radiative forcing **from ozone and methane**, a net positive radiative forcing was found with the CTH approach while there is little net radiative forcing with the IFLUX approach (Finney et al., 2018). **However, the convective available potential energy (CAPE) times the precipitation rate (P) proxy predicts a 12 $\pm$ 5% increase in the Continental US (CONUS) lightning strike rate per kelvin of global warming (Romps et al., 2014), while the IFLUX proxy predicts the**

**lightning will only increase 3.4%/K over the CONUS. Recently, Romps (2019) compared the CAPE × P proxy and IFLUX method in cloud-resolving models. They report that higher CAPE and updraft velocities caused by global warming could lead to the large increases in tropical lightning simulated by CAPE × P proxy, while IFLUX proxy predicts little change in tropical lightning because of the small changes in water mass fluxes.**

6) line 39: Satellite...

Corrected.

**Satellite** measurements of $NO_2$ are a powerful tool compared to conventional platforms, because of its global coverage, constant instrument features and temporal continuity.

7) line 48: convection.

Corrected.

As these methods focus on monthly or yearly mean $NO_2$ column densities, more recent studies applied specific approaches to investigate $LNO_x$ directly over active **convection**.

8) line 71: ....based on a modification of the method.... Need to describe what is different.

Thank you for your advice. We have added more descriptions about the method of Bucsela et al. (2019).

More recently Bucsela et al. (2019) obtained an average production efficiency (PE) of $180 \pm 100$ mol per flash over East Asia, Europe and North America based on **a modification of** the method used in Pickering et al. (2016). **A power function between $LNO_x$ and lightning flash rate was established, while the minimum flash-rate threshold was not applied.**

9) line 75: Are you sure that it is the background subtraction causing the negatives? Many of the negatives resulted from the removal of the stratospheric slant column $NO_2$ from the total $NO_2$ slant column.

Sorry for the incomplete consideration. We have added the effect of overestimated stratospheric slant column $NO_2$ to the sentence.

However, there were negative $LNO_x$ values caused by the overestimation of the tropospheric background **and stratospheric $NO_2$** at some locations.

10) line 80: two parts of $NO_2$ that can be....

Fixed.

They used the regridded Berkeley High-Resolution (BEHR) V3.0A $0.05° \times 0.05°$ "visible only" $NO_2$ VCD ($V_{vis}$) product which includes two parts of $NO_2$ **that** can be "seen" by the satellite.

11) line 81: ....above clouds (pixels with CRF > 0.9) and....

Fixed.

The first part is the $NO_2$ above clouds **(pixels with CRF > 0.9)** and the second part is the $NO_2$ detected from cloud free areas.

12) line 88: ....contamination by anthropogenic....

Corrected.

> To apply the approach used by Bucsela et al. (2010), Pickering et al. (2016), Bucsela et al. (2019) and Lapierre et al. (2019) without geographic restrictions, contamination **by** anthropogenic emissions must be taken into account in detail.

13) lines 129-130: not sure why this phrase about NLDN is here when you are using ENTLN and not NLDN. Remove?

Because we only used ENTLN data instead of the combined ENTLN and NLDN dataset, it is necessary to clarify that the NLDN DE of IC pulses is lower than ENTLN's and ENTLN DE of CG flashes or strokes is high enough to keep unchanged.

> Lapierre et al. (2019) also compared combined ENTLN and the NLDN dataset with data from the LIS during 2014 and found the detection efficiencies of IC flashes and strokes to be 88% and 45%, respectively. **Since we only use the ENTLN data in 2014 as Lapierre et al. (2019) and NLDN detection efficiency of IC pulses should be lower than 33% which is calculated by the data in 2016 (Zhu et al., 2016), only the IC flashes and strokes are divided by 0.88 and 0.45, respectively**, while CG flashes and strokes are unchanged because of the high detection efficiency.

14) line 142: How is flash rate parameterized?

The flash rate is parameterized by the Price and Rind level of neutral buoyancy parameterization.

> In addition, lightning flash rate **based on the level of neutral buoyancy parameterization (Price and Rind, 1992)** and $LNO_x$ parameterizations are activated (200 mol NO flash$^{-1}$, the factor to adjust the predicted number of flashes is set to 1; hereinafter referred to as "1×200 mol NO flash$^{-1}$").

15) line 175: how is this comparison of pixels computed?

The area of one satellite pixel is about 312 km$^2$. The minimum count of 0.5×0.5 degree grids per box is set to 50 ($\sim$ 1300 km$^2$) which is between 3 satellite pixels ($\sim$ 940 km$^2$) used in Bucsela et al. (2019) and 5 satellite pixels ($\sim$ 1560 km$^2$) set in Pickering et al. (2016). In other words, the number of 50 should be enough to minimize the noise. We have written the sentence to make it more clear.

> Then, it is analyzed in $1° \times 1°$ grid boxes with a minimum of fifty valid $0.05° \times 0.05°$ grids **to minimize the noise. The minimum value is between five satellite pixels in Pickering et al. (2016) and three satellite pixels in Bucsela et al. (2019) or Allen et al. (2019).**

16) line 181: Does the Xu and Randall method consider subgrid convective clouds or only grid-scale cloud based on the microphysics parameterization that generates the gridscale clouds? If it is the latter, this method is not appropriate as a criterion to evaluate model convection.

We applied the Xu and Randall method based on the ice and cloud water fields from both Grell 3D cumulus parameterization and Lin microphysics scheme.

17) line 184: a time period......$LNO_x$ to be detected by

Corrected.

Besides cloud properties, **a** time period and sufficient flashes (or strokes) are required for fresh LNO$_x$ **to be** detected by OMI.

18) lines 184-188: Using these criteria will result in a low bias in the PE results. Bucsela et al. (2019) found that PE is larger at small flash rates. These small flash rates are being discarded here.

Thank you for your remind about the smaller PE at high flash rates. We have added these information at the end of the paragraph.

These criteria will result in a low bias in the PE results, as Bucsela et al. (2019) found that the PE is larger at small flash rates which are discarded here.

19) line 203: daily summations Is this what is done in Pickering et al. (2016)?

Yes. But, we discussed with Eric J. Bucsela recently and he suggested us to redo the linear regressions based on the daily mean LNO$_x$ and flashes. As a result, we have updated the plots of linear regression and analysis related to that.

(1) summation method: dividing the sum of LNO$_x$ by the sum of flashes (or strokes) in each $1° \times 1°$ box in MJJA 2014;
   (2) linear regression method: applying the linear regression to daily **mean values** of LNO$_x$ and flashes (or strokes).

20) line 218:....(Table 2), as the CRF criterion increases from 70% to 90% and to 100%.

Fixed.

Apart from the fewer valid days under higher CRF conditions (CRF $\geq$ 90% and CRF = 100%), LNO$_x$/flash increases from 109.0 $\pm$ 15.3 mol/flash to 114.8 $\pm$ 18.2 mol/flash and decreases again to 99.4 $\pm$ 15.3 mol/flash while LNO$_x$/stroke enhances from 16.7 $\pm$ 2.6 mol/stroke to 17.8 $\pm$ 2.9 mol/stroke and drops again to 15.6 $\pm$ 3.1 mol/stroke (Table 3)**, as the CRF criterion increases from 70% to 90% and to 100%.**

21) line 223:...to derive production per flash (production efficiency, PE).

Fixed.

In order for our results to be comparable with those of Pickering et al. (2016) and Lapierre et al. (2019), we choose NO$_2$ instead of NO$_x$ to derive production **per flash (production efficiency, PE)**.

22) line 228: cloud properties
line 229: PEs

Corrected.

The effect of **cloud properties** on LNO$_x$ production will be discussed in more detail in section 3.3. Generally, the order of estimated daily **PEs** is LNO$_2$Clean > LNO$_2$ > NO$_2$Vis > LNO$_2$Vis.

23) line 259: ratio of CG to IC I don't think there was any particular assumption of this ratio in Pickering et al. (2016).

Yes, you are right. Because Lapierre et al. (2019) found evidence that CG strokes produce more $NO_2$ than IC strokes and Pickering et al. (1996) focused on the Gulf of Mexico, that ratio could lead to the different results. Since the geographic location has been mentioned in the sentence, we have deleted the ratio of CG to IC.

... possibly due to the differences in geographic location, lightning data and chemistry model considered by Pickering et al. (2016) and this study.

24) lines 263-264: this is not obvious from the Figure 5 plots

To make it clear, we have replotted that figure with annotation (Fig. R1).

The $LNO_2$ and $LNO_x$ production are both higher in **the Southeast U.S. (denoted by the red box in Fig. 5 panels, 75°W – 95°W, 25°N – 37°N)**, consistent with Lapierre et al. (2019) and Bucsela et al. (2019).

[Figure]

**Figure R1 .** (a) and (c) Maps of 1°× 1°gridded values of mean $LNO_x$ and $LNO_2$ production per flash with CRF $\geq$ 90% for MJJA 2014. (b) and (d) Same as (a) and (c) except for strokes. The southeastern US is denoted by the red box in panels a – d.

25) line 271: peaks of the LNO$_2$ profile are...

Corrected.

The ratio would also be smaller while **peaks of the LNO$_2$ profile are** below the CP.

26) line 275: from 0.8 to 0.2 as the cloud height increases ...

Thanks. We have corrected that sentence with exact interval of the cloud pressure.

Since the ratio of LNO$_2$Vis to LNO$_2$ decreases from 0.8 to 0.2 as the cloud pressure **decreases from 600 to 300 hPa** ...

27) line 278: LNO$_2$ production (< 30 mol/stroke) occurs

Corrected.

However, smaller LNO$_2$ **production (< 30 mol/stroke) occurs** on all levels between 650 hPa and 200 hPa.

28) line 284: Is this necessarily true? LNO$_x$ production per flash may be smaller in high flash rate storms.

Sorry for the misleading. It should be the LNO$_x$ concentrations instead of LNO$_x$ production per flash. Corrected.

Because updrafts are stronger and flash rates are higher in anomalous storms, **UT LNO$_x$ concentrations** is larger in anomalous than normal polarity storms.

29) lines 287-288: ... (postconvection) in which LNO$_x$ has already been redistributed.........uses LNO$_x$ production profiles.......(Allen et al., 2012; Luo et al., 2017)

Corrected.

There are mainly two methods of distributing LNO$_x$ in models: LNO$_x$ profiles (postconvection) **in which LNO$_x$ has already been redistributed** by convective transport, while the other one (preconvection) **uses LNO$_x$ production profiles** made before the redistribution of convective transport **(Allen et al., 2012; Luo et al., 2017)**.

30) line 296: 2×500 mol NO flash$^{-1}$ This designation can be confusing. Some readers may think you mean 1000 mol/flash

We prefer that designation, because we actually changed the adjusting factor and the LNO production at the same time instead of only changing the LNO production per flash. The definition of 1×200 mol NO flash$^{-1}$ has been used in the model description (Section 2.3). If readers think we use 1000 mol/flash, that should be fine because the effect of that on LNO$_x$ estimation is the same.

31) line 308: different than that....

Corrected.

Note that the LNO$_2$ accounts for a fraction of NO$_2$ above the clouds, the magnitude of increasing denominator could be **different than that** of increasing numerator, resulting in a different effect on the AMF$_{LNO_2}$ and AMF$_{LNO_x}$.

32) line 401-402: we find that the effect is regionally dependent. Both........NO$_2$ cause different comprehensive effects due to nonlinear.....

Thanks. We have rewritten these sentences.

Finally, implementing profiles generated with different model settings of lightning ($1\times200$ mol NO flash$^{-1}$ and $2\times500$ mol NO flash$^{-1}$), we find that **the larger LNO production model setting leads to larger retrieval of LNO$_x$ despite some regionally dependent effects**. Both the ratio of the tropospheric LNO$_2$ above the cloud to the total tropospheric LNO$_2$ and the ratio of LNO$_2$ to NO$_2$ **cause different comprehensive effects due to** the nonlinear calculation of AMF$_{LNO_2}$ and AMF$_{LNO_x}$.

33) Table 2: How many grid boxes per day typically qualify under your criteria?

The mean number of valid grid boxes is 4.2 and 5.3 for flash and stroke data, respectively, as shown in the time series of valid grid boxes (Fig. R2).

[Figure]

**Figure R2 .** Time series of valid grid boxes over the CONUS for MJJA 2014 with the criteria (CRF $\geq$ 90%, entln flashes(strokes) $\geq$ 2400(8160), TL $\geq$ 1000 and ratio $\geq$ 50%).

34) Table 3: Need to indicate how the percentage difference was calculated in the caption. Which one was in the denominator?

The percentage difference was calculated by raising and lowering the perturbed value, this will effectively calculate a mean percent difference. The equation is: $PE_{uncertainty} = (Error_{rising\ perturbed\ value} - Error_{lowering\ perturbed\ value})/2$ where $Error_{perturbed\ value} = (PE_{perturbed\ value} - PE_{original\ value})/PE_{original\ value}$. We have added these information to the caption of that table (Table R1).

**Table R1 .** Uncertainties for the estimation of $LNO_2$/flash, $LNO_x$/flash, $LNO_2$/stroke and $LNO_x$/stroke.

| Type | Perturbation | $LNO_2$/flash[5] | $LNO_x$/flash[5] | $LNO_2$/stroke[5] | $LNO_x$/stroke[5] |
|---|---|---|---|---|---|
| BEHR tropopause pressure[1] | NASA product tropopause | 6 | 4 | 6 | 4 |
| Cloud radiance fraction[1] | ± 5% | 2 | 2 | 2 | 2 |
| Cloud pressure[2] | Constant AMF: 0.46 | 23 | 23 | 23 | 23 |
| Surface pressure[1] | ± 1.5% | 0 | 0 | 0 | 0 |
| Surface reflectivity[1] | ± 17% | 0 | 0 | 0 | 0 |
| $LNO_2$ profile[1] | 2×500 mol NO flash$^{-1}$ | 15 | 29 | 14 | 29 |
| Profile location[1] | Quasi-Monte Carlo | 0 | 1 | 0 | 1 |
| Lightning detection efficiency[3] | IC: ± 16%, CG: ± 5% | 15 | 15 | 15 | 15 |
| $t_{window}$[3] | 2 – 4 hours | 10 | 10 | 8 | 8 |
| $LNO_x$ lifetime[3] | 2 – 12 hours | 24 | 24 | 24 | 24 |
| $V_{strat}$[4] | - | 10 | 10 | 10 | 10 |
| Systematic errors in slant column[4] | - | 5 | 5 | 5 | 5 |
| Tropospheric background[4] | - | 10 | 10 | 10 | 10 |
| $NO/NO_2$[4] | - | 20 | 20 | 20 | 20 |
| Net | - | 48 | 54 | 47 | 54 |

$PE_{uncertainty} = (Error_{rising\ perturbed\ value} - Error_{lowering\ perturbed\ value})/2$ where $Error_{perturbed\ value} = (PE_{perturbed\ value} - PE_{original\ value})/PE_{original\ value}$.
[1]Laughner et al. (2019) [2]Beirle et al. (2009) [3]Lapierre et al. (2019) [4]Allen et al. (2019) and Bucsela et al. (2019) [5]Uncertainty (%)

35) Figures 7 and 8: What do the various colors of shading indicate? What are the scales for the horizontal plot at the top and the vertical plot on the side?

The colors of shading correspond to the regions, which contain the respective normalized density estimated by kernel density estimation, using kdeplot in the Python package named seaborn (https://seaborn.pydata.org/generated/seaborn.kdeplot.html). Because the shading has shown the density, the marginal plots have been removed in case of any confusions (Fig. R3).

[revised manuscript text omitted]

February 7, 2020

We thank the reviewer for his/her positive comments and very careful reading of the main article. The individual corrections suggested are addressed below. The reviewer's comments will be shown in **red**, our response in **blue**, and changes made to the paper are shown in **black** block quotes. Unless otherwise indicated, page and line numbers correspond to the original paper. Figures, tables, or equations referenced as "R$n$" are numbered within this response; if these are used in the changes to the paper, they will be replaced with the proper number in the final paper.

**Major Comments**

1) The authors derive equations for $LNO_x$ by treating clouds as reflecting surfaces. This simplification is required for many radiative transfer models which can not handle multiple scattering in 3d clouds. Thus, several previous studies follow this approach. However, the authors should still be aware of this simplification and state this clearly in the manuscript. Formulations like "$NO_2$ above/below the cloud" are misleading, as for thick thunderstorm clouds, most of the $LNO_x$ is WITHIN the cloud, with a high sensitivity from OMI at the cloud top, gradually decreasing towards the cloud bottom. In this context, the authors should discuss what the "cloud top" derived from OMI $O_2$-$O_2$ measurements actually means for a thunderstorm cloud.

We appreciate the reviewer's suggestion on the "cloud top" and have added the discussion of this concept in Sect. 2.4.

[revised manuscript text omitted]

3) 146: The concept of defining an "AMF" for converting $SNO_2$ into $LNO_x$ was also used in Beirle et al., AMT 9, 1077-1094, 2009, (see eq. 9 therein).

Thank you for your remind of that paper we did not read before. It actually could help us explain our methods and results a lot. We have added that concept before discussing the "Cloud Top" (See *No. 1, Sect. Major comments*).

4) 268: I recommend to extend "Production" to "Production estimates"

Extended.

> The negative differences are caused by background $NO_2$ carried by the updraft while parts of the below-cloud $LNO_2$ results in more $LNO_2$ production **estimates** than $NO_2$Vis production **estimates**.

5) 395: "we find that the regionally dependent effect" - unclear, please revise.

Revised.

> Finally, implementing profiles generated with different model settings of lightning ($1{\times}200$ mol NO flash$^{-1}$ and $2{\times}500$ mol NO flash$^{-1}$), **we find that the larger LNO production model setting leads to larger retrieval of $LNO_x$ despite some regionally dependent effects caused by nonlinear calculation of AMF.**

6) Fig. 1: Please make the legends clearer and remove cryptic labels ("crf90_entin_tl")

After we discussed with Eric J. Bucsela privately, we have redone the linear regressions based on the daily mean $LNO_x$ and flashes, as mentioned in the Response to Reviewer#2 (See *No. 19*). Because the different colors and dots could mislead readers, we have decided to summarize the linear regression results in Table 2 (Table R2).

7) Figs. 2 and 9: Please use less, but larger labels on x-axis.

**Table R2 .** LNO$_x$ production for different combinations of criteria defined in Table 1.

| Condition[1] | ENTLN data type[2] | LNO$_x$/flash or LNO$_x$/stroke | R value | Intercept ($10^6$ mol) | Days[3] |
|---|---|---|---|---|---|
| CRF90_ENTLN | Flash | 52.1 ± 51.1 | 0.20 | 0.21 | 99 |
| CRF90_CF40_ENTLN | Flash | 84.2 ± 31.5 | 0.54 | -0.04 | 70 |
| CRF90_ENTLN_TL1000 | Flash | 61.9 ± 49.1 | 0.27 | 0.33 | 83 |
| CRF90_CF40_ENTLN_TL1000 | Flash | 63.4 ± 52.9 | 0.38 | 0.26 | 38 |
| CRF90_ENTLN_TL1000_ratio50 | Flash | 54.5 ± 48.1 | 0.25 | 0.39 | 81 |
| CRF90_CF40_ENTLN_TL1000_ratio50 | Flash | 90.0 ± 65.0 | 0.46 | 0.15 | 32 |
| CRF90_ENTLN | Stroke | 6.7 ± 4.1 | 0.31 | 0.23 | 102 |
| CRF90_CF40_ENTLN | Stroke | 10.3 ± 3.6 | 0.55 | 0.08 | 79 |
| CRF90_ENTLN_TL1000 | Stroke | 7.5 ± 5.1 | 0.29 | 0.38 | 94 |
| CRF90_CF40_ENTLN_TL1000 | Stroke | 8.6 ± 6.2 | 0.39 | 0.27 | 46 |
| CRF90_ENTLN_TL1000_ratio50 | Stroke | 7.0 ± 4.8 | 0.29 | 0.42 | 93 |
| CRF90_CF40_ENTLN_TL1000_ratio50 | Stroke | 8.9 ± 7.0 | 0.39 | 0.31 | 40 |

[1] These conditions are defined in Table 1. [2] The threshold of ENTLN data is 2400 flashes box$^{-1}$ and 8160 strokes box$^{-1}$ during the period of 2.4 h before OMI overpass time. [3] The number of valid days with specific criteria in MJJA 2014.

We have modified the former figure (Fig. 3/ Fig. R1) and deleted the later one as mentioned in the Response to Reviewer#1 (See *No. 22, Sect. Specific Comments*).

[Figure]

[revised manuscript text omitted]
_{\mathrm{LNO}_x\mathrm{Clean}}$ and $AMF_{\mathrm{NO}_2\mathrm{Vis}}$ respectively:

$$AMF_{\mathrm{LNO}_x\mathrm{Clean}} = \frac{(1-f_r)\int_{p_{\mathrm{surf}}}^{p_{\mathrm{tp}}} w_{\mathrm{clear}}(p)LNO_2(p)\,dp + f_r\int_{p_{\mathrm{cloud}}}^{p_{\mathrm{tp}}} w_{\mathrm{cloudy}}(p)LNO_2(p)\,dp}{\int_{p_{\mathrm{surf}}}^{p_{\mathrm{tp}}} LNO_x(p)\,dp} \qquad (3)$$

190

$$AMF_{\mathrm{NO}_2\mathrm{Vis}} = \frac{(1-f_r)\int_{p_{\mathrm{surf}}}^{p_{\mathrm{tp}}} w_{\mathrm{clear}}(p)NO_2(p)\,dp + f_r\int_{p_{\mathrm{cloud}}}^{p_{\mathrm{tp}}} w_{\mathrm{cloudy}}(p)NO_2(p)\,dp}{(1-f_g)\int_{p_{\mathrm{surf}}}^{p_{\mathrm{tp}}} NO_2(p)\,dp + f_g\int_{p_{\mathrm{cloud}}}^{p_{\mathrm{tp}}} NO_2(p)\,dp} \qquad (4)$$

where $f_g$ is the geometric cloud fraction and $LNO_2(p)$ is the modeled $LNO_2$ vertical profile. Besides these AMFs, another AMF called $AMF_{\mathrm{LNO}_2\mathrm{Vis}}$ is developed for comparison later.

$$AMF_{\mathrm{LNO}_2\mathrm{Vis}} = \frac{(1-f_r)\int_{p_{\mathrm{surf}}}^{p_{\mathrm{tp}}} w_{\mathrm{clear}}(p)NO_2(p)\,dp + f_r\int_{p_{\mathrm{cloud}}}^{p_{\mathrm{tp}}} w_{\mathrm{cloudy}}(p)NO_2(p)\,dp}{(1-f_g)\int_{p_{\mathrm{surf}}}^{p_{\mathrm{tp}}} LNO_2(p)\,dp + f_g\int_{p_{\mathrm{cloud}}}^{p_{\mathrm{tp}}} LNO_2(p)\,dp} \qquad (5)$$

195 A full list of definitions of the used AMFs is shown in Appendix A.

$$AMF_{\mathrm{LNO}_2\mathrm{Vis}} = \frac{(1-f_r)\int_{p_{\mathrm{surf}}}^{p_{\mathrm{tp}}} w_{\mathrm{clear}}(p)NO_2(p)\,dp + f_r\int_{p_{\mathrm{cloud}}}^{p_{\mathrm{tp}}} w_{\mathrm{cloudy}}(p)NO_2(p)\,dp}{(1-f_g)\int_{p_{\mathrm{surf}}}^{p_{\mathrm{tp}}} LNO_2(p)\,dp + f_g\int_{p_{\mathrm{cloud}}}^{p_{\mathrm{tp}}} LNO_2(p)\,dp}$$

[revised manuscript text omitted]

---

## Referee Report (RR1)

L31-37: Is there a difference between the IFLUX approach and IFLUX proxy? Why does the IFLUX approach predict a 15% decrease in lightning while the IFLUX proxy predicts a 3.4% / K increase in lightning over the CONUS?

L31-37: How are water mass fluxes related to lightning amounts?

L60: are low → were low

L79: by subtracting temporal average → by subtracting the temporal average

L154-156: I am confused by the sentence beginning with "Although the simulated total flash densities …". I would suggest the following for lines 154-155.
→ simulated total flash densities are higher than observed by ENTLN over the Southeast US and lower than observed in the North Central US (Fig 2). The impact of these biases on LNOx production is discussed and mitigated in sections 3.1 and 3.4.

L164: for freshly produced lightning → to freshly produced lightning

L177: Hence, the tropospheric → Hence, much of the tropospheric

L180: compositions → composition
L180: fraction the NO2 → fraction of the NO2

L195-198: Is it possible to compare the 50 grid box minimum with the 3-5 pixel minima? If no, I would not mention the later minima. If yes, please explain how they relate.

L210-211: You need to state why you continue to use a relatively high flash threshold when it induces a low-bias. What are the advantages of your threshold?

The possible low-bias of your approach begs the question of what the PE would be if you changed your threshold to 1 flash box-1 from 2400 flash box-1. Can this calculation be made with limited effort. If yes, do so. If no, explain why it is not appropriate to use a low threshold.

L216: has a LNOx → has an LNOx

L235: condition of ratio > 50% → condition of more than half of the above-cloud NOx having an LNOx source

L240 needs to be included as part of the previous paragraph.

L240-244: The substantial decrease in PE as the CRF threshold is increased from 90 to 100% is somewhat concerning and not well explained. You hint that this is likely due to the decrease in valid days. Please explain this more clearly.

**Section 3.2**

L300: Improvement of our approach with respect to what?

L308: I don't understand this statement: "Generally, the profiles of background $NO_2$ ratio are C-shape because $LNO_2$ concentrations are higher than background $NO_2$ in the UT". Wouldn't high $NO_2$ concentrations result in lower background ratios?

L316: convections over → convective cases over

L321: The largest relative change is 153.8% among the six grids where the highest clouds occur. Is this what you mean?
-→ The largest change in PE due to changes in methodology (153.8%) occurs at New Orleans, which has the lowest cloud pressure and consequently the smallest visible column.

L333: concentrations is larger → concentrations are larger

L361: "Fortunately, the AMFs and estimated $LNO_2$ change little in that region" Is this a fortuitous coincidence or is this because the profile shape is relatively insensitive to the magnitude of LNOx PE?

L392: Wasn't 0.46 their mean AMF and not their total sensitivity?

L431: Is 20% an uncertainty or a bias?

L458: → 100 – 250 mol per flash which is higher than but overlaps with our estimate

L449: 80 mol per flash is not 50% smaller than your estimate of 90 mol per flash. Please rephrased based on your updated values.

L464: Add sentence supporting your assertion that this method has reduced the sensitivity to background $NO_2$. Presumably, using information from section 3.3.

L466: Unclear, if the method of Pickering et al. overestimates PE due to over-cloud background $NO_2$ in polluted regions as they do an "after-the-fact" 18% adjustment to the PE to account for background pollution.

L468: "we find that the larger production model settings lead to larger retrieval of LNOx …. Be clear as to how important this finding is. How large a bias is induced by the larger production settings?

---

## Author Response (AR2)

**Estimates of Lightning NO$_x$ Production based on High Resolution OMI NO$_2$ Retrievals over the Continental US**

**Response to Anonymous Referee #1**

Xin Zhang, Yan Yin, Ronald van der A, Jeff L. Lapierre, Qian Chen,
Xiang Kuang, Shuqi Yan, Jinghua Chen, Chuan He and Rulin Shi

March 8, 2020

We thank the reviewer for his/her positive comments and very careful reading of the main article. The individual corrections suggested are addressed below. The reviewer's comments will be shown in **red**, our response in **blue**, and changes made to the paper are shown in **black** block quotes. Unless otherwise indicated, page and line numbers correspond to the original paper. Figures, tables, or equations referenced as "R$n$" are numbered within this response; if these are used in the changes to the paper, they will be replaced with the proper number in the final paper.

1) L31-37: Is there a difference between the IFLUX approach and IFLUX proxy?
Why does the IFLUX approach predict a 15% decrease in lightning while the IFLUX proxy predicts a 3.4% / K increase in lightning over the CONUS?

Sorry for the misleading. IFLUX approach is as same as IFLUX proxy. These two values are relative to different regions. 15% decrease is for global lightning while 3.4% / K is limited to the CONUS.

> While **global lightning** is predicted to increase by 5 — 16% over the next century with the CTH approach (Clark et al., 2017; Banerjee et al., 2014; Krause et al., 2014), a 15% decrease in **global lightning** was estimated with IFLUX in 2100 under a strong global warming scenario (Finney et al., 2018).

2) L31-37: How are water mass fluxes related to lightning amounts?

Thanks for pointing this out. As illustrated in the IFLUX method, the lightning amounts are directly related to ice mass flux, although Romps (2019) mentioned that these condensates are ultimately derive from the condensation of water vapor. We have rewritten this sentence.

> They reported that higher CAPE and updraft velocities caused by global warming could lead to the large increases in tropical lightning simulated by CAPE × P proxy, while IFLUX proxy predicts little change in tropical lightning because of the small changes in **the mass flux of ice.**

3) L60: are low –> were low

Fixed.

> But they found LNO$_x$ production to be highly variable and correlations between flash rate densities and LNO$_x$ production **were** low in some cases.

4) L79: by subtracting temporal average –> by subtracting the temporal average

Fixed.

> The tropospheric $NO_x$ background was removed by subtracting **the** temporal average of $NO_x$ at each box where the value was weighted by the number of OMI pixels which meet the optical cloud pressure and CRF criteria required to be considered deep convection but has 1 flash or less instead.

5) L154-156: I am confused by the sentence beginning with "Although the simulated total flash densities ...".
I would suggest the following for lines 154 – 155.
–> simulated total flash densities are higher than observed by ENTLN over the Southeast US and lower than observed in the North Central US (Fig 2). The impact of these biases on $LNO_x$ production is discussed and mitigated in sections 3.1 and 3.4.

Thanks for your suggestion. Improved.

> Simulated total flash densities are higher than ENTLN observations over the Southeast US and lower than observations in the North Central US (Fig. 2). The impact of these biases on $LNO_x$ production is discussed and mitigated in Sect. 3.1 and 3.4.

6) L164: for freshly produced lightning –> to freshly produced lightning

Fixed.

> The concept of $AMF_{LNO_x}$ was also used in Beirle et al. (2009) to investigate the sensitivity of satellite instruments **to** freshly produced lightning $NO_x$.

7) L177: Hence, the tropospheric –> Hence, much of the tropospheric

Fixed.

> Hence, **much of** the tropospheric $NO_2$ measured by OMI lies inside the cloud rather than above the cloud top.

8) L180: compositions –> composition
L180: fraction the $NO_2$ –> fraction of the $NO_2$

Fixed.

> The sensitivity study of Beirle et al. (2009) compared the chemical **composition** from the cloud bottom to the cloud top and revealed that a significant fraction **of** the $NO_2$ within the cloud originating from lightning can be detected by the satellite.

9) L195–198: Is it possible to compare the 50 grid box minimum with the 3–5 pixel minima?
If no, I would not mention the later minima. If yes, please explain how they relate.

50 grid boxes are approximately equal to 3–5 pixels. We have decided to remove the later minima which may confuse readers.

10) L210-211: You need to state why you continue to use a relatively high flash threshold when it induces a low-bias. What are the advantages of your threshold?

The possible low-bias of your approach begs the question of what the PE would be if you changed your threshold to 1 flash box$^{-1}$ from 2400 flash box$^{-1}$.

Can this calculation be made with limited effort. If yes, do so. If no, explain why it is not appropriate to use a low threshold.

There are some reasons why we prefer the relatively higher threshold and we have added the explanation in Sect. 2.5.

According to your advice, we have also checked the 1 flash box$^{-1}$ condition which is applied to both ENTLN and WRF lightning data and added to the Appendix B.

**2.5: Procedures for Deriving LNO$_x$**

... Since our study focuses on developing a new AMF and compare results with other works using the similar lightning thresholds (Lapierre et al., 2020; Pickering et al., 2016), we will only discuss results based on the strict criteria in the main texts. For comparisons between 2400 flashes box$^{-1}$ criterion and 1 flash box$^{-1}$ criterion, scatter diagrams using different lightning criteria are presented in Appendix .

**Appendix B: LNO$_x$ Production based on Lower Lightning Thresholds**

While we used 2400 flashes box$^{-1}$ and 8160 strokes box$^{-1}$ per 2.4 hour time window for detecting LNO$_x$, here we show results obtained when using 1 flash box$^{-1}$ and 3.4 strokes box$^{-1}$ in the same time window. We note that the WRF total lightning threshold is also reduced to 1 flash box$^{-1}$, but we keep the ratio condition unchanged. Briefly, the condition is CRF90_ENTLN1(3.4)_TL1_ratio50 as shown in Table 1.

Similarly, the order of estimated daily PEs is LNO$_2$Clean > LNO$_2$ > NO$_2$Vis > LNO$_2$Vis (Fig. B1 [see Fig. R1]). Compared with Fig. 4, the LNO$_2$ per flash and LNO$_x$ per flash are larger while PEs based on stroke data are smaller. Considering the additional boxes of fewer lightning counts, differences in the daily mean flashes and NO$_x$ results in different PEs and the relationship presents more like the power function as mentioned in Bucsela et al. (2019).

Instead of using the nonlinear regression of power function:

$$y = \alpha x^\beta \tag{1}$$

where $x$ is flashes or strokes and $y$ is NO$_2$ or NO$_x$, we take the logarithm of both sides and apply the linear regression to data:

$$\log_{10} y = \log_{10} \alpha + \beta \log_{10} x \tag{2}$$

As expected, the linear regression based on logarithmized data performs better in this situation and yields $\alpha$ = 38 kmol, and $\beta$ = 0.3 for LNO$_x$ per flash (Fig. B2 [see Fig. R2]). Since we use the unbinned data (flashes not divided into many groups), we compare our results with Bucsela et al. (2019) based on the same kind of data ($\alpha$ = 10.3 kmol, and $\beta$ = 0.42). The large difference of $\alpha$ is related to the method of estimating LNO$_x$, different lightning data and different regions. Note that the resolution (13 × 24 km$^2$) of OMI could weaken the signal of LNO$_x$. We believe the phenomenon of higher production efficiency as flash rate decreases (Fig. B3 [see Fig. R3]) could be explored in much detail with higher resolution data like the TROPOMI data.

11) L216: has a LNO$_x$ –> has an LNO$_x$

Fixed.

The ratio ≥ 50% indicates that more than half of the NO$_x$ above the cloud must has **an** LNO$_x$ source.

12) L235: condition of ratio > 50% –> condition of more than half of the above-cloud $NO_x$ having an $LNO_x$ source.

Fixed.

> The result is almost the same as that under the CRF90_ENTLN_TL1000 condition which is without **the condition of more than half of the above-cloud $NO_x$ having an $LNO_x$ source**.

13) L240 needs to be included as part of the previous paragraph.

Moved.

14) L240-244: The substantial decrease in PE as the CRF threshold is increased from 90 to 100% is somewhat concerning and not well explained.
You hint that this is likely due to the decrease in valid days. Please explain this more clearly.

This is caused by the higher lightning density with fewer $LNO_x$ (Fig. R4). We have added it.

> When the CRF increases from 90% to 100%, the $LNO_x$ PE decreases because of the higher lightning density with fewer $LNO_x$ (not shown). The increment of $LNO_x$ PE caused by the CRF increase from 70% to 90% is opposite to the result of Pickering et al. (2016).

15) L300: Improvement of our approach with respect to what?

We have made it more clear.

> **With respect to the $LNO_2$ production**, the patterns in Fig. 6 indicate the improvement of our approach is different in polluted and clean regions.

16) L308: I don't understand this statement: "Generally, the profiles of background $NO_2$ ratio are C-shape because $LNO_2$ concentrations are higher than background $NO_2$ in the UT".
Wouldn't high $NO_2$ concentrations result in lower background ratios?

The UT is mostly located between the OMI detected cloud pressure (lower black horizontal line, Fig. 7) and tropopause (higher black horizontal line, Fig. 7). The trough of C-shape profile of background ratio ($[NO_2 - LNO_2]/NO_2$) is due to the higher $LNO_2$ concentrations than background $NO_2$ there.

> Generally, the profiles of **the ratio of background $NO_2$ over total $NO_2$** are C-shape because **UT** $LNO_2$ concentrations are higher than **UT** background $NO_2$ concentrations.

17) L316: convections over –> convective cases over

Fixed.

> As a result, our approach is less sensitive to background $NO_2$ and more suitable for **convective cases** over polluted locations.

18) L321: The largest relative change is 153.8% among the six grids where the highest clouds occur. Is this what you mean?
–> The largest change in PE due to changes in methodology (153.8%) occurs at New Orleans, which has the lowest cloud pressure and consequently the smallest visible column.

Yes, you are right. We have rewritten the sentence according to your suggestion.

> The largest relative change (153.8%) occurs at New Orleans, which has the lowest cloud pressure and consequently the smallest visible column.

19) L333: concentrations is larger –> concentrations are larger

Fixed.

> Because updrafts are stronger and flash rates are higher in anomalous storms, UT $LNO_x$ concentrations **are** larger in anomalous than normal polarity storms.

20) L361: "Fortunately, the AMFs and estimated $LNO_2$ change little in that region".
Is this a fortuitous coincidence or is this because the profile shape is relatively insensitive to the magnitude of $LNO_x$ PE?

This is caused by the relatively insensitive to the magnitude of $LNO_x$ PE, because the flash density could affect the value of AMF. We have made it clearer after the sentence.

> Because the Southeast U.S. has the highest flash density (Fig. 2), the $NO_2$ in the numerator of AMF is dominated by $LNO_2$. Both the SCD and VCD will increase when the model uses higher $LNO_2$. In other words, the sensitivity to the LNO setting decreases and the relative distribution of $LNO_2$ matters.

21) L392: Wasn't 0.46 their mean AMF and not their total sensitivity?

Yes, 0.46 is the mean AMF which is named sensitivity in their paper.
Note that we have used another method to estimate the uncertainty caused by CP (See *No. 1, Response to Reviewer #3*).

22) L431: Is 20% an uncertainty or a bias?

Thanks for the correction, that is a bias. We have recalculated the uncertainty using the bias (See *No. 1, Response to Reviewer #3*).

> We assign **a 20% bias with $\pm$ 15% uncertainty** to this error considering the possible positive $NO_2$ measurements interferences (Allen et al., 2019; Bucsela et al., 2019) and **estimate the uncertainty to be 15% for $LNO_x$ PE**.

23) L448: –> 100 – 250 mol per flash which is higher than but overlaps with our estimate

Fixed.

> Bucsela et al. (2010) estimated $LNO_x$ PE of 100 – 250 mol/flash **which is higher than but overlaps with our estimate**.

24) L449: 80 mol per flash is not 50% smaller than your estimate of 90 mol per flash.
Please rephrased based on your updated values.

The linear regression in Pickering et al. (2016) is based on daily summed $NO_x$ and flashes. We used this method before and switched to daily mean values now. As a result, it is better to compare with their results using the same method. We have pointed this special background below:

Pickering et al. (2016) estimated LNO$_x$ PE to be 80 $\pm$ 45 mol per flash for the Gulf of Mexico. This is 50% smaller than our flash-based results over the CONUS, **if we use the same linear regression method which is based on the daily summed values instead of daily mean values**.

25) L464: Add sentence supporting your assertion that this method has reduced the sensitivity to background NO$_2$. Presumably, using information from section 3.3.

L466: Unclear, if the method of Pickering et al. overestimates PE due to over-cloud background NO$_2$ in polluted regions as they do an "after-the-fact" 18% adjustment to the PE to account for background pollution.

The reduce of the sensitivity to background NO$_2$ is inferred from the comparison between our results and others. We have rewritten these conclusions to make them clearer.

Compared with Lapierre et al. (2020), we find that the LNO$_2$ production could be larger when the below-cloud LNO$_2$ is taken into account, especially for the high clouds. Meanwhile, if the method of Pickering et al. (2016) is applied without the background NO$_2$ correction, the derived LNO$_x$ production efficiency is similar to ours in clean regions or regions with high LNO$_2$ concentration above the cloud, but it could be overestimated more than 18% in polluted regions.

27) L468: "we find that the larger production model settings lead to larger retrieval of LNOx ..."
Be clear as to how important this finding is. How large a bias is induced by the larger production settings?

Fixed.

[revised manuscript text omitted]

March 8, 2020

We thank the reviewer for his/her positive comments and very careful reading of the main article. The individual corrections suggested are addressed below. The reviewer's comments will be shown in **red**, our response in **blue**, and changes made to the paper are shown in **black** block quotes. Unless otherwise indicated, page and line numbers correspond to the original paper. Figures, tables, or equations referenced as "R$n$" are numbered within this response; if these are used in the changes to the paper, they will be replaced with the proper number in the final paper.

1) In the response to Referee #1:
Items 3 and 7:
Pickering et al. (2016) did subtract a tropospheric background. It was not done as part of the algorithm, but it was done as a final adjustment to the PE. In Section 6 of the Pickering et al. (2016) paper, 18% was subtracted from the derived PE value of 97 mol/flash which led to 80 mol/flash as the final value. The 18% was the mean of the two estimates of background derived from aircraft flights in the Gulf of Mexico region (3% and 33%). It needs to be made clear in the manuscript that Pickering et al. (2016) did not ignore the influence of background NOx.

Thanks. We have corrected these according to your advice.

> Since they considered NO$_2$ above the cloud as LNO$_2$ in the algorithm due to the difficulty and uncertainty in determining the background NO$_2$, their AMF and derived VCD of LNO$_x$ (LNO$_2$) is named as AMF$_{\text{LNO}_x\text{Clean}}$ (AMF$_{\text{LNO}_2\text{Clean}}$) and LNO$_x$Clean (LNO$_2$Clean), respectively. **Note that Pickering et al. (2016) considered the two estimates of background derived from aircraft flights in the Gulf of Mexico region (3% and 33%) and subtracted the mean value (18%) from the estimated mean LNO$_x$ production efficiency (PE) for the background bias. However, we use the original algorithm directly without correction to distinguish the effect of different AMFs on LNO$_x$ estimation in the remainder of this paper.**

2) In the response to Referee #1:
Item 4: The revision still sounds like (3% to >30%) is a range of uncertainty. The authors should say that this is the range of estimated background amounts.

Thank you for your remind. We have fixed it and merged it with the method of Pickering et al. (2016) (See *No. 1*).

3) Response to Referee #2:

Item 14:

The WRF-Chem level of neutral buoyancy parameterization stems from Price and Rind, but the details are based on Wong et al. (2013) Geosci. Model Dev., 6, 429–443, 2013 www.geosci-model-dev.net/6/429/2013/ doi:10.5194/gmd-6-429-2013. Please add the Wong et al. reference.

Thanks, added.

In addition, lightning flash rate based on the level of neutral buoyancy parameterization **(Price and Rind, 1992; Wong et al., 2013)** and $LNO_x$ parameterizations are activated (200 mol NO flash$^{-1}$, the factor to adjust the predicted number of flashes is set to 1; hereinafter referred to as "1×200 mol NO flash$^{-1}$").

[revised manuscript text omitted]